# LLaRA: Supercharging Robot Learning Data for Vision-Language Policy

**Xiang Li**[1]  **Cristina Mata**[1]  **Jongwoo Park**[1]  **Kumara Kahatapitiya**[1]
**Yoo Sung Jang**[1]  **Jinghuan Shang**[1]  **Kanchana Ranasinghe**[1]  **Ryan Burgert**[1]
**Mu Cai**[2]  **Yong Jae Lee**[2]  **Michael S. Ryoo**[1]

[1]Stony Brook University    [2]University of Wisconsin-Madison

xiangli8@cs.stonybrook.edu

## ABSTRACT

Vision Language Models (VLMs) have recently been leveraged to generate robotic actions, forming Vision-Language-Action (VLA) models. However, directly adapting a pretrained VLM for robotic control remains challenging, particularly when constrained by a limited number of robot demonstrations. In this work, we introduce LLaRA: **L**arge **L**anguage **a**nd **R**obotics **A**ssistant, a framework that formulates robot action policy as visuo-textual conversations and enables an efficient transfer of a pretrained VLM into a powerful VLA, motivated by the success of *visual instruction tuning* in Computer Vision. First, we present an automated pipeline to generate *conversation-style* instruction tuning data for robots from existing behavior cloning datasets, aligning robotic actions with image pixel coordinates. Further, we enhance this dataset in a self-supervised manner by defining six auxiliary tasks, without requiring any additional action annotations. We show that a VLM finetuned with a limited amount of such datasets can produce meaningful action decisions for robotic control. Through experiments across multiple simulated and real-world tasks, we demonstrate that LLaRA achieves state-of-the-art performance while preserving the generalization capabilities of large language models. The code, datasets, and pretrained models are available at https://github.com/LostXine/LLaRA.

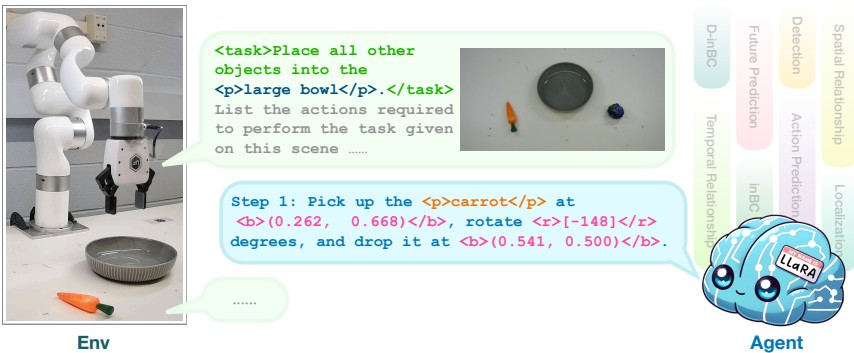

Figure 1: **A real-world demonstration of LLaRA solving an unseen task.** In this setting, LLaRA converts only eight thousand simulated expert trajectories from VIMA (Jiang et al., 2023) into a strong instruction-tuning data and generates six auxiliary datasets in a self-supervised manner. With such data and about one thousand additional in-domain images, a pretrained Vision Language Model (VLM) is finetuned into a powerful robot policy that can solve many unseen real-world tasks. At inference, the VLM is queried to generate actions in natural language where the robot action is aligned to image pixel coordinates.

## 1 INTRODUCTION

Large Language Models (LLMs) such as GPT-4 (OpenAI, 2023), Llama (Touvron et al., 2023; AI, 2024), and Gemini (Anil et al., 2023) exhibit unprecedented abilities across diverse language tasks. Such LLMs are often trained together with visual data by taking advantage of pretrained

vision encoders, forming powerful Vision Language Models (VLMs). Effective and efficient training of such VLMs significantly depends on the style and format of the vision-language data used for training, leading to the exploration of various *instruction tuning* (Taori et al., 2023; Chiang et al., 2023; Liu et al., 2023a; 2024b) strategies. The resulting VLMs exhibit strong vision-language skills, but not without limitations such as spatial awareness (Chen et al., 2023; Ranasinghe et al., 2024b) or niche-domain understanding. The latter has led to domain-specific visual instruction tuning (Li et al., 2023; Kuckreja et al., 2024; Hong et al., 2023).

Several robot learning approaches also leverage pretrained LLMs / VLMs (Zeng et al., 2022; Brohan et al., 2023c; Driess et al., 2023; Vemprala et al., 2024; Kim et al., 2024; Niu et al., 2024; Zheng et al., 2024) with recent work focusing on low-level robot actions (Brohan et al., 2023b;a; Padalkar et al., 2023). The strong performance of such approaches can be attributed to the extensive world knowledge (*e.g.*, understanding of physics, human common-sense) (Yu et al., 2023; Zhao et al., 2024) and powerful reasoning abilities (Creswell & Shanahan, 2022; Liu et al., 2023b) of underlying language models. However, studies on curating and using conversation-style data for instruction tuning in robotics have been rather limited (Zhen et al., 2024).

Motivated by the promising attributes of VLMs, we explore a process, called *Visuomotor Instruction Tuning*, of adapting a VLM to a robot action policy that can handle diverse visuomotor control challenges. More specifically, we transform a typical behavior cloning (BC) dataset into a conversational dataset, *Instruct-BC* (*inBC*) for VLM finetuning. Given a state described in visual-textual modalities, our VLM is trained to generate suitable actions in text, aligned to image pixel coordinates. Such a formulation based on conversation-style instruction-response data enables us to convert a VLM into a robot action policy effortlessly. However, we also note that the effectiveness of an instruction-tuned VLM depends heavily on the quality of data formulation (Liu et al., 2023a; Ranasinghe et al., 2024b; Kuckreja et al., 2024), which is non-trivial to automate (or, scale) for new domains (Li et al., 2023; Kuckreja et al., 2024; Hong et al., 2023; Chen et al., 2023; Ranasinghe et al., 2024b). Therefore, to strengthen (or, *supercharge*) such data, we construct auxiliary datasets based on the same BC dataset, in a self-supervised fashion without any extra action label.

Our overall framework termed **LLaRA** (**L**arge **L**anguage **a**nd **R**obotics **A**ssistant) generates visuomotor instruction data that can efficiently and effectively finetune a VLM into a robot action policy. It is motivated by LLaVA (Liu et al., 2023a), which is designed primarily for vision tasks. Similarly, LLaRA offers the entire framework of data generation, model formulation, and training pipeline for VLMs, now specialized for robot learning. Our key contributions are:

1. Formulating robot manipulation tasks into instruction-response pairs described in natural language, which enables successful instruction tuning of a VLM as a policy.

2. Aligning robot actions to image pixel coordinates, makes the transfer of a pretrained VLM to the robotics domain more efficient, particularly when the training data is limited.

3. Identifying and generating auxiliary data that further enhances robot policy learning in a self-supervised manner.

We conduct extensive experiments on the proposed framework to establish the effectiveness of both our automated data generation pipeline and instruction-tuned VLM in solving robotics tasks.

## 2 RELATED WORK

**Instruction data generation.** Finetuning LLMs / VLMs with instruction data has shown a lot of potential for vision tasks (Taori et al., 2023; Chiang et al., 2023; Liu et al., 2023a; 2024b). While effective construction of such datasets with high quality and quantity is beyond trivial (Liu et al., 2023a), especially for specialized visual domains (Li et al., 2023; Bazi et al., 2024; Kuckreja et al., 2024; Thawkar et al., 2023) such as robotics. Yet these modified VLMs suffer in the robotics domains (Ranasinghe et al., 2024a).

**Spatial reasoning in VLMs.** Several recent works investigate how VLMs can be modified for spatial awareness within images (Zhang et al., 2023; Zhao et al., 2023; Zang et al., 2023; Peng et al., 2023; Chen et al., 2023; Ranasinghe et al., 2024b; You et al., 2023; Cai et al., 2024; Shtedritski et al., 2023), which is a critical ability of a visuomotor policy. One line of work explores prompting using textual or specialized tokens to encode locations, followed by instruction tuning on localization-specific datasets (Peng et al., 2023; Chen et al., 2023; Ranasinghe et al., 2024b; You et al., 2023). In

contrast, our framework extends beyond localization to directly predict robot actions. An alternate line of work explores visual prompting but without any focus on robot learning (Shtedritski et al., 2023; Cai et al., 2024). An extension of such ideas to robotics is explored in PIVOT (Nasiriany et al., 2024). These ideas are complementary to LLaRA, where additional visuomotor instruction tuning directly generates actions optimal for robotics tasks.

**Robot learning via sequence models, LLMs, and VLMs.** The use of sequence models in robot learning has a longstanding tradition (Zheng et al., 2022; Shang et al., 2022a;b). Recent foundation LLMs / VLMs provide a source of common-sense data for robot policy training (Qian et al., 2024; Ingelhag et al., 2024; Wu et al., 2023b; Yoneda et al., 2023). More recent work such as Robotics Transformer (Brohan et al., 2023b;a; Padalkar et al., 2023), Gato (Reed et al., 2022), GR-1 (Wu et al., 2023a) and Octo (Octo Model Team et al., 2024) learn generalist robot policies with multi-modal sequence models, taking advantage of the flexibility of encoding and decoding images/actions via token representations similar to VLMs.

PALM-E (Driess et al., 2023) utilizes a pretrained VLM for embodied reasoning and planning. However, it assumes access to additional policies capable of performing low-level skills from a limited vocabulary, with the VLM generating natural language that conditions these low-level commands. SpatialVLM (Chen et al., 2024) focuses on enhancing the spatial reasoning ability of the VLM, resulting in a VLM that serves as a spatial relationship predictor rather than a robot policy. To embed SpatialVLM into a policy, chain-of-thought spatial reasoning is required.

There are concurrent works (Kim et al., 2024; Yuan et al., 2024; Niu et al., 2024; Zheng et al., 2024) which also utilize a LLaVA style VLM as a robot policy. All such approaches employ a VLM as a robot action policy, processing both visual observations and textual instructions as inputs. OpenVLA (Kim et al., 2024) employs specialized tokens to represent quantized actions, a method akin to RT-2 (Brohan et al., 2023a). OpenVLA also incorporates DINOv2 (Oquab et al., 2023) as an additional visual encoder along with SigLIP (Zhai et al., 2023), compared to a single CLIP (Radford et al., 2021) vision encoder used in LLaVA. RoboPoint (Yuan et al., 2024) introduces a novel point-based action space and a scalable data pipeline tailored for spatial affordance prediction. Actions are delineated via points on an RGB image, which are subsequently translated to 3D space using depth data. This approach eliminates the reliance on predefined action primitives (Liang et al., 2023; Singh et al., 2023), external object detectors (Huang et al., 2023b; Liu et al., 2024a), and iterative visual prompting (Nasiriany et al., 2024). LLARVA (Niu et al., 2024), differs by generating both 2D visual traces in image coordinates and corresponding textual actions as outputs, with the former functioning as an auxiliary task.

In contrast, our method describes robot action using image coordinates in natural language, which creates a more intuitive link between vision and action without modifying the existing VLM model architecture. This enables a much more efficient transfer of the pretrained VLM to our robot VLA model. Moreover, all the aforementioned studies lack of comprehensive investigation into the methods for generating self-supervised *auxiliary datasets* from existing robot data, as well as the implications of integrating such datasets for the training. Our work LLaRA distinctively addresses this gap, marking a critical divergence from the concurrent studies.

**Self-supervised learning in robotics.** Self-supervised learning (SSL) has long been explored in robotics (Pinto & Gupta, 2016; Sermanet et al., 2018; Yarats et al., 2021; Laskin et al., 2020; Schwarzer et al., 2021), with recent work showing a flavor of visual representation learning (Li et al., 2022; 2024b; Wang et al., 2023). We introduce auxiliary SSL objectives (*i.e.*, pretext tasks) based on instruction prompts, motivated by similar ideas in computer vision (Doersch et al., 2015; Yun et al., 2022; Ranasinghe et al., 2022; Walker et al., 2021). More specifically, we create a set of domain-specific auxiliary datasets (*e.g.*, action or future prediction), that enable LLaRA to understand information that is useful for the downstream robotics tasks, when used for finetuning.

## 3 PRELIMINARY: VISUAL INSTRUCTION TUNING

Visual Instruction Tuning is a framework to finetune a large Vision Language Model (VLM) using task-specific instructions. The objective is to develop a general-purpose multimodal model capable of solving diverse vision tasks described by language (Huang et al., 2023a). A typical example of

Visual Instruction Tuning is LLaVA (Liu et al., 2023a), whose model consists of three components: a large language model $\theta_{\text{LLM}}$, a visual encoder $\theta_V$ and an adapter layer $\theta_{\text{MLP}}$ (see Fig. 2).

Consider a conversation data sample that contains a single image $x_v$ and user instruction text $x_l$ as inputs. During inference, the visual encoder $\theta_V$ and adapter layer $\theta_{\text{MLP}}$ process $x_v$ successively to produce visual tokens that can be directly concatenated with the textual tokens of $x_l$ in language embedding space of $\theta_{\text{LLM}}$. Next, $\theta_{\text{LLM}}$ autoregressively generates new tokens conditioned on both the visual tokens from $x_v$ and text tokens from $x_l$. Finally, generated tokens are decoded into natural language as the output of the model.

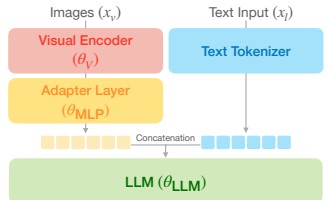

Figure 2: **LLaVA overview.** A Large Language Model (LLM) is connected to the visual domain with suitable encoder and adaptor neural networks.

A two-phase training process is executed in Liu et al. (2023a) with a next-token prediction objective over the $\theta_{\text{LLM}}$ output in each phase. First, only $\theta_{\text{MLP}}$ is trained using image-caption pairs (similar to datasets for CLIP (Radford et al., 2021) training). Then, both $\theta_{\text{MLP}}$ and $\theta_{\text{LLM}}$ are finetuned using a specialized instruction tuning dataset termed `LLaVA_Instruct_150k`. `LLaVA_Instruct_150k` contains human-style instruction-response conversations, automatically created by a powerful LLM (OpenAI, 2023) from a generic object detection dataset. This *instruction tuning* process is what we focus on in this paper. We start from a pretrained LLaVA model and only perform a single-stage finetune, updating $\theta_{\text{LLM}}$ and $\theta_{\text{MLP}}$ based on the instruction tuning data formulation we introduce in this paper.

## 4 VISUOMOTOR INSTRUCTION TUNING

We leverage large Vision Language Models (VLMs) as a generalist to address diverse visuomotor control challenges. Specifically, we transform a typical behavior cloning dataset into an instruction tuning dataset and subsequently finetune a pretrained VLM on this tailored dataset, a process we call *Visuomotor Instruction Tuning*. The resulting LLaRA framework benefits from the broad, inherent knowledge embedded within the VLM and the alignment between robot action and image pixel coordinates, enabling more efficient visuomotor policy learning.

In this section, we present our LLaRA framework to *a)* convert a set of robot manipulation expert trajectories into a visuomotor instruction tuning dataset, and *b)* finetune a VLM on such a dataset as a robot policy. In the next section, we show that *c)* such VLMs can be further improved for robot control using auxiliary instruction tuning data created in a self-supervised manner.

**Problem formulation.** We consider a behavioral cloning (BC) setting over a Markov Decision Process (MDP), described by the tuple $(\mathcal{S}, \mathcal{A}, P)$. Here, $s_t \in \mathcal{S}$ denotes the state at timestamp $t$, $a_t \in \mathcal{A}$ represents the action at $t$, and $P$ encapsulates the transition dynamics, expressed as $P(s_{t+1}|s_t, a_t)$. Our goal is to finetune a VLM as a robot policy $\pi$ that best recovers an unknown policy $\pi^*$ using a demonstration dataset containing multiple tasks $\mathcal{D} = \{(s^i, a^i)\}$ collected by $\pi^*$. The policy $\pi$ is designed to be conditioned on the task description, and our empirical findings suggest adding historical actions is a beneficial condition. Consequently, the policy can be articulated as $\pi(a_t^i|s_t^i, a_h^i)$, where $h = 1, 2, ..., t - 1$, indicating that $\pi$ takes into account all preceding actions.

**Instruction tuning data from trajectories.** Next, we turn a set of expert trajectories $\mathcal{D}$ into a visuo-motor instruction tuning dataset. For each state transition, we convert the state action pair $(s_t, a_t)$ into a single-round conversation. The current visual observation and textual task description, forming $s_t$, can be directly assimilated by a VLM as the user instruction. However, the numerical action $a_t$ needs conversion into textual format to be generated by a VLM. In contrast to the approach employed by GATO (Reed et al., 2022), RT-2 (Brohan et al., 2023a), and OpenVLA (Kim et al., 2024), which utilizes special tokens to encode quantized action values (binning tokenization), we adopt *2D Image Coordinates*—normalized to $[0, 1]$—to represent the positions in actions. Additionally, any rotational component of the action, denoted as $a_r$ (such as the angle of a joint), is expressed textually as "`rotate <r>`$a_r$`</r> degrees`". To further improve the performance, we incorporate the *history of executed actions* into the query and structure the response to predict *multiple future action steps* until the episode concludes. More details are covered in Appendix A.1. Our approach creates an intuitive connection between visual input and robotic actions, enabling effortless transfer across various robotic embodiments. By implementing these strategies, we effectively convert a trajectory

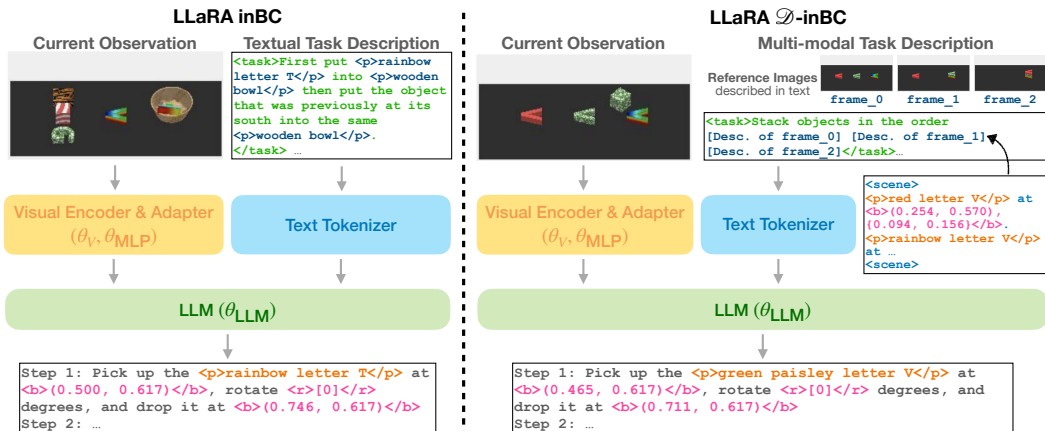

Figure 3: **Example data for visuomotor instruction tuning.** (left) *inBC* is our instruction tuning data created from BC, only taking textual task description. (right) $\mathcal{D}$-*inBC* is designed for multi-modal task description that considers reference images by taking their corresponding textual description. Both *inBC* and $\mathcal{D}$-*inBC* belong to the LLaRA family.

into a format comprising both image and text, which is readily processed by the VLM. We name the converted dataset **Instruct-BC (*inBC*)** and present examples in Fig. 3, Tab. 8 and Tab. 9.

**Aggregate multiple images for a single-image VLM.** The VLM policy we use here closely follows the design of LLaVA-1.5 (Liu et al., 2024b). The model consumes a *single* image and language instruction and generates language output. In scenarios where a single observation $s_t$ comprises an observation image and multiple reference images, systems such as LLaVA suffer from low performance because it was never trained on multiple images (see Tab. 15). Instead, we convert each additional reference image into a language description with the help of object detection on the *additional images* (*e.g.*, images in the task description, rather than the main visual observation). The main image is still directly used as the visual input by the VLM without any changes. Compared to *inBC*, we name this method that takes additional object detection results as **Description-Instruct-BC ($\mathcal{D}$-*inBC*)**, and our model trained on this data is referred to with the same name as earlier. Examples are also available in Fig. 3, Tab. 8 and Tab. 9. In Appendix A.1.3 we also discuss another variant of $\mathcal{D}$-*inBC* that uses natural language instead of structured text to describe the reference images: $\mathcal{D}$-*inBC* (L).

**Inference pipeline.** During inference, the prompt for the VLM is prepared using the same template as either *inBC* or $\mathcal{D}$-*inBC*. As illustrated in Fig. 3, for a model trained on *inBC*, each conversation turn encapsulates the current visual observation, the task description, and the previous actions described in the text. For a model trained on $\mathcal{D}$-*inBC*, the process starts with object detection to convert any *extra* images from the task description into text. The task description, now including the detection outcomes, is then combined with the current visual observation and the action history to get the complete query. Then LLaRA generates the text output that contains numerical values (*e.g.*, 2D image coordinates) for a robot action. The extracted rotation angle can be directly mapped to the rotation of the end effector. The 2D image coordinates will be further converted to robot action space via predefined mapping, which can be estimated using visual calibration in both simulated and real-world environments. After that, the robot executes the action, and a new observation can be captured from the environment. This initiates another round of interaction, setting the stage for the subsequent action.

## 5 SUPERCHARGING VISUOMOTOR INSTRUCTION TUNING DATASET

Motivated by the success of LLaVA (Liu et al., 2023a) instruction tuning data, and subsequent domain-specific instruction tuning datasets (*e.g.*, for reasoning, grounding, or referring) (Chen et al., 2023; Ranasinghe et al., 2024b; Peng et al., 2023), LLaRA creates auxiliary robotics instruction tuning datasets that enhance a VLM in a self-supervised manner.

More specifically, given an expert trajectory, we generate conversations that reveal useful information for learning a policy implicitly via scripted templates. In addition to visual observations, we make use of automatically generated object labels, locations, and geometries (*e.g.*, rotation) together with task descriptions as expert (or oracle) information. We primarily introduce six auxiliary

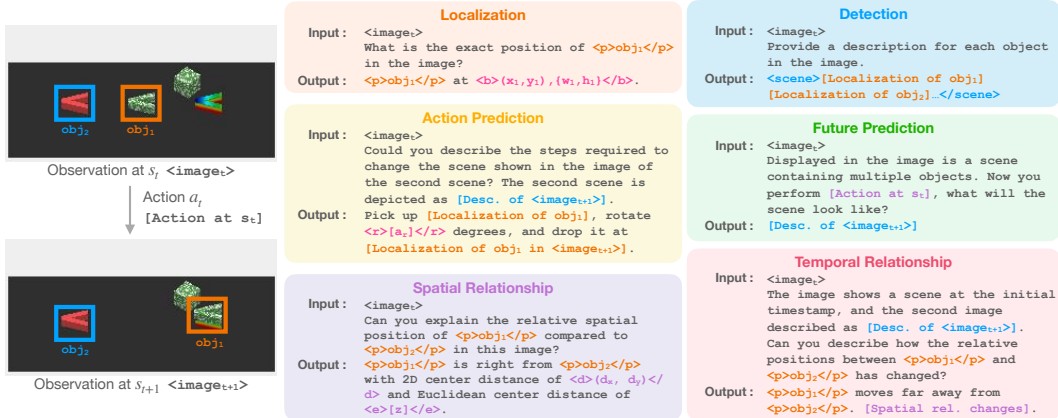

Figure 4: **Auxiliary datasets for visuomotor instruction tuning.** Given an input trajectory, we make use of expert information (*e.g.*, object detections) to formulate conversations related to auxiliary semantics. Please refer to Tab. 11 and Tab. 12 for the actual examples of each dataset.

datasets, augmenting existing *inBC* or *D-inBC* to better finetune VLMs. In each dataset, we consider a standard instruction, which is further rephrased multiple times using GPT-4 (OpenAI, 2023) to infuse a reasonable diversity. The answers follow simple rule-based templates that format expert information as expected responses for each conversation. The following paragraphs briefly describe each auxiliary dataset. We kindly refer the readers to Fig. 4 and Appendix A.1.4 for formatting details and qualitative samples.

**Localization and detection.** To enhance the visual grounding ability of VLM, we first generate single-turn conversations for both localization of individual objects and higher-level detection of entire scenes. For localization, each conversation involves pinpointing an object within an observation, where the query specifies the object's location and the response is given in the format "`<p>OBJ</p> at (x, y), {w, h}`". Here, $(x, y)$ indicates the midpoint of the object's bounding box, while $\{w, h\}$ denotes its width and height, with coordinates normalized to $[0, 1]$ as detailed in Sec. 4. For detection, the setup extends to multiple objects, describing the scene via a list of bounding boxes.

**Action and future prediction.** These datasets are designed to improve the VLM's capability to understand dynamics. In the action prediction dataset, given an initial observation, the query includes a text description of the subsequent observation, formatted similarly to the detection dataset. The VLM identifies the specific action that connects these two states. For future prediction, the roles of query and answer are reversed: the query now presents a single-step action and requests a text description of the next observation.

**Spatial and temporal relationships.** We further enrich the dataset with the relationships between objects. For spatial relationships, given two objects in an image, we format a conversation to address their 2D configuration (*e.g.*, left, right, above, below) including Euclidean and axial distances, with an exemplar provided for better instruction comprehension by the VLM. For temporal relationships, the dataset examines changes in these spatial relationships over time (*i.e.*, between two timesteps). This involves two observations—the initial in image form and the subsequent described in text—focusing on queries about movements (*e.g.*, getting closer or further away), with responses detailing changes in Euclidean and axial distances.

By introducing these auxiliary datasets, we explicitly drive the VLM to strengthen the ability of spatial and temporal understanding, which benefits robot learning. Note that all these datasets can be automatically generated from *existing trajectories*, without introducing external task/action supervision. In Sec. 6, we empirically explore the best practices to apply these auxiliary datasets.

# 6 EXPERIMENTS

We conduct experiments in both simulated environments and the real world. For the simulated tasks, we tune a VLM on diverse tasks as a generalist. Then, we conduct real-world robot experiments using three protocols: *zero-shot generalization*, *finetuning*, and *joint training*.

## 6.1 SIMULATION EXPERIMENTS

**Settings.** We employ VIMA-Bench (Jiang et al., 2023), a simulated table-top robot manipulation environment to evaluate VLMs trained by our instruction tuning dataset. The environment contains 17 tasks and each task is associated with a multi-modal instruction, including text instructions and reference images that refer to objects of interest or a particular scene arrangement. The robot action space is two 2D coordinates for pick and place positions and two quaternions for rotations. We uniformly subsample the VIMA dataset (Jiang et al., 2023) to form three subsets with different sizes: VIMA-0.8k, VIMA-8k, and VIMA-80k where the number indicates the number of expert trajectories in the dataset.

**Methods.** We compare variants of our method with baselines that follow the recipe of RT-2 (Brohan et al., 2023a), *RT-2 Style*, and *$\mathcal{D}$-RT-2 Style*. As introduced in Sec. 4, *inBC* is the dataset converted from the expert trajectories. Prefix $\mathcal{D}$ means the additional reference images in the task description are described by object detection results in text and the object detection will be performed on the reference images first during inference. Suffix *Aux* means the auxiliary datasets introduced in Sec. 5 are included in the training set, and the letter after it (*e.g.*, *(D)*) indicates the detailed configuration of the auxiliary dataset (see Tab. 2). *Oracle* means that the groundtruth bounding box of objects is used as the object detection results only in the reference images. *RT-2 Style* is similar to *inBC* but the output of the VLM (i.e., robot actions) is a set of special tokens that can be directly mapped to quantized actions. Specifically, in VIMA, the action space is two positions in *robot* coordinates and one rotation. *$\mathcal{D}$-RT-2 Style (I)* is similar to *$\mathcal{D}$-RT-2 Style* but the output tokens present quantized two positions in *image* coordinates and one rotation angle, which is the quantized version of what LLaRA generates in natural language, presented by the same set of special tokens.

We train all methods on these three datasets and evaluate them with 3 levels of difficulties following the test protocol (L1 to L3). Due to missing data in the original VIMA dataset, we were not able to evaluate our approach on L4 properly. For each task, we evaluate all the methods with 20 random seeds and report the average success rates of each level and the average of three levels.

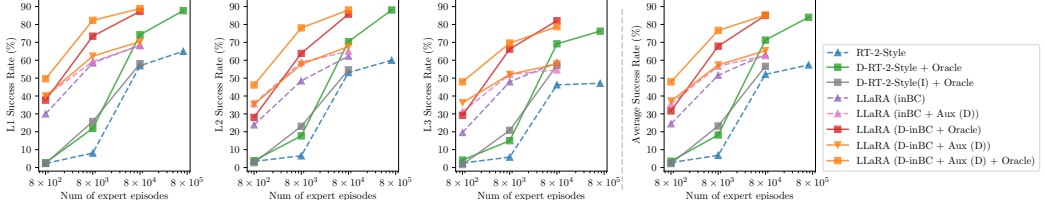

Figure 5: Success rates of the models trained on VIMA subsets. The log-scale x-axis shows the number of expert episodes used in the training set. See Appendix E for the underlying numerical data.

**Effectiveness of LLaRA framework.** Fig. 5 illustrates the performance of selected methods across various dataset scales. All methods generally benefit from increased training data. Key observations include:

- *inBC* consistently surpasses the *RT-2 Style* baseline, and similarly, *$\mathcal{D}$-inBC* outperforms *$\mathcal{D}$-RT-2 Style*. This supports the effectiveness of our approach, which utilizes instruction-tuning style data aligned to image pixel coordinates. It is easier to benefit from the existing VLMs pretrained in a conversation style. Methods based on *RT-2 Style* improve when more robot supervision data is available; however, they significantly underperform compared to our methods when data is limited. Even with 100% data, *RT-2 Style* still underperforms our method trained on only 12% data (Appendix B.2).

- *$\mathcal{D}$-inBC* generally excels over *inBC* owing to the additional information from reference images. This trend is similarly observed between *$\mathcal{D}$-RT-2 Style* and *RT-2 Style*.

- Auxiliary datasets prove to be highly beneficial, particularly when the size of the training set is limited. However, in scenarios like VIMA-80k where the dataset is substantial, sometimes *$\mathcal{D}$-inBC* performs better and the gain from the auxiliary datasets is marginal.

We hypothesize that when training data is adequate, the benefits of the auxiliary dataset diminish, and these data may even distract the model from effectively performing the behavior cloning tasks. It is advisable to carefully regulate the amount of auxiliary data when there is an abundance of expert

episodes to avoid overwhelming the primary learning objectives. By default, we randomly sample from each auxiliary dataset so that the size of each auxiliary dataset is identical to *inBC* or $\mathcal{D}$-*inBC*.

**Comparison to VIMA.** VIMA takes both front and top view images from the environment while ours only takes the front view. We test the most capable model released by VIMA, which is trained on 660k expert tra-

Table 1: Comparison to VIMA (Jiang et al., 2023). Our best model not only achieves better performance but also requires less input and is trained on only 12% of the expert trajectories used in VIMA.

| Method | Config | Data | L1 (%) | L2 (%) | L3 (%) |
|---|---|---|---|---|---|
| VIMA | VIMA-200M + Oracle | 100% | 80.7 | 81.9 | 77.9 |
| LLaRA (Ours) | $\mathcal{D}$-*inBC* + Aux (B) + Oracle | 12% | **90.0** | **88.1** | **79.2** |

jectories, and the results are shown in Tab. 1. Compared to VIMA, our best model not only achieves better performance but also requires less input and is trained on only 12% of the expert trajectories used in VIMA. The full comparison is in Tab. 13.

**Effectiveness of auxiliary datasets.** Tab. 2 shows the different combinations of the auxiliary datasets we studied. For each setting, we randomly sample the same amount of examples from each auxiliary dataset and combine them with the converted behavior cloning dataset. The '*' after a '✓' means the reference images that appeared only in the task description are not used to generate this dataset, which makes the dataset less diverse. We train the model on both *inBC* and $\mathcal{D}$-*inBC* with different auxiliary dataset settings. On VIMA-0.8k, we control the total number of samples from the auxiliary dataset relative to the samples from the converted BC datasets and train all models for 2 epochs. Results in Fig. 6 show that in general, on VIMA-0.8k, the model performs better with more auxiliary data. Tab. 3 shows that on VIMA-8k, auxiliary datasets can significantly benefit *RT-2 Style* models as well.

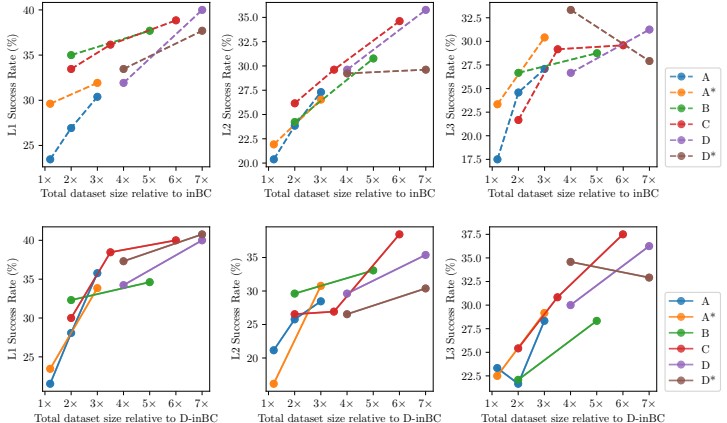

Table 2: Naming of different auxiliary dataset configurations. We always randomly sample the same amount of examples from each dataset. **Det.**: detection; **Loc.**: localization; **Act.**: action prediction; **Fut.**: future prediction; **Spa.**: spatial relationship; **Temp.**: temporal relationship.

| | Loc. | Det. | Act. | Fut. | Spa. | Temp. |
|---|---|---|---|---|---|---|
| A | ✓ | ✓ | | | | |
| A* | ✓* | ✓* | | | | |
| B | ✓ | ✓ | ✓ | ✓ | | |
| C | ✓ | ✓ | ✓ | ✓ | ✓ | |
| D | ✓ | ✓ | ✓ | ✓ | ✓ | ✓ |
| D* | ✓* | ✓* | ✓ | ✓ | ✓* | ✓ |

Table 3: *RT-2 Style* benefits from auxiliary datasets, all models are trained on VIMA-8k.

| Method | L1 (%) | L2 (%) | L3 (%) |
|---|---|---|---|
| *RT-2 Style* | 3.8 | 3.1 | 1.7 |
| *RT-2 Style* + Aux (D) | 36.9 | 35.8 | 32.1 |
| *inBC* | 57.3 | 46.2 | 42.9 |
| *inBC* + Aux (D) | 59.2 | 58.8 | 52.1 |

Figure 6: *inBC* (top) and $\mathcal{D}$-*inBC* (bottom) with different auxiliary dataset settings. Each model is trained on VIMA-0.8k for 2 epochs. In general, the model performs better with more auxiliary data. See Appendix E for the underlying numerical data.

A comprehensive ablation of other design choices is available in Appendix B.

## 6.2 REAL-WORLD ROBOT EXPERIMENTS

We further conduct zero-shot generalization, finetuning, and joint training experiments in a novel real-world environment, in which there is a robot arm with a gripper and a fixed RGB camera positioned above the arm to collect observations (see Fig. 7). The action space is the same as the one in VIMA. As all the objects are placed on a plain surface and the camera is fixed, similar to VIMA, a linear mapping between the image coordinates and the robot action space can be estimated by calibration. For the methods that rely on object detection *i.e.*, $\mathcal{D}$-*inBC*, we employed one-shot detection using OWLv2 (Minderer et al., 2024). We benchmark each policy on three tasks:

- **T1**: "Move the {object} into the large bowl."
- **T2**: "Rotate the {object} by {angle} degrees."
- **T3**: "Move the {object} on top of the tray."

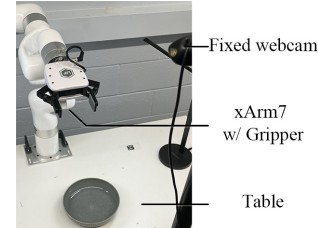

Fixed webcam

xArm7 w/ Gripper

Table

Figure 7: Our real-world robot setting.

where the {object} is chosen from a set of 10 plastic toys and the {angle} in T2 is chosen from the following set: {30, 90, 120, 180}. In all tasks, all objects are placed randomly on the table before running an episode and they never appear in the training set. A success in T1 and T3 is if the object is placed more than 50% inside the bowl or on the tray. A success in T2 is if the object appears to be clockwise rotated by the angle by visual inspection. We run 20 randomly initialized episodes for each setting and report the average success rate. A visual reference for typical task start states and success ends are provided in Fig. 8.

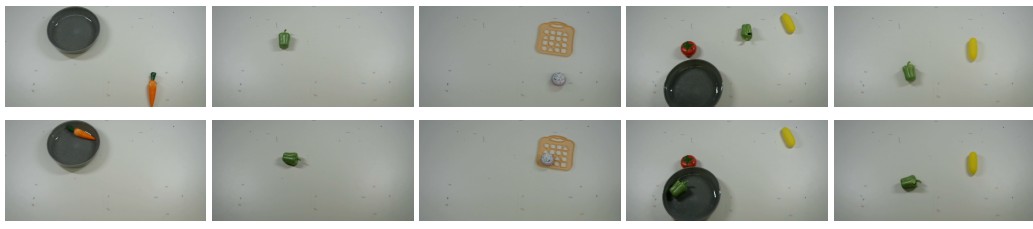

|  |  |  |  |  |
|:--:|:--:|:--:|:--:|:--:|
| T1 | T2 | T3 | T1 w/ Distractors | T2 w/ Distractor |

Figure 8: Visual reference for the initial image (top row) and successful end positions (bottom row) in three real-world tasks. We also show some examples that have distracting objects in the scene.

Table 4: Real-world robot experiment results. All models are trained on VIMA-8k unless otherwise stated. 'Aux' means trained on auxiliary datasets and 'OD' means using an object detector (OWLv2-base-16) to describe the observation before the VLM query. In **Protocol**, 'ZS' stands for zero-shot evaluation, 'FT' means the model is first trained on VIMA-8k for 2 epochs and then finetuned on **Add. Data** for 1 epoch. 'JT' means the model is trained on the combination of VIMA-8k and **Add. Data** for 2 epochs.

| Protocol | Method | | Add. Data | T1 (%) | T2 (%) | T3 (%) | Avg. (%) |
|:--|:--|:--|:--|:--:|:--:|:--:|:--:|
| ZS | GPT-4o | ver. 2024-05-13 | | 20 | 45 | 30 | 31.6 |
| | *RT-2 Style* | VIMA-8k | | 0 | 0 | 0 | 0 |
| | *RT-2 Style* | VIMA-660k | | 0 | 0 | 0 | 0 |
| | LLaRA (Ours) | *inBC* | | 40 | 50 | 20 | 36.6 |
| | | *inBC* + Aux (D) | | 10 | 30 | 10 | 16.6 |
| | | *D-inBC* + OD | | 30 | 40 | 25 | 31.6 |
| | | *D-inBC* + Aux (C) + OD | | 35 | 40 | 5 | 26.6 |
| FT | *RT-2 Style* | VIMA-8k | xArm-Det | 0 | 0 | 0 | 0 |
| | *RT-2 Style* | VIMA-660k | xArm-Det | 0 | 0 | 0 | 0 |
| | LLaRA (Ours) | *inBC* | xArm-Det | 0 | 0 | 0 | 0 |
| | | *inBC* | xArm-Action | 30 | 45 | 5 | 26.6 |
| | | *inBC* + Aux (D) | xArm-Det | **100** | 95 | 70 | 88.3 |
| | | *inBC* + Aux (D) | xArm-Action | 70 | 75 | 80 | 75 |
| | | *D-inBC* + OD | xArm-Det | 0 | 0 | 0 | 0 |
| | | *D-inBC* + OD | xArm-Action | 45 | 80 | 55 | 60 |
| | | *D-inBC* + Aux (C) + OD | xArm-Det | 70 | 90 | 70 | 76.6 |
| | | *D-inBC* + Aux (C) + OD | xArm-Action | 90 | **100** | **85** | **91.6** |
| JT | LLaRA (Ours) | *inBC* | xArm-Det | 80 | 75 | 70 | 75 |
| | | *inBC* | xArm-Action | 60 | 75 | 55 | 63.3 |
| | | *inBC* + Aux (D) | xArm-Det | 75 | 85 | 70 | 76.6 |
| | | *inBC* + Aux (D) | xArm-Action | 50 | 90 | 50 | 63.3 |
| | | *D-inBC* + OD | xArm-Det | 65 | 95 | 45 | 68.3 |
| | | *D-inBC* + OD | xArm-Action | 65 | 55 | 25 | 48.3 |
| | | *D-inBC* + Aux (C) + OD | xArm-Det | 70 | 95 | 85 | 83.3 |
| | | *D-inBC* + Aux (C) + OD | xArm-Action | 45 | 70 | 20 | 53.3 |

**Zero-shot generalization.** For zero-shot generalization and finetuning, we use the pretrained models from Sec. 6.1 only trained on simulated VIMA data. We also make the setting very challenging by selecting the pretrained model only using 1.2% of the VIMA training data (VIMA-8k). We benchmark the models without any further training, along with a new baseline, GPT-4o (OpenAI, 2023). The results are presented in the upper part of Tab. 4. In this setting, *inBC* achieves the best overall performance. We hypothesize that the auxiliary datasets used in *inBC* + Aux and *D-inBC* + Aux drive the model to focus more on the VIMA domain, leading to overfitting. Meanwhile, GPT-4o achieves commendable performance, largely attributable to its extensive dataset and substantial parameterization.

**Finetuning on real-world data.** In the finetuning setting, we first collect two in-domain real-world datasets xArm-Det and xArm-Action (see Appendix C). Then the models trained on VIMA-8k introduced in Sec. 6.1 are further tuned on the new real-world datasets for 1 epoch and the evaluation results are presented in the middle of Tab. 4. In general, finetuning outperforms other settings because it allows the model to focus more on real-world data distribution, rather than be distracted by simulated VIMA data. Auxiliary data contributes to a large boost in performance during finetuning, showing that explicitly uncovering information from the existing in-domain data can benefit the model. It seems *inBC* is not trained to take advantage of such new in-domain robotics data, performing worse than zero-shot. Similar to our findings in simulation experiments, LLaRA in general benefits from pretrained VLM since LLaRA is trained on a dataset that has been organized in a conversation style that is similar to what is used to pretrain the VLM. In contrast, *RT-2 Style* requires a large amount of data so it suffers more from limited data in our real-world experiments.

**Joint training on both simulated and real-world data.** In the joint training setting, we combine both VIMA-8k with xArm-Det or xArm-Action and tune a VLM jointly on both simulated and real-world datasets. The full results are presented at the lower part of Tab. 4. In the joint training setting, xArm-Det is equally or more beneficial than xArm-Action, highlighting the importance of detection and localization information from the real-world setting. Auxiliary data proves most beneficial when jointly trained with xArm-Det, achieving the highest performance across all joint training settings. This observation suggests that careful exploration of auxiliary dataset usage is crucial to maximize their benefits when joint training is performed.

**More generalization results.** We further validate the generalization ability of LLaRA on tasks with distractors. There are three test settings and none of them is included in the simulated training set or the dataset for finetuning: **T1 w/ distractor**: where we randomly put another random toy on the table as a distractor;

Table 5: More generalization results of $\mathcal{D}$-*inBC* + Aux (C). The model is tested on three unseen tasks using two protocols and it shows strong performance with minimal finetuning on in-domain vision data.

| Protocol | ZS | FT on xArm-Action |
|---|---|---|
| **T1 w/ distractor (%)** | 10 | 75 |
| **T1 w/ complex rephrasing (%)** | 0 | 40 |
| **T4 (%)** | 0 | 20 |

**T1 w/ complex rephrasing**: where we rephrase the task description in 10 different unseen ways using GPT-4; **Unseen task T4**: the task is "Weight the object using the scale then place it back on the table." which is never included in VIMA or xArm-Action. The results are presented in Tab. 5. Our model demonstrates robust performance on generalized tasks, achieving a 75% success rate in T1 with distractors. This suggests that our VLM retains its generalization ability even after being trained on our instruction dataset. With a small amount of in-domain vision data, it can achieve strong performance on previously unseen tasks.

**Summary.** Our findings highlight the strong generalization ability of LLaRA. When pretrained on simulated VIMA data, LLaRA requires significantly less effort and data to adapt to real-world tasks. By leveraging image coordinates, LLaRA establishes a clear, straightforward mapping between visual inputs and robot actions. In contrast, *RT-2 Style* struggles under limited data conditions, largely because of the extensive data needed to learn the semantics of newly introduced tokens.

We discuss all implementation details and qualitative results in Appendix C.

## 7 CONCLUSION

We present LLaRA, a framework that turns a pretrained vision language model (VLM) into a robot policy using curated instruction tuning datasets. Firstly, we create a conversation-style instruction-tuning data using existing behavior cloning data and align the robot action to image pixel coordinates. Instruction tuning the VLM on such data confirms the effectiveness of our framework, particularly when the training data is limited. Then we construct auxiliary datasets using the same robot trajectory data through self-supervision, without any additional action labels. Experiments across several synthetic environments and robot manipulation tasks in the real world validate the effectiveness of our proposed LLaRA framework. We believe the new observations on the VLM to Vision-Language-Action Model (VLA) representation transfer we share with this paper are valuable not only to the robot learning community but also to the computer vision community.

ACKNOWLEDGMENTS

This work was supported by Electronics and Telecommunications Research Institute (ETRI) grant funded by the Korean government foundation [24ZB1200, Research of Human-centered autonomous intelligence system original technology]. The authors would like to thank Bohong Chen for verifying our code implementation and reproducing experiment results, as well as Xiaodong Liu and Hanyi Yu for their valuable input.

REPRODUCIBILITY STATEMENT

The code, datasets, and pretrained models are available at https://github.com/LostXine/LLaRA.

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

## APPENDIX

The appendix covers all the implementation details, expanded experiment results in both simulation and real-world, further discussions, raw results, and all the prompt templates we used in the paper.

## A   IMPLEMENTATION DETAILS

In this section, we provide more implementation details regarding dataset preparation, model training, inference, and RT-2 style baselines.

### A.1   DATASET PREPARATION

In this subsection, we give more details and examples of the datasets used in this paper.

#### A.1.1   BUILD *inBC* DATASET FROM EXPERT TRAJECTORIES

We formulate the dataset in a *single-image single-turn conversation* setting to emulate a policy where the user queries the Vision Language Model (VLM) with the current observation, and the VLM generates a response that can be directly translated into a concrete action, as introduced in Sec. 4. The image and text from the current observation can be directly processed by the VLM using a fixed template, while the conversion of numerical actions into text is facilitated through the use of normalized 2D image coordinates. Tab. 8 contains an example using a VIMA-Bench sample.

Given the current limitations of LLaVA (Liu et al., 2023a), to optimize performance, we propose two techniques:

- **Action history in query.** Each query to the VLM includes a history of actions previously executed in the episode. This approach enhances the VLM's understanding of task context within the constraints of a single-turn conversation. (See the orange text in Tab. 8)

- **Plan 'twice' act once - multi-step planning.** In action generation, the dataset is prepared such that the VLM is designed to generate *all* successive actions in the response, literally performing multi-step action planning. However, only the first action is executed. For the next state, we query the VLM again with a new observation and a question and only take the first action generated by the VLM.

These designs proved beneficial in our ablation study, as detailed in Tab. 14.

The rest of this section details the process on VIMA-Bench (Jiang et al., 2023). VIMA-Bench introduces 17 simulated robot manipulation tasks using multimodal prompts that combine text and reference images. The full accompanying dataset includes 660k expert trajectories, covering 13 of the 17 tasks. All tasks occur in an environment where a robot arm, equipped with either a spatula or a suction cup, is positioned alongside a flat table. Multiple objects with diverse textures are randomly initialized on the table according to the specific task setting.

Each episode in the dataset features a multimodal task description that clarifies the episode's goal, incorporating images referred to as 'reference images'. Two third-person view cameras capture the scene, providing a top-down view and a front view looking down at the table. The dataset includes RGB, instance segmentation masks and meta information (*e.g.*, textures and shapes) of the objects in images for the reference images and the observations captured by these cameras. We first extract the bounding boxes of each object from the instance-level segmentation mask, as an object detection oracle. Additionally, the dataset contains the expert action sequence required to complete the episode. The action space consists of two poses: for the robot equipped with a spatula, these poses indicate the start and end points of a push; for the robot with a suction cup, they specify where the robot picks up and places an object.

Due to the constraints of LLaVA (Liu et al., 2023a) and our findings presented in Tab. 15, in each conversation, only the current image from the front view camera is retained as the visual input to the VLM. Other reference images are substituted with short captions that describe the texture (color) and shape of the referred object, which are details extracted from the meta information (See *inBC* in Tab. 8). However, in principle, there is nothing stopping us from extending the current framework to a multi-image setting, taking advantage of a VLM that can handle multiple interleaving images in a conversation.

### A.1.2 BUILD $\mathcal{D}$-*inBC* FROM *inBC*

While *inBC* in Tab. 8 has shown its great power in many aspects when the reference image contains a scene that has multiple objects instead of one, the *inBC* will fail to deliver any useful information (*e.g.*, Tab. 9). To address this issue, we utilize an object detector to parse an image of a scene into a list of objects with its corresponding bounding box. The new dataset is named $\mathcal{D}$-*inBC* because a reference image will be 'described' as a list of objects now. Tab. 9 shows an example where $\mathcal{D}$-*inBC* delivers more critical information than *inBC*.

### A.1.3 $\mathcal{D}$-*inBC (L)*: DESCRIBING THE REFERENCE IMAGES IN NATURAL LANGUAGE

Following the suggestion from the reviewer, we trained a variant of $\mathcal{D}$-*inBC*, which completely removes all the coordinate numbers in the prompt even from the reference images. Instead, we use a pretrained VLM, Qwen2-VL-7B-Instruct (Wang et al., 2024), to generate natural language descriptions of all reference images, omitting any use of image coordinates. These textual descriptions replaced the original lists of image coordinates in the input prompt for the $\mathcal{D}$-*inBC* setting, resulting in a variant we refer to as $\mathcal{D}$-*inBC (L)*, where *(L)* stands for language. Then we train on VIMA-8k and the conversations describing the reference image. During inference, LLaRA first generates textual descriptions of all reference images using its own capabilities and subsequently leverages the generated text to perform the task.

The following prompt is used to generate image description: "`Write short descriptions of each object in the image, specifying their color, texture, shape, and relative positioning.`"

Tab. 10 presents an example of generated image description of the {frame_0} in Tab. 9. We present the performance of $\mathcal{D}$-*inBC (L)* at Tab. 6 (VIMA) and Tab. 7 (Real-world).

In general, $\mathcal{D}$-*inBC (L)* achieves comparable but slightly worse performance than $\mathcal{D}$-*inBC*, which uses structured text to describe the reference images.

Table 6: $\mathcal{D}$-*inBC (L)* performance when trained on VIMA-8k

| Method | L1 (%) | L2 (%) | L3 (%) |
|---|---|---|---|
| *inBC* + Aux (D) | 59.2 | **58.8** | 52.1 |
| $\mathcal{D}$-*inBC* + Aux (D) | **62.3** | 57.7 | 52.1 |
| $\mathcal{D}$-*inBC (L)* + Aux (D) | 61.9 | 54.2 | **55.4** |

Table 7: $\mathcal{D}$-*inBC (L)* real-world experiment results.

| Method | Add. Data | T1 (%) | T2 (%) | T3 (%) | Avg. (%) |
|---|---|---|---|---|---|
| *inBC* + Aux (D) | xArm-Det | 100 | 95 | 70 | 88.3 |
| *inBC* + Aux (D) | xArm-Action | 70 | 75 | 80 | 75 |
| $\mathcal{D}$-*inBC (L)* + Aux (D) | xArm-Det | 65 | 90 | 70 | 75 |
| $\mathcal{D}$-*inBC (L)* + Aux (D) | xArm-Action | 75 | 90 | 75 | 80 |

### A.1.4 AUXILIARY DATASETS

The auxiliary datasets are created using the template outlined in Fig. 4. During dataset generation, for each sample, one template is randomly selected from a pool of 15 templates. These templates were initially rephrased from a single sentence using GPT-4 (OpenAI, 2023). The full list of the pools are listed in Tab. 30, Tab. 31, Tab. 29, Tab. 32, Tab. 33, Tab. 34, Tab. 35, Tab. 36, Tab. 37, Tab. 38, and Tab. 39. The qualitative examples are available in Tab. 11 and Tab. 12.

### A.2 TRAINING

We initiate training using a pretrained LLaVA-1.5-7B (Liu et al., 2024b) model and finetune all parameters, including the language model and the projection layer, with the exception of the vision encoder. The training settings closely align with those of the original LLaVA stage 2. Specifically, we utilize a single-cycle cosine annealing scheduling with $0.03$ warm-up ratio and a maximum learning rate of $2 \times 10^{-5}$. However, for VIMA-0.8k and VIMA-8k, we employ a batch size of 32, whereas for VIMA-80k, we restore the batch size to 128.

### A.3 INFERENCE

During inference, for the models trained on *inBC*, a new conversation is initiated at each step. The model is queried with a single current image observation from the front-view camera and an instruction that has been prepared in the same format as the dataset. Additionally, a prompt randomly selected from Tab. 28 is added to the instruction during inference only, although it appears to have limited impact in retrospect (see Appendix B.7).

Model sampling is disabled during training, ensuring that the model consistently outputs the token with the highest probability. This approach is designed to maximize the consistency and reliability of the model's responses.

For the models trained on $\mathcal{D}$-*inBC*, before we query the VLM, we would first run object detection on all reference images and take the detection results to fill the instruction template of $\mathcal{D}$-*inBC*. Besides this, the settings are identical to the models introduced ahead.

In this paper, we explore three approaches to object detection. The first method involves using a single query from Tab. 30 and Tab. 31 and employing the same VLM as the policy to perform object detection.

The second approach utilizes a Mask-RCNN (He et al., 2017) model, which features a ResNet50 (He et al., 2016) backbone pretrained on the COCO dataset (Lin et al., 2014) and subsequently finetuned using the VIMA dataset. A suffix 'OD' will be added to the model name if the model uses this method.

Finally, we test the versions that use the groundtruth detection results. A suffix 'Oracle' will be added to the model name if the model uses this information.

Table 8: Comparison between our converted instruction tuning datasets *inBC* and $\mathcal{D}$-*inBC* for a VIMA-Bench sample. The task description of the episode is in green. The description of an object in the reference image (oracle detection results) is in cerulean. The action is in orange and the action history is in magenta.

| | |
|---|---|
| **{dragged_obj}** | |
| **{base_obj}** | |
| **Task** | First put **{dragged_obj}** into **{base_obj}** then put the object that was previously at its south into the same **{base_obj}**. |
| *inBC* **Input** | (current front camera view.)
<task>First put <p>rainbow letter T</p> into <p>wooden bowl</p> then put the object that was previously at its south into the same <p>wooden bowl</p>.</task>
Every action you take must include two locations in the format of (x, y) and one clockwise rotation angle in the format of <r>[r]</r>. The first location is the image coordinate where you use a suction cup to pick up the object, and the second location is where you place the object. The image coordinate ranges from 0 to 1. The rotation angle indicates how many degrees you rotate the object clockwise, and it ranges from -359 to 359.
You have finished: Step 1: Pick up the object at (0.480, 0.367), rotate <r>[0]</r> degrees, and drop it at (0.727, 0.547). |
| *inBC* **Output** | Step 2: Pick up the <p>rainbow letter V</p> at (0.500, 0.617), rotate <r>[0]</r> degrees, and drop it at (0.746, 0.617). |
| $\mathcal{D}$-*inBC* **Input** | (current front camera view.)
<task>First put <p>rainbow letter T</p> at (0.500, 0.594), {0.102, 0.188} into <p>wooden bowl</p> at (0.457, 0.531), {0.195, 0.328} then put the object that was previously at its south into the same <p>wooden bowl</p> at (0.457, 0.531), {0.195, 0.328}</task>
Every action you take must include two locations in the format of (x, y) and one clockwise rotation angle in the format of <r>[r]</r>. The first location is the image coordinate where you use a suction cup to pick up the object, and the second location is where you place the object. The image coordinate ranges from 0 to 1. The rotation angle indicates how many degrees you rotate the object clockwise, and it ranges from -359 to 359.
You have finished: Step 1: Pick up the object at (0.480, 0.367), rotate <r>[0]</r> degrees, and drop it at (0.727, 0.547). |
| $\mathcal{D}$-*inBC* **Output** | [Same as *inBC* ] |

Table 9: Another comparison between two converted instruction tuning datasets. In this example, the reference images in the task description depict reference images with multiple objects. The task description of the episode is in green. The description of an object in the reference image (oracle detection results) is in cerulean and the action is in orange.

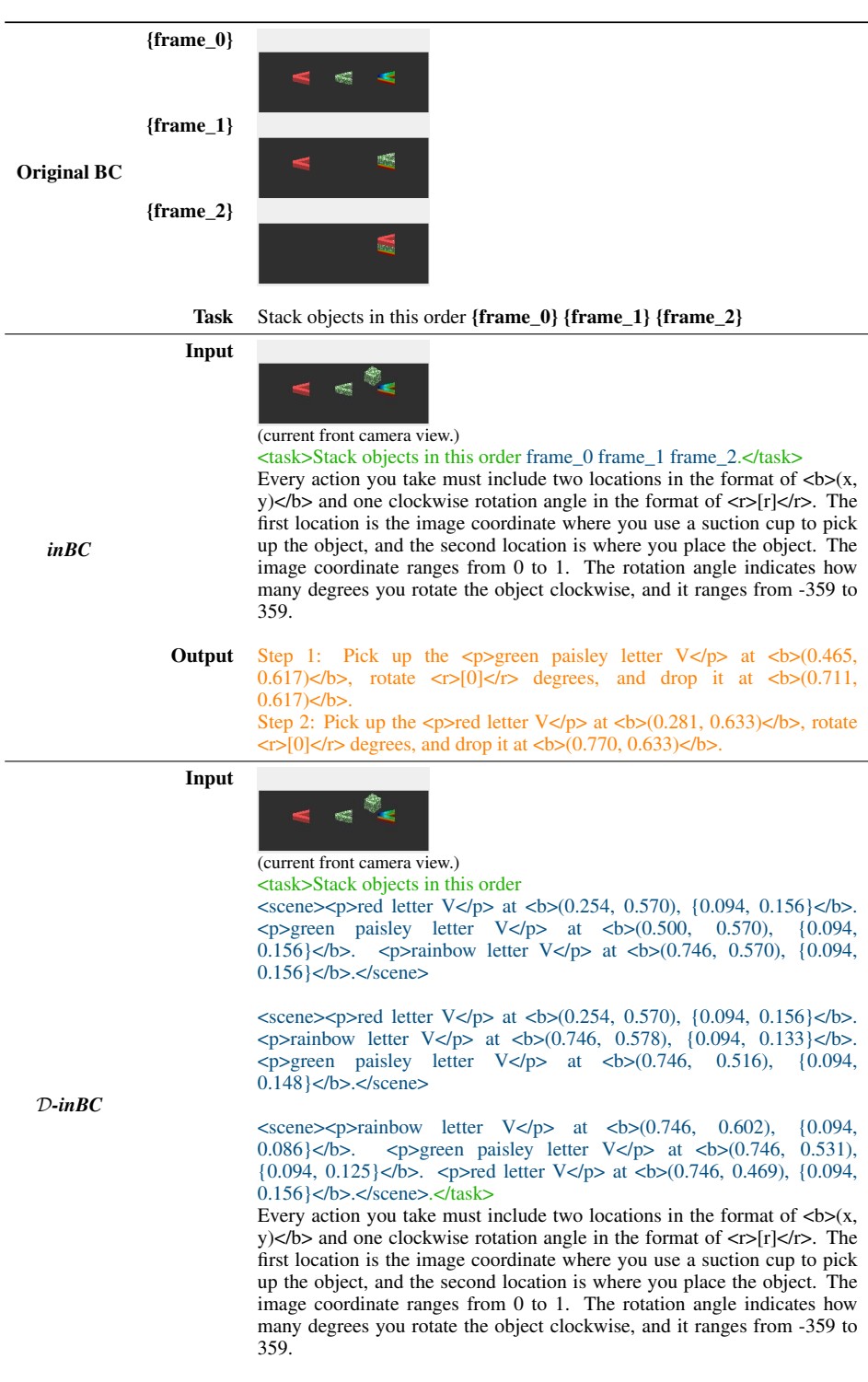

Table 10: Image description used in $\mathcal{D}$-*inBC (L)*, an example of {frame_0} in Tab. 9.

The image displays three distinct objects, each with unique colors, shapes, textures, and spatial relationships.
1. Object on the Left:
- Color: Red
- Texture: Matte
- Shape: Triangular
- Spatial Relationship: Positioned to the far left of the image.
2. Object in the Middle:
- Color: Green and white
- Texture: Speckled
- Shape: Arrowhead
- Spatial Relationship: Situated to the right of the first object.
3. Object on the Right:
- Color: Multicolored (red, blue, green, and yellow)
- Texture: Glossy
- Shape: Trapezoidal
- Spatial Relationship: Located to the far right of the image.
These objects are arranged in a linear sequence, with each subsequent object placed to the right of the previous one, creating a horizontal alignment. The colors and textures vary, adding visual interest to the composition.

## A.4 RT-2-Style baselines

For the RT-2-Style baseline, we follow the original RT-2's design (Brohan et al., 2023a) and adapt special tokens to present the numerical action in VIMA action space. The original action space of VIMA contains two points for pick and place. Each point contains a 2D coordinate in *robot coordinates* for position and a 3D orientation of the gripper. Since we only have yaw rotation in the dataset, we simplify two 3D orientations into one yaw angle. So the action space is now 5 Degrees of Freedom (DoF): 4 numerical values present the 2D pick and place locations in *robot coordinates*, and 1 value presents yaw rotation. In our implementation, the procedure begins by normalizing each element of the action to a range from 0 to 1. We then quantize these values into 256 bins and map the resulting discrete values to tokens indexed from 31000 to 31255 in the tokenizer. The dataset employed is similar to *inBC*, but it omits the prompt for the output format, and the action is represented by the mapped special tokens. Additionally, we have prepared a version similar to $\mathcal{D}$-*inBC*, named $\mathcal{D}$-*RT-2 Style*, which takes advantage of object detection on reference images. $\mathcal{D}$-*RT-2 Style (I)* is similar to $\mathcal{D}$-*RT-2 Style*. However, four special tokens present pick and place positions in two normalized 2D *image coordinates* for each action, and one token presents the rotation, instead of an action in VIMABench action space. It is the same action presentation generated by our LLaRA but our method responds in natural language while $\mathcal{D}$-*RT-2 Style (I)* responds in special tokens.

## B Expanded simulation experiments

In this section, we include more experiment results. All the numerical results reported in this paper, especially in the figures, are collected in Tab. 21 and Tab. 22 (VIMA-0.8k), Tab. 23 and Tab. 24 (VIMA-8k), Tab. 25 and Tab. 26 (VIMA-80k), and Tab. 27 (full VIMA dataset).

## B.1 Clarification on L4 results

We adopted the testing protocol from VIMA-Bench (Jiang et al., 2023), conducting evaluations across four levels of difficulty, from L1 to L4. However, during our analysis, we discovered an inconsistency in the training set: the rotation information for the robot end effector was recorded as zero for all sweeping tasks where the end effector is a spatula. Given the importance of the spatula orientation in a sweeping task and the fact that sweeping tasks constitute 25% of the evaluations at the L4 difficulty level, we concluded that our ability to accurately evaluate our method at L4 was compromised. Considering that the original VIMA model released by the authors appears to include this rotation information, we have chosen not to report the results for L4 in our study.

Table 11: Qualitative examples of the auxiliary datasets. In this table, the examples of all the datasets take the same input image as shown in the first row. The description of an object or a scene (oracle detection results) is in blue. The action is in orange.

| Image Input | | 
(current front camera view.) |
|---|---|---|
| **Object Localization** | Input | Could you tell me the location of \<p>red letter V\</p> in the image? Use the format \(x, y), {w, h}\, where x and y denote the center of the bounding box and w and h are the width and height. Coordinates start at the top left corner and should be normalized between 0 and 1. |
| | Output | \<p>red letter V\</p> at \(0.254, 0.570), {0.094, 0.156}\. |
| **Object Detection** | Input | Enumerate and describe every object found in the image. For each object, utilize the format \(x, y), {w, h}\, where x, y are the bounding box center coordinates and w, h are the dimensions (width and height) of the bounding box. The coordinates begin at the top left corner and are normalized between 0 and 1. |
| | Output | \<scene>
\<p>red letter V\</p> at \(0.254, 0.570), {0.094, 0.156}\.
\<p>green paisley letter V\</p> at \(0.500, 0.570), {0.094, 0.156}\.
\<p>rainbow letter V\</p> at \(0.746, 0.570), {0.094, 0.156}\.
\<p>green paisley block\</p> at \(0.668, 0.414), {0.094, 0.227}\.
\</scene> |
| **Action Prediction** | Input | Can you explain what needs to be done to adjust the scene shown in the image to resemble the second scene? The second scene \<scene>\<p>red letter V\</p> at \(0.254, 0.594), 0.094, 0.156\. \<p>green paisley letter V\</p> at \(0.742, 0.547), 0.098, 0.156\. \<p>rainbow letter V\</p> at \(0.746, 0.609), 0.094, 0.125\. \<p>green paisley block\</p> at \(0.668, 0.414), 0.094, 0.227\.\</scene> consists of object bounding boxes provided in the format \(x, y), {w, h}\. Here, x and y represent the center coordinates, and w and h are the width and height. The coordinates should start from the top left corner and be normalized to a scale of 0 to 1. Every action you take must include two locations in the format of \(x, y)\ and one clockwise rotation angle in the format of \<r>[r]\</r>. The first location is the image coordinate where you use a suction cup to pick up the object, and the second location is where you place the object. The image coordinate ranges from 0 to 1. The rotation angle indicates how many degrees you rotate the object clockwise, and it ranges from -359 to 359. |
| | Output | Pick up the \<p>green paisley letter V\</p> at \(0.465, 0.617)\, rotate \<r>[0]\</r> degrees, and drop it at \(0.711, 0.617)\. |
| **Future Prediction** | Input | An image depicts a scene containing multiple objects. Now you pick up the \<p>green paisley letter V\</p> at \(0.465, 0.617)\, rotate \<r>[0]\</r> degrees, and drop it at \(0.711, 0.617)\, what will the scene look like? Write the list of object bounding boxes. The bounding boxes should be formatted as \(x, y), {w, h}\, where x and y denote the center coordinates, and w and h are the width and height. The coordinates start from the top left corner and are normalized to a scale of 0 to 1. |
| | Output | \<scene>
\<p>red letter V\</p> at \(0.254, 0.594), {0.094, 0.156}\.
\<p>green paisley letter V\</p> at \(0.742, 0.547), {0.098, 0.156}\.
\<p>rainbow letter V\</p> at \(0.746, 0.609), {0.094, 0.125}\.
\<p>green paisley block\</p> at \(0.668, 0.414), {0.094, 0.227}\.
\</scene> |

Table 12: (Continued) Qualitative examples of the auxiliary datasets. In this table, the examples of all the datasets take the same input image as shown in the first row. The description of an object or a scene (oracle detection results) is in blue. The examplar is in purple.

| Image Input | |  |
| --- | --- | --- |
| | | (current front camera view.) |
| **Spatial Relationship** | Input | Can you explain the relative spatial positions of <p>red letter V</p> compared to <p>green paisley letter V</p> in this image? Use terms such as left, right, above, below, etc. Also, determine the 2D center distance and the Euclidean center distance between them. Your output should match this format: <p>green paisley block</p> is left and top from <p>rainbow letter V</p> with 2d center distance (x,y) of <d>(-0.078, -0.156)</d> and euclidean center distance of <e>0.175</e>. The coordinates are image coordinates normalized to a scale of 0 to 1 starting from the top left corner. |
| | Output | <p>red letter V</p> is left and bottom from <p>green paisley letter V</p> with 2d center distance (x,y) of <d>(-0.246, 0.000)</d> and euclidean center distance of <e>0.246</e>. |
| **Temporal Relationship** | Input | At the initial timestamp, the image shows a scene, and the second image described as <scene><p>red letter V</p> at (0.254, 0.594), {0.094, 0.156}. <p>green paisley letter V</p> at (0.742, 0.547), {0.098, 0.156}. <p>rainbow letter V</p> at (0.746, 0.609), {0.094, 0.125}. <p>green paisley block</p> at (0.668, 0.414), {0.094, 0.227}.</scene> depicts the next timestamp. Can you describe the change in the relative position of <p>red letter V</p> compared to <p>green paisley letter V</p> between these two timestamps? Use relative distance terms such as getting closer or moving further away, etc. Also, find the change in the 2D center distance and the Euclidean center distance between the two images. Your output must follow this format: <p>rainbow letter V</p> moves far away from <p>green paisley block</p>. 2d center distance (x,y) of <p>green paisley block</p> from <p>rainbow letter V</p> changes by <d>(0.000, -0.039)</d> and Euclidean center distance between them <e>0.036</e>. The coordinates are image coordinates normalized to a scale of 0 to 1 starting from the top left corner. |
| | Output | <p>green paisley letter V</p> moves far away from <p>red letter V</p>. 2d center distance (x,y) of <p>red letter V</p> from <p>green paisley letter V</p> changes by <d>(-0.242, 0.047)</d> and Euclidean center distance between them <e>0.244</e>. |

## B.2 EXTENDED COMPARISON TO VIMA AND OTHER BASELINES

We would first clarify the difference between our method and VIMA (Jiang et al., 2023) in terms of the inputs. VIMA takes both front and top view images from the environment while ours only takes the front view. We test the most capable model released by VIMA, which is trained on 660k expert trajectories, as well as other RT-2-Style baselines trained on the same whole 660k dataset. The results are listed in Tab. 13. Compared to VIMA (Jiang et al., 2023) and other baselines, our best model not only achieves better performance but also requires less input and is trained on only 12% of the data used in VIMA.

## B.3 WILL BASELINES BENEFIT FROM LONGER TRAINING?

The introduction of auxiliary datasets in the training set naturally increases the number of optimization steps per training epoch. In this subsection, we confirm that the performance improvement of LLaRA with auxiliary datasets is due to the data itself, rather than simply more optimization steps. To validate this, we trained several baselines without auxiliary data for longer epochs on three VIMA subsets. As shown in Fig. 9, Fig. 10, and Fig. 11, extending the training epochs does not enhance performance, indicating that the auxiliary data provides additional information that boosts model performance.

Table 13: Comparison to VIMA (Jiang et al., 2023) and other baselines. Our best model not only achieves better performance but also requires less input and is trained on only 12% of the data used in VIMA.

| Method | Config | Data | L1 (%) | L2 (%) | L3 (%) |
|---|---|---|---|---|---|
| VIMA (Jiang et al., 2023) | VIMA-200M + Oracle | 100% | 80.7 | 81.9 | 77.9 |
| *RT-2 Style* | | 12% | 56.9 | 53.1 | 46.2 |
| | | 100% | 65.0 | 59.6 | 42.5 |
| $\mathcal{D}$-*RT-2 Style* | $\mathcal{D}$-*RT-2 Style* + Oracle | 12% | 74.2 | 70.4 | 69.2 |
| | $\mathcal{D}$-*RT-2 Style* + Oracle | 100% | 86.2 | **88.1** | 75.8 |
| $\mathcal{D}$-*RT-2 Style (I)* | $\mathcal{D}$-*RT-2 Style (I)* + Oracle | 12% | 58.1 | 54.2 | 55.4 |
| | $\mathcal{D}$-*RT-2 Style (I)* + Oracle | 100% | 70.0 | 70.0 | 59.2 |
| LLaRA (Ours) | $\mathcal{D}$-*inBC* + Oracle | 12% | 87.3 | 85.4 | **82.1** |
| | $\mathcal{D}$-*inBC* + Aux (B) + Oracle | 12% | **90.0** | **88.1** | 79.2 |
| | $\mathcal{D}$-*inBC* + Aux (D) + Oracle | 12% | 83.8 | **88.1** | 78.8 |

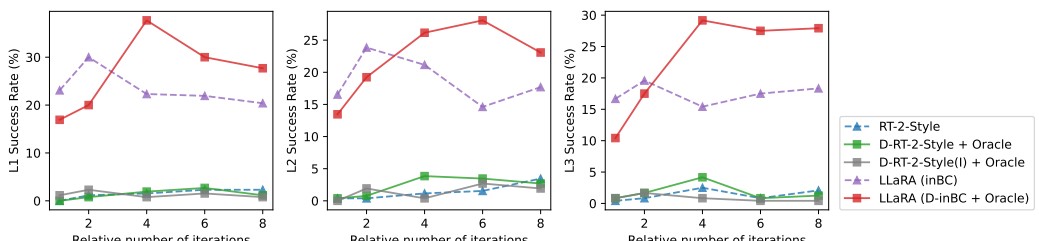

Figure 9: Model performances on VIMA-0.8k for longer epochs.

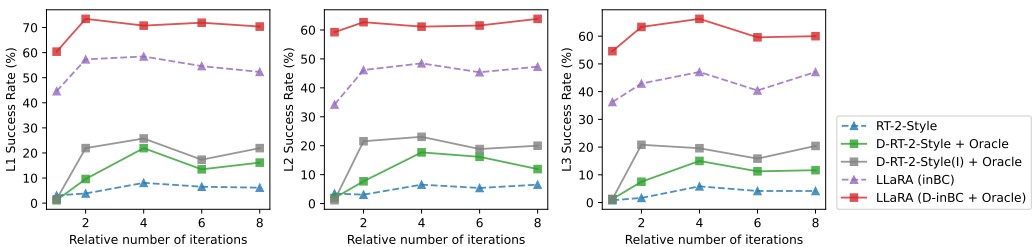

Figure 10: Model performances on VIMA-8k for longer epochs.

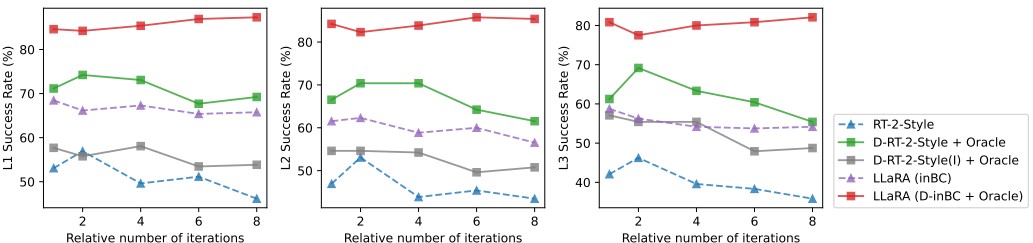

Figure 11: Model performances on VIMA-80k for longer epochs.

## B.4 ABLATION ON ACTION HISTORY AND MULTI-STEP PLANNING

As described in Appendix A.1.1, we enable action history and multi-step planning when generating *inBC* and $\mathcal{D}$-*inBC*. Tab. 14 shows that these designs are helpful.

Table 14: Ablation on action history and multi-step planning. **His.** stands for enabling action history and **Plan** means enabling multi-step planning. All models are trained on VIMA-8k for 2 epochs.

| Method | His. | Plan | L1 (%) | L2 (%) | L3 (%) |
|---|---|---|---|---|---|
| *inBC* | ✓ | ✓ | **57.3** | **46.2** | **42.9** |
| *inBC* | ✓ | ✗ | 43.5 | 36.9 | 36.7 |
| *inBC* | ✗ | ✓ | 45.8 | 39.6 | 35.8 |
| *D-inBC* + OD | ✓ | ✓ | **63.5** | **51.5** | **50.0** |
| *D-inBC* + OD | ✓ | ✗ | 58.5 | 41.5 | 47.1 |
| *D-inBC* + OD | ✗ | ✓ | 53.5 | 39.2 | 37.5 |

## B.5    ABLATION ON MULTIPLE IMAGE INPUTS

This ablation studies multiple image inputs within a single conversation. Each image is processed by the vision encoder and the projection layer to generate a series of tokens. These image tokens are then integrated into the textual conversation at the points where the corresponding images are referenced.

However, because LLaVA is trained with one image per conversation instead of an interleaving style. The performance drops significantly when applying multiple image inputs, listed in Tab. 15.

Table 15: Ablation on multiple image inputs (**Mul.**). All models are trained on VIMA-8k for 2 epochs.

| Method | Mul. | L1 (%) | L2 (%) | L3 (%) |
|---|---|---|---|---|
| *inBC* | ✗ | **57.3** | **46.2** | **42.9** |
| *inBC* | ✓ | 44.6 | 27.7 | 34.2 |
| *D-inBC* + OD | ✗ | **63.5** | **51.5** | **50.0** |
| *D-inBC* + OD | ✓ | 27.7 | 24.6 | 34.2 |
| *D-inBC* + Aux (C) | ✗ | **64.6** | **58.8** | **49.6** |
| *D-inBC* + Aux (C) | ✓ | 31.2 | 28.5 | 27.1 |

## B.6    ABLATION ON OBJECT DETECTOR

This ablation studies three types of object detectors in this paper: the VLM model itself, an external object detector separately trained on the training set (the methods with a suffix *OD*), and an oracle from the groundtruth dataset (Oracle). Fig. 12 shows that a reliable object detector is highly beneficial, enhancing the accuracy of image-based inputs and consequently improving model performance.

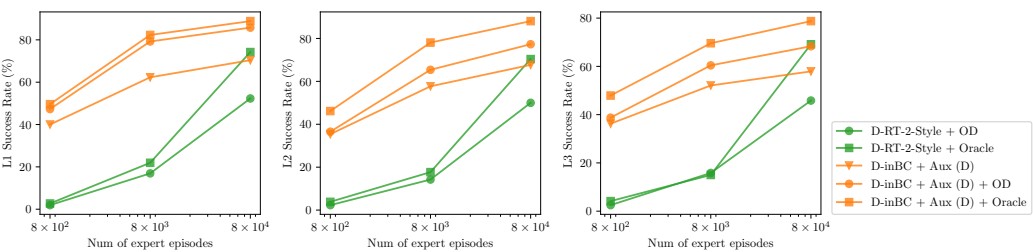

Figure 12: Impact of different object detectors.

### B.7 ABLATION ON ACTION PROMPT DURING INFERENCE

This ablation studies the effect of the prompt sentence used in action generation. By default, during test time we formulate an instruction similar to Tab. 8 to query the VLM. In addition, we randomly sample one sentence from Tab. 28, named action prompt, and add it to the instruction. We study the following scenarios to understand the effect of this action prompt selection:

- *random*: this is the default behavior, a random action prompt will be added to each instruction;

- *no_prompt*: none instruction has the action prompt;

- *prompt $p_a$*, where $p_a \in [0, 14]$: all the instructions in this setting use the $p_a^{th}$ prompt (0-index) in Tab. 28.

The results are presented in Fig. 13, where two variants of $\mathcal{D}$-*inBC* are trained on VIMA-80k. In general, the selection of the action prompt does not result in a significant difference in performance except that $\mathcal{D}$-*inBC* + Aux (D) seems to perform better on L1 and L3 without the action prompt.

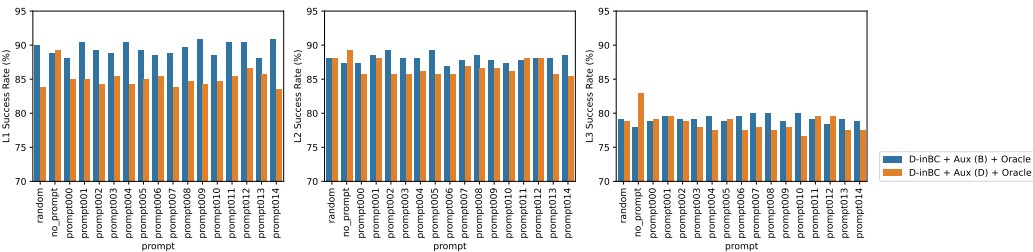

Figure 13: Effect of prompt sentence for action generation.

### B.8 EXTENDED EXPERIMENTS ON *RT-2 Style* VARIANTS

As suggested by the reviewer, we evaluate two new variants of the *RT-2 Style* settings, *RT-2 Style (2R)* and *RT-2 Style* + Aux (initially reported in Tab. 3). In the original *RT-2 Style* setting, a single token is assigned to present the rotation in the action space. However, considering there are only 256 possible values of a token, it causes an unfair comparison between our method and *RT-2 Style* in terms of action space of rotation and may compromise *RT-2 Style*'s performance. Therefore, we propose *RT-2 Style (2R)*, which uses two tokens to represent rotation, resulting in 65,536 options to encode rotation. This new configuration introduces a significantly larger action space compared to our method. At the same time, we evaluate another variant *RT-2 Style* + Aux (D) by simply mixing *RT-2 Style* training data with auxiliary datasets used in LLaRA. Evaluation results of these models on VIMA-8k are summarized in Tab. 16.

Table 16: Performance of *RT-2 Style* variants on VIMA-8k

| Method | L1 (%) | L2 (%) | L3 (%) |
|---|---|---|---|
| *RT-2 Style* | 3.8 | 3.1 | 1.7 |
| *RT-2 Style (2R)* | 1.9 | 2.3 | 1.7 |
| *RT-2 Style* + Aux (D) | 36.9 | 35.8 | 32.1 |
| *inBC* | 57.3 | 46.2 | 42.9 |
| *inBC* + Aux (D) | 59.2 | 58.8 | 52.1 |

As shown in Tab. 16, an additional token for rotation in *RT-2 Style (2R)* does not significantly change the performance of the *RT-2 Style* baseline, however, the introduction of auxiliary data significantly improves the performance of *RT-2 Style*, demonstrating the effectiveness of our method.

## B.9 Robustness of the model in failure grasp scenarios

As suggested by the reviewer, to show the robustness of our method, especially the ability to recover from failure, we run additional tests in a revised VIMA environment. The revised environment has a 10% chance that ignores the action from the model to simulate the action failure, which we believe is much higher than the actual case.

Tab. 17 shows that without any changes, our existing model is robust to such failure.

Table 17: Robostness evaluation results of $\mathcal{D}$-inBC + Aux (B) + Oracle (VIMA-80k, 8 epochs)

| Prob. of Failure | L1 (%) | L2 (%) | L3 (%) |
|---|---|---|---|
| 0 | 90.0 | 88.1 | 79.2 |
| 10 | 87.7 | 86.2 | 75.8 |

## B.10 Few-shot experiments using LLM / VLM

We expanded our evaluation to include the few-shot performance of existing Large Language Models (LLMs) and Vision Language Models (VLMs). Consistent with the inference settings described in Appendix A.3, we provided three additional examples from either *inBC* or *$\mathcal{D}$-inBC* for each test scenario. The results of these evaluations are detailed in Tab. 18, where we examine three variables:

**Vision**: When this option is enabled, the model will have access to the current image observation, allowing it to integrate visual data into its response. If disabled, the model operates solely as a text-based system.
**Oracle**: This variable is checked when the examples are drawn from the $\mathcal{D}$-inBC dataset. It indicates that the examples provided for few-shot learning include descriptive textual information about the reference images, potentially offering richer context.
**Same-task**: When this is checked, the examples used for prompting are from the same task type as those in the evaluation environment.

From the observed low performance, we infer that the data utilized for robot control significantly deviates from the conversation contexts in which these models were originally trained. This indicates a pressing need to further finetune the models specifically for robot control tasks to achieve adequate performance. Additionally, our findings hint that visual input considerably enhances the functionality of GPT-4o.

Table 18: Three-shot performance of LLMs / VLMs

| Output | Vision | Oracle | Same-task | L1 (%) | L2 (%) | L3 (%) | L4 (%) |
|---|---|---|---|---|---|---|---|
| GPT-4o | ✗ | ✗ | ✓ | 1.4 | 1.4 | 0.0 | N/A |
| GPT-4o | ✗ | ✗ | ✗ | 0.4 | 0.4 | 0.4 | 0.0 |
| GPT-4o | ✗ | ✓ | ✓ | 1.4 | 1.8 | 0.5 | N/A |
| GPT-4o | ✗ | ✓ | ✗ | 1.2 | 0.4 | 0.4 | 1.2 |
| GPT-4o | ✓ | ✗ | ✓ | 5.9 | 4.5 | 4.0 | N/A |
| GPT-4o | ✓ | ✗ | ✗ | 6.9 | 5.8 | 4.2 | 1.2 |
| Llama 3 70B (4-bit) | ✗ | ✗ | ✓ | 0.0 | 0.0 | 0.0 | N/A |
| Llama 3 70B (4-bit) | ✗ | ✗ | ✗ | 0.8 | 1.2 | 0.4 | 1.2 |
| Llama 3 70B (4-bit) | ✗ | ✓ | ✓ | 0.0 | 1.4 | 0.5 | N/A |
| Llama 3 70B (4-bit) | ✗ | ✓ | ✗ | 0.0 | 0.4 | 0.4 | 0.0 |
| LlaVA-1.5-7B | ✓ | ✗ | ✓ | 0.9 | 1.4 | 0.5 | N/A |
| LlaVA-1.5-7B | ✓ | ✗ | ✗ | 0.0 | 0.0 | 1.0 | 1.2 |
| LlaVA-1.6-34B | ✓ | ✗ | ✓ | 1.4 | 1.4 | 0.5 | N/A |
| LlaVA-1.6-34B | ✓ | ✗ | ✗ | 1.2 | 0.8 | 2.1 | 0.0 |

## C  DETAILS ON REAL-WORLD ROBOT EXPERIMENTS

In this section, we provide more details on real-world robot experiments first introduced in Sec. 6.2.

### C.1  ENVIRONMENT SETTING

We utilize an xArm7 robot arm equipped with a gripper and a Logitech C140 RGB webcam positioned above the arm to gather observations. The camera is mounted in a fixed position above and slightly in front of the arm, providing a third-person view of the scene. The captured images have a raw resolution of $640 \times 480$, which are then center-cropped to $640 \times 320$ to match the aspect ratio of VIMA images. The robot's action space includes 4 DoF for pick-and-place positioning and 1 DoF for rotation, consistent with the simplified VIMA setup.

The object detector for all real-world experiments is an OWLv2 (Minderer et al., 2024) (ViT-base, patch size 16) running in one-shot detection mode. The object detector takes a prompt image from the same domain and detects the objects in the observation. The list of plastic toys we used as {object} (and detected) is: {duck, croissant, cupcake, blueberries, carrot, banana, grapes, donut, tomato, corn, pepper}

### C.2  GPT-4O BASELINE

In the GPT-4o baseline, the model is fed the same prompt as in RT-2, with an additional prompt describing the robot arm: "You are operating a robot arm at a table to complete a task specified between <task></task>. Given the current image, you need to generate a series of actions to control the robot arm to accomplish the task.". The model's answer is parsed to extract the pick and place points.

### C.3  REAL-WORLD ROBOT DATASET

We further collect 1,256 real-world images from the same real-world robot setting as the in-domain data. Out of these images, 721 feature a single object placed randomly and uniformly on the table by a robotic arm, while the remaining images display multiple objects scattered across the table. We employed one-shot detection using OWLv2 (Minderer et al., 2024) to identify and generate bounding boxes for each object in the images.

These images, along with their bounding box data, have been utilized to create two specialized in-domain instruction tuning datasets, each containing 2,447 single-turn conversations of equivalent size: xArm-Det and xArm-Action. The xArm-Det dataset includes conversations that mirror the structure from our auxiliary object localization and detection dataset, with 1,191 examples dedicated to object localization and 1,256 to object detection. In contrast, the xArm-Action dataset is designed similarly to data for $\mathcal{D}$-inBC, featuring two distinct tasks: *Rotation*: rotating an object (akin to Task T2 with 1,191 examples) and *Put on top*: putting one object on top of another (reminiscent of Task T3 with 1,256 examples).

In the *Rotation* task, the agent is instructed to rotate a specific object by a random angle ranging from -180 to 180 degrees, while ensuring the object's position remains unchanged post-rotation. The expert action is straightforwardly generated by directly replicating the specified rotation angle from the task description and the pick and place points are set to the centroid of the object bounding box from object detection.

In the *Put on Top* task, applicable to images containing more than two objects, we randomly select two objects. The task involves generating a command that directs placing one object atop the other. The expert actions, specifically the pick and place points, are determined using the centroids of the bounding boxes of the two selected objects from object detection.

Tab. 19 shows examples of the real-world datasets we collected.

Table 19: Qualitative examples of our real-world datasets xArm-Det and xArm-Action. The description of an object in the image (one-shot detection results) is in blue. The task description of the episode is in green and the action is in orange.

| | | |
|---|---|---|
| **Image Input** | | |
| **xArm-Det** Localization | **Input** | Could you tell me the location of <p>blueberry</p> in the image? Use the format (x, y), {w, h}, where x and y denote the center of the bounding box and w and h are the width and height. Coordinates start at the top left corner and should be normalized between 0 and 1. |
| | **Output** | <p>blueberry</p> at (0.478, 0.725), {0.058, 0.125}. |
| **xArm-Det** Detection | **Input** | Describe all the objects seen in the image, and list them using the format (x, y), {w, h}. The x and y values are the coordinates for the center of the bounding box, while w and h represent its width and height. The coordinates should be normalized from the top left corner, within a range of 0 to 1. |
| | **Output** | <scene> <p>cupcake</p> at (0.440, 0.128), {0.063, 0.123}. <p>corn</p> at (0.237, 0.826), {0.061, 0.246}. <p>carrot</p> at (0.608, 0.135), {0.152, 0.113}. <p>donut</p> at (0.396, 0.355), {0.079, 0.145}. <p>grape</p> at (0.318, 0.534), {0.077, 0.143}. <p>duck</p> at (0.795, 0.280), {0.066, 0.128}. <p>green_pepper</p> at (0.688, 0.411), {0.062, 0.187}. <p>blueberry</p> at (0.478, 0.725), {0.058, 0.125}. <p>tomato</p> at (0.543, 0.421), {0.078, 0.154}. </scene> |
| **xArm-Action** Rotation | **Input** | <task>Rotate the <p>cupcake</p> at (0.440, 0.128), {0.063, 0.123} by -50 degrees.</task> Every action you take must include two locations in the format of (x, y) and one clockwise rotation angle in the format of <r>[r]</r>. The first location is the image coordinate where you use a suction cup to pick up the object, and the second location is where you place the object. The image coordinate ranges from 0 to 1. The rotation angle indicates how many degrees you rotate the object clockwise, and it ranges from -359 to 359. |
| | **Output** | Pick up the <p>cupcake</p> at (0.440, 0.128), rotate <r>[-50]</r> degrees, and drop it at (0.440, 0.128). |
| **xArm-Action** Put on top | **Input** | <task>Move the <p>cupcake</p> at (0.440, 0.128), {0.063, 0.123} on the top of the <p>corn</p> at (0.237, 0.826), {0.061, 0.246}.</task> Every action you take must include two locations in the format of (x, y) and one clockwise rotation angle in the format of <r>[r]</r>. The first location is the image coordinate where you use a suction cup to pick up the object, and the second location is where you place the object. The image coordinate ranges from 0 to 1. The rotation angle indicates how many degrees you rotate the object clockwise, and it ranges from -359 to 359. |
| | **Output** | Step 1: Pick up the <p>cupcake</p> at (0.440, 0.128), rotate <r>[0]</r> degrees, and drop it at (0.237, 0.826). |

## C.4 EXTENDED REAL-WORLD EXPERIMENTS

In addition, following the suggestion from the reviewer, we have added experiments with a new set of tasks that require more language understanding and spatial reasoning. These tasks include:

- **T5: Put all yellow objects into the bowl**. There is a bowl and 3 objects on the table. Between 1 and 3 of the objects are yellow. A success state is one in which all yellow objects are in the bowl and no non-yellow objects are in the bowl.

- **T6: Transport the baked goods to the large bowl**. There is a bowl and 3 objects on the table. Either 1 or 2 of the objects are baked goods. A success state is one in which all baked goods are in the large bowl and none of the non-baked goods are in the large bowl.

- **T7: Feed the pet healthy food**. There is an animal on the table and 2 objects. One of the objects is a fruit or vegetable and the other is a baked good. The 2 objects are equidistant from the animal to start. A success state is one in which the healthy food is sitting next to the animal while the unhealthy food is further away from the animal.

Without a proper language understanding, these tasks cannot be easily solved. These tasks are particularly challenging as none of these tasks were seen in the training data in either sim or real. Some of the objects in the scene (e.g., baked goods and bowl) were never seen by the model either during the robot data training, and each episode takes multiple steps to finish.

We compare *RT-2 Style* and LLaRA (inBC + Aux (D)) both first trained on VIMA-8k and then finetuned on xArm-Det, which is the model used in Tab. 4 without any changes. That is, the new tasks and the unseen tasks for the model. For each task, we run 10 randomly initialized episodes and report the success rate. Tab. 20 shows the results.

Table 20: Real-world experiment results of more unseen tasks that require challenging semantic understanding.

| Method | Add. Data | T5 (%) | T6 (%) | T7 (%) |
|---|---|---|---|---|
| *RT-2 Style* | xArm-Det | 0 | 0 | 0 |
| *inBC* + Aux (D) | xArm-Det | 60 | 55 | 50 |

As we are able to observe, LLaRA is able to handle tasks requiring spatial language understanding tasks to a certain degree. It sometimes gets confused due to its smaller LLM size (7B), but it shows the potential to effectively leverage the knowledge of LLaVA, enabling efficient transfer to robot control with minimal data. This capability is primarily facilitated by our key contributions: the alignment between action and image coordinates, and the integration of self-supervised auxiliary data.

*RT-2 Style* on the other hand, simply fails to utilize the limited amount of training data provided in our setting, despite using the same model architecture.

## C.5 QUALITATIVE EXAMPLES ON UNSEEN REAL-WORLD TASKS

In Fig. 14 we qualitatively show that LLaRA can complete some unseen tasks that are not in the training dataset. In the first example, we ask LLaRA to `place all other objects into the large bowl`. LLaRA is able to recognize the objects to move and not to move and outputs the correct actions. In the second example, we let LLaRA `get the weight of the tomato and put it back`. LLaRA can put the tomato on the scale and move the tomato back to the similar place as the initial position, showing the understanding of 'putting it back'. These examples show the potential generalization ability of LLaRA. Video demonstrations are also included in the supplementary material.

## D LIMITATIONS

While our model has demonstrated significant achievements and holds considerable potential, it is important to acknowledge that there remains space for further optimization.

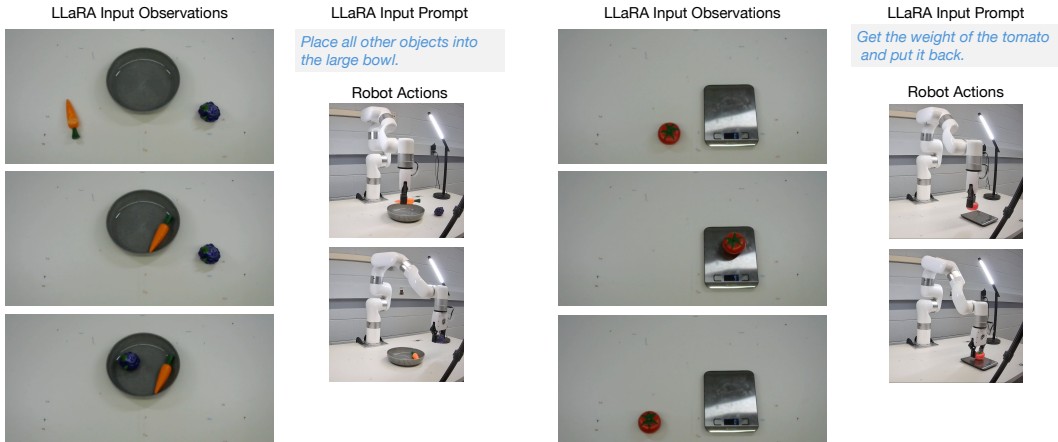

Figure 14: Examples of using LLaRA for unseen tasks. We show the input texts from the user and input images at each step of the task, and images about the robot's actions to accomplish each step.

First, some concepts in the reference images still can not be easily and precisely described by language, while in many cases these features are critical for robot control. Due to the single image limit of the current VLM, we still rely on the object detector trained on a limited number of classes to describe the additional images in the task description, which limits the generalization ability of this method. In principle, in the future, this can be addressed by a VLM trained on interleave image text data like (Bai et al., 2023; Li et al., 2024a).

Second, the information extracted from the dataset can be noisy, which may lead to model confusion. In Tab. 8, there are discrepancies such as the location of an object in a reference image differing from its actual location in the current image observation. At the same time, in VIMA dataset, when referring to the shape of an object without a color, the object in the reference image will appear in gray, which is a totally different color than the object with the same shape in the current observations.

Finally, our method relies on 2D to 2D mapping to convert image coordinates into actual actions, which may face challenges when the complex 3D movement of the robot is required.

We believe enhancements in the mentioned aspects could further improve performance and broader applicability in complex, real-world applications.

## E    RAW RESULTS AND PROMPT TEMPLATES

This section contains all the numerical results of VIMA experiments and all prompt templates we used in the paper.

Table 21: Results on VIMA-0.8k dataset. **Ep**: number of epoch; **U**: normalized number of model iterations. The following sizes of datasets are relative to the size of *inBC*. **Loc.**: relative size of the localization dataset; **Det.**: relative size of the detection dataset; **Act.**: relative size of the action prediction dataset; **Fut.**: relative size of the future prediction dataset; **Spa.**: relative size of the spatial relationship dataset; **Temp.**: relative size of the temporal relationship dataset. The '*' means the reference images that appeared only in the task description are not used to generate this dataset.

| Method | Ep | U | Loc. | Det. | Act. | Fut. | Spa. | Temp. | L1 (%) | L2 (%) | L3 (%) |
|---|---|---|---|---|---|---|---|---|---|---|---|
| *RT-2 Style* | 1 | 1 | | | | | | | 0.0 | 0.4 | 0.4 |
| *RT-2 Style* | 2 | 2 | | | | | | | 1.2 | 0.4 | 0.8 |
| *RT-2 Style* | 4 | 4 | | | | | | | 1.5 | 1.2 | 2.5 |
| *RT-2 Style* | 6 | 6 | | | | | | | 2.3 | 1.5 | 0.8 |
| *RT-2 Style* | 8 | 8 | | | | | | | 2.3 | 3.5 | 2.1 |
| *inBC* | 1 | 1 | | | | | | | 23.1 | 16.5 | 16.7 |
| *inBC* | 2 | 2 | | | | | | | 30.0 | 23.8 | 19.6 |
| *inBC* | 4 | 4 | | | | | | | 22.3 | 21.2 | 15.4 |
| *inBC* | 6 | 6 | | | | | | | 21.9 | 14.6 | 17.5 |
| *inBC* | 8 | 8 | | | | | | | 20.4 | 17.7 | 18.3 |
| *inBC* | 10 | 10 | | | | | | | 20.4 | 18.1 | 14.6 |
| *inBC* + Aux | 1 | 1.2 | 0.1* | 0.1* | | | | | 13.8 | 13.1 | 9.2 |
| *inBC* + Aux | 1 | 3 | 1* | 1* | | | | | 28.8 | 26.5 | 24.2 |
| *inBC* + Aux | 2 | 2.4 | 0.1* | 0.1* | | | | | 29.6 | 21.9 | 23.3 |
| *inBC* + Aux | 2 | 2.4 | 0.1 | 0.1 | | | | | 23.5 | 20.4 | 17.5 |
| *inBC* + Aux | 2 | 4 | 0.5 | 0.5 | | | | | 26.9 | 23.8 | 24.6 |
| *inBC* + Aux | 2 | 4 | 0.2 | 0.2 | 0.2 | 0.2 | | | 35.0 | 24.2 | 26.7 |
| *inBC* + Aux | 2 | 4 | 0.2 | 0.2 | 0.2 | 0.2 | 0.2 | | 33.5 | 26.2 | 21.7 |
| *inBC* + Aux | 2 | 6 | 1* | 1* | | | | | 31.9 | 26.5 | 30.4 |
| *inBC* + Aux | 2 | 6 | 1 | 1 | | | | | 30.4 | 27.3 | 27.1 |
| *inBC* + Aux | 2 | 7 | 0.5 | 0.5 | 0.5 | 0.5 | 0.5 | | 36.2 | 29.6 | 29.2 |
| *inBC* + Aux | 2 | 8 | 0.5* | 0.5* | 0.5 | 0.5 | 0.5* | 0.5 | 33.5 | 29.2 | 33.3 |
| *inBC* + Aux | 2 | 8 | 0.5 | 0.5 | 0.5 | 0.5 | 0.5 | 0.5 | 31.9 | 29.6 | 26.7 |
| *inBC* + Aux | 2 | 10 | 1 | 1 | 1 | 1 | | | 37.7 | 30.8 | 28.7 |
| *inBC* + Aux | 2 | 12 | 1 | 1 | 1 | 1 | 1 | | 38.8 | 34.6 | 29.6 |
| *inBC* + Aux | 2 | 14 | 1* | 1* | 1 | 1 | 1* | 1 | 37.7 | 29.6 | 27.9 |
| *inBC* + Aux | 2 | 14 | 1 | 1 | 1 | 1 | 1 | 1 | 40.0 | 35.8 | 31.2 |
| $\mathcal{D}$-*inBC* + Aux | 1 | 1.2 | 0.1* | 0.1* | | | | | 18.5 | 17.7 | 13.8 |
| $\mathcal{D}$-*inBC* + Aux | 1 | 3 | 1* | 1* | | | | | 28.8 | 25.4 | 23.3 |
| $\mathcal{D}$-*inBC* + Aux | 2 | 2.4 | 0.1* | 0.1* | | | | | 23.5 | 16.2 | 22.5 |
| $\mathcal{D}$-*inBC* + Aux | 2 | 2.4 | 0.1 | 0.1 | | | | | 21.5 | 21.2 | 23.3 |
| $\mathcal{D}$-*inBC* + Aux | 2 | 4 | 0.5 | 0.5 | | | | | 28.1 | 25.8 | 21.7 |
| $\mathcal{D}$-*inBC* + Aux | 2 | 4 | 0.2 | 0.2 | 0.2 | 0.2 | | | 32.3 | 29.6 | 22.1 |
| $\mathcal{D}$-*inBC* + Aux | 2 | 4 | 0.2 | 0.2 | 0.2 | 0.2 | 0.2 | | 30.0 | 26.5 | 25.4 |
| $\mathcal{D}$-*inBC* + Aux | 2 | 6 | 1* | 1* | | | | | 33.8 | 30.8 | 29.2 |
| $\mathcal{D}$-*inBC* + Aux | 2 | 6 | 1 | 1 | | | | | 35.8 | 28.5 | 28.3 |
| $\mathcal{D}$-*inBC* + Aux | 2 | 7 | 0.5 | 0.5 | 0.5 | 0.5 | 0.5 | | 38.5 | 26.9 | 30.8 |
| $\mathcal{D}$-*inBC* + Aux | 2 | 8 | 0.5* | 0.5* | 0.5 | 0.5 | 0.5* | 0.5 | 37.3 | 26.5 | 34.6 |
| $\mathcal{D}$-*inBC* + Aux | 2 | 8 | 0.5 | 0.5 | 0.5 | 0.5 | 0.5 | 0.5 | 34.2 | 29.6 | 30.0 |
| $\mathcal{D}$-*inBC* + Aux | 2 | 10 | 1 | 1 | 1 | 1 | | | 34.6 | 33.1 | 28.3 |
| $\mathcal{D}$-*inBC* + Aux | 2 | 12 | 1 | 1 | 1 | 1 | 1 | | 40.0 | 38.5 | 37.5 |
| $\mathcal{D}$-*inBC* + Aux | 2 | 14 | 1* | 1* | 1 | 1 | 1* | 1 | 40.8 | 30.4 | 32.9 |
| $\mathcal{D}$-*inBC* + Aux | 2 | 14 | 1 | 1 | 1 | 1 | 1 | 1 | 40.0 | 35.4 | 36.2 |

Table 22: Results on VIMA-0.8k dataset with object detector or Oracle. **Ep**: number of epoch; **U**: normalized number of model iterations. The following sizes of datasets are relative to the size of *inBC*. **Loc.**: relative size of the localization dataset; **Det.**: relative size of the detection dataset; **Act.**: relative size of the action prediction dataset; **Fut.**: relative size of the future prediction dataset; **Spa.**: relative size of the spatial relationship dataset; **Temp.**: relative size of the temporal relationship dataset.

| Method | Ep | U | Loc. | Det. | Act. | Fut. | Spa. | Temp. | L1 (%) | L2 (%) | L3 (%) |
|---|---|---|---|---|---|---|---|---|---|---|---|
| $\mathcal{D}$-RT-2 *Style* + OD | 1 | 1 | | | | | | | 0.0 | 0.0 | 0.0 |
| $\mathcal{D}$-RT-2 *Style* + OD | 2 | 2 | | | | | | | 0.4 | 1.5 | 2.5 |
| $\mathcal{D}$-RT-2 *Style* + OD | 4 | 4 | | | | | | | 1.9 | 1.5 | 2.5 |
| $\mathcal{D}$-RT-2 *Style* + OD | 6 | 6 | | | | | | | 1.9 | 2.3 | 0.8 |
| $\mathcal{D}$-RT-2 *Style* + OD | 8 | 8 | | | | | | | 1.9 | 1.2 | 0.0 |
| $\mathcal{D}$-*inBC* + OD | 1 | 1 | | | | | | | 17.3 | 11.9 | 12.5 |
| $\mathcal{D}$-*inBC* + OD | 2 | 2 | | | | | | | 17.3 | 14.6 | 15.0 |
| $\mathcal{D}$-*inBC* + OD | 4 | 4 | | | | | | | 32.7 | 24.2 | 24.2 |
| $\mathcal{D}$-*inBC* + OD | 6 | 6 | | | | | | | 28.1 | 23.1 | 21.7 |
| $\mathcal{D}$-*inBC* + OD | 8 | 8 | | | | | | | 26.2 | 17.7 | 22.9 |
| $\mathcal{D}$-*inBC* + Aux + OD | 2 | 6 | 1 | 1 | | | | | 36.9 | 25.8 | 30.4 |
| $\mathcal{D}$-*inBC* + Aux + OD | 2 | 10 | 1 | 1 | 1 | 1 | | | 41.2 | 31.9 | 32.9 |
| $\mathcal{D}$-*inBC* + Aux + OD | 2 | 12 | 1 | 1 | 1 | 1 | 1 | | 44.6 | 33.5 | 37.9 |
| $\mathcal{D}$-*inBC* + Aux + OD | 2 | 14 | 1 | 1 | 1 | 1 | 1 | 1 | 47.3 | 36.5 | 38.8 |
| $\mathcal{D}$-RT-2 *Style* + Oracle | 1 | 1 | | | | | | | 0.0 | 0.4 | 0.8 |
| $\mathcal{D}$-RT-2 *Style* + Oracle | 2 | 2 | | | | | | | 0.8 | 0.8 | 1.7 |
| $\mathcal{D}$-RT-2 *Style* + Oracle | 4 | 4 | | | | | | | 1.9 | 3.8 | 4.2 |
| $\mathcal{D}$-RT-2 *Style* + Oracle | 6 | 6 | | | | | | | 2.7 | 3.5 | 0.8 |
| $\mathcal{D}$-RT-2 *Style* + Oracle | 8 | 8 | | | | | | | 1.2 | 2.7 | 1.2 |
| $\mathcal{D}$-RT-2 *Style* (I) + Oracle | 1 | 1 | | | | | | | 1.2 | 0.0 | 0.8 |
| $\mathcal{D}$-RT-2 *Style* (I) + Oracle | 2 | 2 | | | | | | | 2.3 | 1.9 | 1.7 |
| $\mathcal{D}$-RT-2 *Style* (I) + Oracle | 4 | 4 | | | | | | | 0.8 | 0.4 | 0.8 |
| $\mathcal{D}$-RT-2 *Style* (I) + Oracle | 6 | 6 | | | | | | | 1.5 | 2.7 | 0.4 |
| $\mathcal{D}$-RT-2 *Style* (I) + Oracle | 8 | 8 | | | | | | | 0.8 | 1.9 | 0.4 |
| $\mathcal{D}$-*inBC* + Oracle | 1 | 1 | | | | | | | 16.9 | 13.5 | 10.4 |
| $\mathcal{D}$-*inBC* + Oracle | 2 | 2 | | | | | | | 20.0 | 19.2 | 17.5 |
| $\mathcal{D}$-*inBC* + Oracle | 4 | 4 | | | | | | | 37.7 | 26.2 | 29.2 |
| $\mathcal{D}$-*inBC* + Oracle | 6 | 6 | | | | | | | 30.0 | 28.1 | 27.5 |
| $\mathcal{D}$-*inBC* + Oracle | 8 | 8 | | | | | | | 27.7 | 23.1 | 27.9 |
| $\mathcal{D}$-*inBC* + Aux + Oracle | 2 | 6 | 1 | 1 | | | | | 45.0 | 36.9 | 39.2 |
| $\mathcal{D}$-*inBC* + Aux + Oracle | 2 | 10 | 1 | 1 | 1 | 1 | | | 43.8 | 38.5 | 37.9 |
| $\mathcal{D}$-*inBC* + Aux + Oracle | 2 | 12 | 1 | 1 | 1 | 1 | 1 | | 52.7 | 44.6 | 50.4 |
| $\mathcal{D}$-*inBC* + Aux + Oracle | 2 | 14 | 1 | 1 | 1 | 1 | 1 | 1 | 49.6 | 46.2 | 47.9 |

Table 23: Results on VIMA-8k dataset. **Ep**: number of epoch; **U**: normalized number of model iterations. The following sizes of datasets are relative to the size of *inBC*. **Loc.**: relative size of the localization dataset; **Det.**: relative size of the detection dataset; **Act.**: relative size of the action prediction dataset; **Fut.**: relative size of the future prediction dataset; **Spa.**: relative size of the spatial relationship dataset; **Temp.**: relative size of the temporal relationship dataset. The '*' means the reference images that appeared only in the task description are not used to generate this dataset.

| Method | Ep | U | Loc. | Det. | Act. | Fut. | Spa. | Temp. | L1 (%) | L2 (%) | L3 (%) |
|---|---|---|---|---|---|---|---|---|---|---|---|
| *RT-2 Style* | 1 | 1 | | | | | | | 3.1 | 3.5 | 0.8 |
| *RT-2 Style* | 2 | 2 | | | | | | | 3.8 | 3.1 | 1.7 |
| *RT-2 Style (2R)* | 2 | 2 | | | | | | | 1.9 | 2.3 | 1.7 |
| *RT-2 Style* | 4 | 4 | | | | | | | 8.1 | 6.5 | 5.8 |
| *RT-2 Style* | 6 | 6 | | | | | | | 6.5 | 5.4 | 4.2 |
| *RT-2 Style* | 8 | 8 | | | | | | | 6.2 | 6.5 | 4.2 |
| *RT-2 Style* + Aux | 2 | 14 | 1 | 1 | 1 | 1 | 1 | 1 | 36.9 | 35.8 | 32.1 |
| *inBC* | 1 | 1 | | | | | | | 44.6 | 34.2 | 36.2 |
| *inBC* | 2 | 2 | | | | | | | 57.3 | 46.2 | 42.9 |
| *inBC* | 4 | 4 | | | | | | | 58.5 | 48.5 | 47.1 |
| *inBC* | 6 | 6 | | | | | | | 54.6 | 45.4 | 40.4 |
| *inBC* | 8 | 8 | | | | | | | 52.3 | 47.3 | 47.1 |
| *inBC* | 10 | 10 | | | | | | | 53.1 | 47.7 | 47.9 |
| *inBC* + Aux | 2 | 2.4 | 0.1* | 0.1* | | | | | 59.6 | 50.0 | 45.4 |
| *inBC* + Aux | 2 | 2.4 | 0.1 | 0.1 | | | | | 57.3 | 52.3 | 45.4 |
| *inBC* + Aux | 2 | 4 | 0.5 | 0.5 | | | | | 60.4 | 56.2 | 52.1 |
| *inBC* + Aux | 2 | 4 | 0.2 | 0.2 | 0.2 | 0.2 | | | 59.6 | 59.2 | 51.2 |
| *inBC* + Aux | 2 | 6 | 1* | 1* | | | | | 59.2 | 52.3 | 48.3 |
| *inBC* + Aux | 2 | 6 | 1 | 1 | | | | | 57.7 | 56.2 | 53.8 |
| *inBC* + Aux | 2 | 7 | 0.5 | 0.5 | 0.5 | 0.5 | 0.5 | | 58.1 | 55.0 | 47.9 |
| *inBC* + Aux | 2 | 10 | 1 | 1 | 1 | 1 | | | 60.4 | 55.8 | 54.2 |
| *inBC* + Aux | 2 | 12 | 1 | 1 | 1 | 1 | 1 | | 60.8 | 55.4 | 50.0 |
| *inBC* + Aux | 2 | 14 | 1* | 1* | 1 | 1 | 1* | 1 | 63.1 | 58.8 | 53.8 |
| *inBC* + Aux | 2 | 14 | 1 | 1 | 1 | 1 | 1 | 1 | 59.2 | 58.8 | 52.1 |
| *D-inBC* + Aux | 2 | 2.4 | 0.1* | 0.1* | | | | | 53.5 | 48.8 | 45.4 |
| *D-inBC* + Aux | 2 | 2.4 | 0.1 | 0.1 | | | | | 58.5 | 55.8 | 50.8 |
| *D-inBC* + Aux | 2 | 4 | 0.5 | 0.5 | | | | | 58.8 | 50.8 | 54.6 |
| *D-inBC* + Aux | 2 | 4 | 0.2 | 0.2 | 0.2 | 0.2 | | | 58.1 | 52.3 | 51.2 |
| *D-inBC* + Aux | 2 | 6 | 1* | 1* | | | | | 63.1 | 57.7 | 55.4 |
| *D-inBC* + Aux | 2 | 6 | 1 | 1 | | | | | 60.0 | 53.5 | 50.0 |
| *D-inBC* + Aux | 2 | 7 | 0.5 | 0.5 | 0.5 | 0.5 | 0.5 | | 62.3 | 55.8 | 48.8 |
| *D-inBC* + Aux | 2 | 10 | 1 | 1 | 1 | 1 | | | 63.1 | 58.1 | 49.6 |
| *D-inBC* + Aux | 2 | 12 | 1 | 1 | 1 | 1 | 1 | | 64.6 | 58.8 | 49.6 |
| *D-inBC* + Aux | 2 | 14 | 1* | 1* | 1 | 1 | 1* | 1 | 62.3 | 54.6 | 51.2 |
| *D-inBC* + Aux | 2 | 14 | 1 | 1 | 1 | 1 | 1 | 1 | 62.3 | 57.7 | 52.1 |
| *D-inBC (L)* + Aux | 2 | 14 | 1 | 1 | 1 | 1 | 1 | 1 | 61.9 | 54.2 | 55.4 |
| *D-inBC* + Aux | 4 | 4.8 | 0.1 | 0.1 | | | | | 58.5 | 47.7 | 49.6 |
| *D-inBC* + Aux | 4 | 12 | 1 | 1 | | | | | 61.2 | 57.7 | 49.6 |
| *D-inBC* + Aux | 4 | 20 | 1 | 1 | 1 | 1 | | | 62.3 | 56.5 | 49.6 |
| *D-inBC* + Aux | 4 | 24 | 1 | 1 | 1 | 1 | 1 | | 63.5 | 56.2 | 52.5 |

Table 24: Results on VIMA-8k dataset with object detector or Oracle. **Ep**: number of epoch; **U**: normalized number of model iterations. The following sizes of datasets are relative to the size of *inBC*. **Loc.**: relative size of the localization dataset; **Det.**: relative size of the detection dataset; **Act.**: relative size of the action prediction dataset; **Fut.**: relative size of the future prediction dataset; **Spa.**: relative size of the spatial relationship dataset; **Temp.**: relative size of the temporal relationship dataset. The '*' means the reference images that appeared only in the task description are not used to generate this dataset.

| Method | Ep | U | Loc. | Det. | Act. | Fut. | Spa. | Temp. | L1 (%) | L2 (%) | L3 (%) |
|---|---|---|---|---|---|---|---|---|---|---|---|
| $\mathcal{D}$-RT-2 Style + OD | 1 | 1 | | | | | | | 1.5 | 3.1 | 1.2 |
| $\mathcal{D}$-RT-2 Style + OD | 2 | 2 | | | | | | | 10.4 | 8.8 | 7.1 |
| $\mathcal{D}$-RT-2 Style + OD | 4 | 4 | | | | | | | 16.9 | 14.2 | 15.8 |
| $\mathcal{D}$-RT-2 Style + OD | 6 | 6 | | | | | | | 14.2 | 11.5 | 12.9 |
| $\mathcal{D}$-RT-2 Style + OD | 8 | 8 | | | | | | | 15.8 | 10.4 | 6.7 |
| $\mathcal{D}$-inBC + OD | 1 | 1 | | | | | | | 55.4 | 45.8 | 45.0 |
| $\mathcal{D}$-inBC + OD | 2 | 2 | | | | | | | 63.5 | 51.5 | 50.0 |
| $\mathcal{D}$-inBC + OD | 4 | 4 | | | | | | | 63.8 | 46.9 | 55.8 |
| $\mathcal{D}$-inBC + Aux + OD | 2 | 6 | 1 | 1 | | | | | 74.2 | 58.5 | 56.7 |
| $\mathcal{D}$-inBC + Aux + OD | 2 | 10 | 1 | 1 | 1 | 1 | | | 75.4 | 66.2 | 57.9 |
| $\mathcal{D}$-inBC + Aux + OD | 2 | 12 | 1 | 1 | 1 | 1 | 1 | | 76.9 | 63.8 | 57.9 |
| $\mathcal{D}$-inBC + Aux + OD | 2 | 12 | 1 | 1 | 1 | 1 | 1 | | 79.2 | 64.2 | 58.3 |
| $\mathcal{D}$-inBC + Aux + OD | 2 | 14 | 1* | 1* | 1 | 1 | 1* | 1 | 72.7 | 55.4 | 57.1 |
| $\mathcal{D}$-inBC + Aux + OD | 2 | 14 | 1 | 1 | 1 | 1 | 1 | 1 | 79.2 | 65.4 | 60.4 |
| $\mathcal{D}$-inBC + Aux + OD | 2 | 14 | 1 | 1 | 1 | 1 | 1 | 1 | 80.8 | 66.2 | 60.0 |
| $\mathcal{D}$-inBC + Aux + OD | 4 | 4.8 | 0.1 | 0.1 | | | | | 65.4 | 50.0 | 55.0 |
| $\mathcal{D}$-inBC + Aux + OD | 4 | 12 | 1 | 1 | | | | | 76.9 | 63.1 | 55.8 |
| $\mathcal{D}$-RT-2 Style + Oracle | 1 | 1 | | | | | | | 1.2 | 1.9 | 1.2 |
| $\mathcal{D}$-RT-2 Style + Oracle | 2 | 2 | | | | | | | 9.6 | 7.7 | 7.5 |
| $\mathcal{D}$-RT-2 Style + Oracle | 4 | 4 | | | | | | | 21.9 | 17.7 | 15.0 |
| $\mathcal{D}$-RT-2 Style + Oracle | 6 | 6 | | | | | | | 13.5 | 16.2 | 11.2 |
| $\mathcal{D}$-RT-2 Style + Oracle | 8 | 8 | | | | | | | 16.2 | 11.9 | 11.7 |
| $\mathcal{D}$-RT-2 Style (I) + Oracle | 1 | 1 | | | | | | | 1.5 | 1.2 | 1.2 |
| $\mathcal{D}$-RT-2 Style (I) + Oracle | 2 | 2 | | | | | | | 21.9 | 21.5 | 20.8 |
| $\mathcal{D}$-RT-2 Style (I) + Oracle | 4 | 4 | | | | | | | 25.8 | 23.1 | 19.6 |
| $\mathcal{D}$-RT-2 Style (I) + Oracle | 6 | 6 | | | | | | | 17.3 | 18.8 | 15.8 |
| $\mathcal{D}$-RT-2 Style (I) + Oracle | 8 | 8 | | | | | | | 21.9 | 20.0 | 20.4 |
| $\mathcal{D}$-inBC + Oracle | 1 | 1 | | | | | | | 60.4 | 59.2 | 54.6 |
| $\mathcal{D}$-inBC + Oracle | 2 | 2 | | | | | | | 73.5 | 62.7 | 63.3 |
| $\mathcal{D}$-inBC + Oracle | 4 | 4 | | | | | | | 70.8 | 61.2 | 66.2 |
| $\mathcal{D}$-inBC + Oracle | 6 | 6 | | | | | | | 71.9 | 61.5 | 59.6 |
| $\mathcal{D}$-inBC + Oracle | 8 | 8 | | | | | | | 70.4 | 63.8 | 60.0 |
| $\mathcal{D}$-inBC + Aux + Oracle | 2 | 6 | 1 | 1 | | | | | 78.1 | 73.8 | 65.4 |
| $\mathcal{D}$-inBC + Aux + Oracle | 2 | 10 | 1 | 1 | 1 | 1 | | | 78.1 | 73.8 | 70.0 |
| $\mathcal{D}$-inBC + Aux + Oracle | 2 | 12 | 1 | 1 | 1 | 1 | 1 | | 78.5 | 78.1 | 71.2 |
| $\mathcal{D}$-inBC + Aux + Oracle | 2 | 14 | 1* | 1* | 1 | 1 | 1* | 1 | 76.2 | 65.8 | 66.2 |
| $\mathcal{D}$-inBC + Aux + Oracle | 2 | 14 | 1 | 1 | 1 | 1 | 1 | 1 | 82.3 | 78.1 | 69.6 |
| $\mathcal{D}$-inBC + Aux + Oracle | 4 | 4.8 | 0.1 | 0.1 | | | | | 71.9 | 64.6 | 61.7 |
| $\mathcal{D}$-inBC + Aux + Oracle | 4 | 12 | 1 | 1 | | | | | 78.8 | 75.4 | 67.1 |

Table 25: Results on VIMA-80k dataset. **Ep**: number of epoch; **U**: normalized number of model iterations. The following sizes of datasets are relative to the size of *inBC*. **Loc.**: relative size of the localization dataset; **Det.**: relative size of the detection dataset; **Act.**: relative size of the action prediction dataset; **Fut.**: relative size of the future prediction dataset; **Spa.**: relative size of the spatial relationship dataset; **Temp.**: relative size of the temporal relationship dataset.

| Method | Ep | U | Loc. | Det. | Act. | Fut. | Spa. | Temp. | L1 (%) | L2 (%) | L3 (%) |
|---|---|---|---|---|---|---|---|---|---|---|---|
| *RT-2 Style* | 1 | 1 | | | | | | | 53.1 | 46.9 | 42.1 |
| *RT-2 Style* | 2 | 2 | | | | | | | 56.9 | 53.1 | 46.2 |
| *RT-2 Style* | 4 | 4 | | | | | | | 49.6 | 43.8 | 39.6 |
| *RT-2 Style* | 6 | 6 | | | | | | | 51.2 | 45.4 | 38.3 |
| *RT-2 Style* | 8 | 8 | | | | | | | 46.2 | 43.5 | 35.8 |
| *inBC* | 1 | 1 | | | | | | | 68.5 | 61.5 | 58.8 |
| *inBC* | 2 | 2 | | | | | | | 66.2 | 62.3 | 56.2 |
| *inBC* | 4 | 4 | | | | | | | 67.3 | 58.8 | 54.2 |
| *inBC* | 6 | 6 | | | | | | | 65.4 | 60.0 | 53.8 |
| *inBC* | 8 | 8 | | | | | | | 65.8 | 56.5 | 54.2 |
| *inBC* + Aux | 1 | 3 | 1 | 1 | | | | | 62.7 | 60.4 | 50.8 |
| *inBC* + Aux | 1 | 5 | 1 | 1 | 1 | 1 | | | 66.9 | 63.8 | 56.2 |
| *inBC* + Aux | 1 | 6 | 1 | 1 | 1 | 1 | 1 | | 66.2 | 59.6 | 55.8 |
| *inBC* + Aux | 1 | 7 | 1 | 1 | 1 | 1 | 1 | 1 | 68.1 | 60.8 | 54.6 |
| *inBC* + Aux | 2 | 6 | 1 | 1 | | | | | 65.8 | 64.6 | 56.7 |
| *inBC* + Aux | 2 | 10 | 1 | 1 | 1 | 1 | | | 69.2 | 66.9 | 55.8 |
| *inBC* + Aux | 2 | 12 | 1 | 1 | 1 | 1 | 1 | | 65.4 | 60.0 | 50.0 |
| *inBC* + Aux | 2 | 14 | 1 | 1 | 1 | 1 | 1 | 1 | 68.1 | 65.0 | 51.7 |
| *inBC* + Aux | 4 | 12 | 1 | 1 | | | | | 65.8 | 62.7 | 49.6 |
| *inBC* + Aux | 4 | 20 | 1 | 1 | 1 | 1 | | | 65.8 | 63.1 | 52.5 |
| *inBC* + Aux | 4 | 24 | 1 | 1 | 1 | 1 | 1 | | 63.8 | 59.6 | 48.8 |
| *inBC* + Aux | 4 | 28 | 1 | 1 | 1 | 1 | 1 | 1 | 68.1 | 65.0 | 52.9 |
| *inBC* + Aux | 6 | 18 | 1 | 1 | | | | | 67.7 | 61.5 | 51.2 |
| *inBC* + Aux | 6 | 30 | 1 | 1 | 1 | 1 | | | 67.3 | 60.4 | 49.2 |
| *inBC* + Aux | 6 | 36 | 1 | 1 | 1 | 1 | 1 | | 61.9 | 60.4 | 47.1 |
| *inBC* + Aux | 6 | 42 | 1 | 1 | 1 | 1 | 1 | 1 | 65.4 | 63.8 | 52.9 |
| *inBC* + Aux | 8 | 24 | 1 | 1 | | | | | 67.3 | 62.7 | 52.1 |
| *inBC* + Aux | 8 | 40 | 1 | 1 | 1 | 1 | | | 66.9 | 61.2 | 51.7 |
| *inBC* + Aux | 8 | 48 | 1 | 1 | 1 | 1 | 1 | | 66.2 | 58.1 | 50.0 |
| *inBC* + Aux | 8 | 56 | 1 | 1 | 1 | 1 | 1 | 1 | 66.9 | 63.8 | 50.8 |
| $\mathcal{D}$-*inBC* + Aux | 1 | 3 | 1 | 1 | | | | | 66.2 | 63.5 | 57.5 |
| $\mathcal{D}$-*inBC* + Aux | 1 | 5 | 1 | 1 | 1 | 1 | | | 66.2 | 60.8 | 57.1 |
| $\mathcal{D}$-*inBC* + Aux | 1 | 6 | 1 | 1 | 1 | 1 | 1 | | 65.4 | 60.4 | 52.5 |
| $\mathcal{D}$-*inBC* + Aux | 1 | 7 | 1 | 1 | 1 | 1 | 1 | 1 | 63.8 | 61.9 | 54.2 |
| $\mathcal{D}$-*inBC* + Aux | 2 | 6 | 1 | 1 | | | | | 68.5 | 61.9 | 54.6 |
| $\mathcal{D}$-*inBC* + Aux | 2 | 10 | 1 | 1 | 1 | 1 | | | 70.0 | 65.4 | 56.2 |
| $\mathcal{D}$-*inBC* + Aux | 2 | 12 | 1 | 1 | 1 | 1 | 1 | | 69.6 | 62.7 | 56.2 |
| $\mathcal{D}$-*inBC* + Aux | 2 | 14 | 1 | 1 | 1 | 1 | 1 | 1 | 70.4 | 63.5 | 56.7 |
| $\mathcal{D}$-*inBC* + Aux | 4 | 12 | 1 | 1 | | | | | 67.7 | 65.4 | 60.0 |
| $\mathcal{D}$-*inBC* + Aux | 4 | 20 | 1 | 1 | 1 | 1 | | | 68.1 | 66.5 | 57.5 |
| $\mathcal{D}$-*inBC* + Aux | 4 | 24 | 1 | 1 | 1 | 1 | 1 | | 66.5 | 61.2 | 52.9 |
| $\mathcal{D}$-*inBC* + Aux | 4 | 28 | 1 | 1 | 1 | 1 | 1 | 1 | 69.6 | 58.5 | 52.9 |
| $\mathcal{D}$-*inBC* + Aux | 6 | 18 | 1 | 1 | | | | | 69.2 | 63.8 | 57.9 |
| $\mathcal{D}$-*inBC* + Aux | 6 | 30 | 1 | 1 | 1 | 1 | | | 70.0 | 68.1 | 58.8 |
| $\mathcal{D}$-*inBC* + Aux | 6 | 36 | 1 | 1 | 1 | 1 | 1 | | 66.5 | 62.3 | 47.5 |
| $\mathcal{D}$-*inBC* + Aux | 6 | 42 | 1 | 1 | 1 | 1 | 1 | 1 | 70.0 | 61.9 | 54.2 |
| $\mathcal{D}$-*inBC* + Aux | 8 | 24 | 1 | 1 | | | | | 67.7 | 61.5 | 50.8 |
| $\mathcal{D}$-*inBC* + Aux | 8 | 40 | 1 | 1 | 1 | 1 | | | 69.6 | 66.5 | 57.1 |
| $\mathcal{D}$-*inBC* + Aux | 8 | 48 | 1 | 1 | 1 | 1 | 1 | | 71.2 | 69.2 | 51.7 |
| $\mathcal{D}$-*inBC* + Aux | 8 | 56 | 1 | 1 | 1 | 1 | 1 | 1 | 68.1 | 67.7 | 57.9 |

Table 26: Results on VIMA-80k dataset with object detector or Oracle. **Ep**: number of epoch; **U**: normalized number of model iterations. The following sizes of datasets are relative to the size of *inBC*. **Loc.**: relative size of the localization dataset; **Det.**: relative size of the detection dataset; **Act.**: relative size of the action prediction dataset; **Fut.**: relative size of the future prediction dataset; **Spa.**: relative size of the spatial relationship dataset; **Temp.**: relative size of the temporal relationship dataset.

| Method | Ep | U | Loc. | Det. | Act. | Fut. | Spa. | Temp. | L1 (%) | L2 (%) | L3 (%) |
|---|---|---|---|---|---|---|---|---|---|---|---|
| *D-RT-2 Style* + OD | 1 | 1 | | | | | | | 49.2 | 43.1 | 36.2 |
| *D-RT-2 Style* + OD | 2 | 2 | | | | | | | 52.3 | 50.0 | 45.8 |
| *D-RT-2 Style* + OD | 4 | 4 | | | | | | | 52.3 | 46.5 | 37.9 |
| *D-RT-2 Style* + OD | 6 | 6 | | | | | | | 47.3 | 47.3 | 36.2 |
| *D-RT-2 Style* + OD | 8 | 8 | | | | | | | 44.2 | 42.3 | 36.7 |
| *D-inBC* + OD | 1 | 1 | | | | | | | 81.9 | 75.0 | 67.5 |
| *D-inBC* + OD | 2 | 2 | | | | | | | 78.8 | 75.0 | 69.2 |
| *D-inBC* + OD | 4 | 4 | | | | | | | 81.2 | 75.4 | 68.3 |
| *D-inBC* + OD | 6 | 6 | | | | | | | 81.9 | 75.8 | 67.1 |
| *D-inBC* + OD | 8 | 8 | | | | | | | 83.8 | 73.8 | 69.2 |
| *D-inBC* + Aux + OD | 1 | 3 | 1 | 1 | | | | | 85.0 | 79.2 | 65.8 |
| *D-inBC* + Aux + OD | 1 | 5 | 1 | 1 | 1 | 1 | | | 85.0 | 77.7 | 68.3 |
| *D-inBC* + Aux + OD | 1 | 6 | 1 | 1 | 1 | 1 | 1 | | 79.6 | 73.5 | 64.2 |
| *D-inBC* + Aux + OD | 1 | 7 | 1 | 1 | 1 | 1 | 1 | 1 | 84.2 | 74.6 | 66.2 |
| *D-inBC* + Aux + OD | 2 | 6 | 1 | 1 | | | | | 81.5 | 75.8 | 64.2 |
| *D-inBC* + Aux + OD | 2 | 10 | 1 | 1 | 1 | 1 | | | 86.2 | 83.1 | 69.6 |
| *D-inBC* + Aux + OD | 2 | 12 | 1 | 1 | 1 | 1 | 1 | | 80.0 | 73.1 | 62.5 |
| *D-inBC* + Aux + OD | 2 | 14 | 1 | 1 | 1 | 1 | 1 | 1 | 85.8 | 77.3 | 66.7 |
| *D-inBC* + Aux + OD | 4 | 12 | 1 | 1 | | | | | 78.5 | 73.8 | 68.8 |
| *D-inBC* + Aux + OD | 4 | 20 | 1 | 1 | 1 | 1 | | | 82.3 | 77.3 | 71.7 |
| *D-inBC* + Aux + OD | 4 | 24 | 1 | 1 | 1 | 1 | 1 | | 82.3 | 71.9 | 63.7 |
| *D-inBC* + Aux + OD | 4 | 28 | 1 | 1 | 1 | 1 | 1 | 1 | 83.5 | 71.5 | 62.5 |
| *D-inBC* + Aux + OD | 6 | 18 | 1 | 1 | | | | | 80.0 | 75.0 | 67.1 |
| *D-inBC* + Aux + OD | 6 | 30 | 1 | 1 | 1 | 1 | | | 86.2 | 81.5 | 69.6 |
| *D-inBC* + Aux + OD | 6 | 36 | 1 | 1 | 1 | 1 | 1 | | 79.6 | 74.6 | 64.6 |
| *D-inBC* + Aux + OD | 6 | 42 | 1 | 1 | 1 | 1 | 1 | 1 | 85.0 | 77.3 | 62.1 |
| *D-inBC* + Aux + OD | 8 | 24 | 1 | 1 | | | | | 78.5 | 73.1 | 68.3 |
| *D-inBC* + Aux + OD | 8 | 40 | 1 | 1 | 1 | 1 | | | 86.5 | 82.7 | 70.8 |
| *D-inBC* + Aux + OD | 8 | 48 | 1 | 1 | 1 | 1 | 1 | | 82.3 | 78.1 | 65.4 |
| *D-inBC* + Aux + OD | 8 | 56 | 1 | 1 | 1 | 1 | 1 | 1 | 82.7 | 76.5 | 68.3 |
| *D-RT-2 Style* + Oracle | 1 | 1 | | | | | | | 71.2 | 66.5 | 61.3 |
| *D-RT-2 Style* + Oracle | 2 | 2 | | | | | | | 74.2 | 70.4 | 69.2 |
| *D-RT-2 Style* + Oracle | 4 | 4 | | | | | | | 73.1 | 70.4 | 63.3 |
| *D-RT-2 Style* + Oracle | 6 | 6 | | | | | | | 67.7 | 64.2 | 60.4 |
| *D-RT-2 Style* + Oracle | 8 | 8 | | | | | | | 69.2 | 61.5 | 55.4 |
| *D-RT-2 Style (I)* + Oracle | 1 | 1 | | | | | | | 57.7 | 54.6 | 57.1 |
| *D-RT-2 Style (I)* + Oracle | 2 | 2 | | | | | | | 55.8 | 54.6 | 55.4 |
| *D-RT-2 Style (I)* + Oracle | 4 | 4 | | | | | | | 58.1 | 54.2 | 55.4 |
| *D-RT-2 Style (I)* + Oracle | 6 | 6 | | | | | | | 53.5 | 49.6 | 47.9 |
| *D-RT-2 Style (I)* + Oracle | 8 | 8 | | | | | | | 53.8 | 50.8 | 48.8 |
| *D-inBC* + Oracle | 1 | 1 | | | | | | | 84.6 | 84.2 | 80.8 |
| *D-inBC* + Oracle | 2 | 2 | | | | | | | 84.2 | 82.3 | 77.5 |
| *D-inBC* + Oracle | 4 | 4 | | | | | | | 85.4 | 83.8 | 80.0 |
| *D-inBC* + Oracle | 6 | 6 | | | | | | | 86.9 | 85.8 | 80.8 |
| *D-inBC* + Oracle | 8 | 8 | | | | | | | 87.3 | 85.4 | 82.1 |
| *D-inBC* + Aux + Oracle | 1 | 3 | 1 | 1 | | | | | 88.8 | 85.0 | 79.2 |
| *D-inBC* + Aux + Oracle | 1 | 5 | 1 | 1 | 1 | 1 | | | 86.2 | 84.6 | 79.2 |
| *D-inBC* + Aux + Oracle | 1 | 6 | 1 | 1 | 1 | 1 | 1 | | 82.3 | 77.3 | 69.6 |
| *D-inBC* + Aux + Oracle | 1 | 7 | 1 | 1 | 1 | 1 | 1 | 1 | 86.2 | 83.5 | 77.5 |
| *D-inBC* + Aux + Oracle | 2 | 6 | 1 | 1 | | | | | 88.5 | 84.2 | 77.5 |
| *D-inBC* + Aux + Oracle | 2 | 10 | 1 | 1 | 1 | 1 | | | 91.5 | 88.1 | 80.4 |
| *D-inBC* + Aux + Oracle | 2 | 12 | 1 | 1 | 1 | 1 | 1 | | 81.2 | 80.4 | 67.5 |
| *D-inBC* + Aux + Oracle | 2 | 14 | 1 | 1 | 1 | 1 | 1 | 1 | 88.5 | 84.6 | 78.3 |
| *D-inBC* + Aux + Oracle | 4 | 12 | 1 | 1 | | | | | 85.0 | 87.3 | 78.8 |
| *D-inBC* + Aux + Oracle | 4 | 20 | 1 | 1 | 1 | 1 | | | 88.8 | 84.6 | 79.6 |
| *D-inBC* + Aux + Oracle | 4 | 24 | 1 | 1 | 1 | 1 | 1 | | 81.5 | 77.3 | 65.8 |
| *D-inBC* + Aux + Oracle | 4 | 28 | 1 | 1 | 1 | 1 | 1 | 1 | 84.2 | 80.0 | 71.7 |
| *D-inBC* + Aux + Oracle | 6 | 18 | 1 | 1 | | | | | 84.6 | 85.4 | 76.7 |
| *D-inBC* + Aux + Oracle | 6 | 30 | 1 | 1 | 1 | 1 | | | 87.7 | 87.3 | 78.3 |
| *D-inBC* + Aux + Oracle | 6 | 36 | 1 | 1 | 1 | 1 | 1 | | 81.2 | 80.8 | 70.8 |
| *D-inBC* + Aux + Oracle | 6 | 42 | 1 | 1 | 1 | 1 | 1 | 1 | 88.8 | 81.2 | 70.0 |
| *D-inBC* + Aux + Oracle | 8 | 24 | 1 | 1 | | | | | 86.9 | 81.5 | 75.8 |
| *D-inBC* + Aux + Oracle | 8 | 40 | 1 | 1 | 1 | 1 | | | 90.0 | 88.1 | 79.2 |
| *D-inBC* + Aux + Oracle | 8 | 48 | 1 | 1 | 1 | 1 | 1 | | 82.3 | 83.1 | 70.4 |
| *D-inBC* + Aux + Oracle | 8 | 56 | 1 | 1 | 1 | 1 | 1 | 1 | 83.8 | 88.1 | 78.8 |

Table 27: Results on full VIMA dataset with Oracle. **Ep**: number of epoch; **U**: normalized number of model iterations. The following sizes of datasets are relative to the size of *inBC*. **Loc.**: relative size of the localization dataset; **Det.**: relative size of the detection dataset; **Act.**: relative size of the action prediction dataset; **Fut.**: relative size of the future prediction dataset; **Spa.**: relative size of the spatial relationship dataset; **Temp.**: relative size of the temporal relationship dataset.

| Method | Ep | U | Loc. | Det. | Act. | Fut. | Spa. | Temp. | L1 (%) | L2 (%) | L3 (%) |
|---|---|---|---|---|---|---|---|---|---|---|---|
| *RT-2 Style* | 1 | 1 | | | | | | | 61.9 | 55.8 | 47.1 |
| *RT-2 Style* | 2 | 2 | | | | | | | 65.0 | 59.6 | 42.5 |
| *RT-2 Style* | 3 | 3 | | | | | | | 63.1 | 57.7 | 44.6 |
| *RT-2 Style* | 4 | 4 | | | | | | | 64.2 | 60.0 | 44.2 |
| *D-RT-2 Style* + Oracle | 1 | 1 | | | | | | | 82.3 | 80.8 | 74.6 |
| *D-RT-2 Style* + Oracle | 2 | 2 | | | | | | | 86.2 | 88.1 | 75.8 |
| *D-RT-2 Style* + Oracle | 3 | 3 | | | | | | | 87.7 | 83.5 | 76.2 |
| *D-RT-2 Style* + Oracle | 4 | 4 | | | | | | | 87.3 | 85.4 | 74.6 |
| *D-RT-2 Style (I)* + Oracle | 1 | 1 | | | | | | | 68.8 | 68.1 | 59.6 |
| *D-RT-2 Style (I)* + Oracle | 2 | 2 | | | | | | | 68.5 | 66.2 | 58.3 |
| *D-RT-2 Style (I)* + Oracle | 3 | 3 | | | | | | | 70.0 | 70.0 | 59.2 |
| *D-RT-2 Style (I)* + Oracle | 4 | 4 | | | | | | | 69.6 | 69.2 | 58.3 |

Table 28: Prompt candidates for action generation during inference.

'Could you write down what needs to be done to complete the task on this scene?'
'List out the actions needed to accomplish the task in this scene.'
'What actions are necessary to perform the task on this scene?'
'Can you describe what needs to be done on this scene to complete the task?'
'What steps are required to perform the task shown in this scene?'
'List the actions needed to perform the task given below.'
'On the following scene, could you list what actions are required to perform the task?'
'Describe what actions are needed on this scene to complete the task.'
'What do you need to do on this scene to accomplish the task?'
'List the actions required to perform the task given on this scene.'
'Could you please describe the steps needed to perform the task on this scene?'
'Write down the actions required to perform the task on this scene.'
'Please write down the actions required to perform the task shown below.'
'Can you explain what needs to be done to perform the task in this scene?'
'Describe the actions required to complete the task on this scene.'

Table 29: Prompt candidates for object localization. {object} will be replaced with the actual object name in the dataset.

'Where is {object} located in the image? Please use the format (x, y),{w, h} where x and y represent the center coordinates of the bounding box, and w and h are the width and height. The coordinates start from the top left corner and are normalized to a scale of 0 to 1.'

'Can you provide the location of {object} in the image? Format it as (x, y),{w, h}, with x and y as the center coordinates of the bounding box and w and h as the width and height. The coordinates should begin at the top left corner and be normalized from 0 to 1.'

'What are the coordinates of {object} in the image? Use the format (x, y),{w, h}, where x and y are the center of the bounding box, and w and h represent the width and height. Coordinates should start at the top left corner and be normalized to a range of 0 to 1.'

'Please specify the location of {object} in the image. List it in the format (x, y),{w, h}, where x and y denote the bounding box center coordinates, and w and h are the width and height. The coordinates begin from the top left corner and should be normalized to 0 to 1.'

'What is the position of {object} within the image? Use the format (x, y),{w, h} to describe it, with x and y as the center coordinates of the bounding box, and w and h as the width and height. The coordinates start at the top left corner and are normalized to a scale of 0 to 1.'

'Describe the location of {object} in the image using the format (x, y),{w, h}. In this format, x and y denote the center coordinates of the bounding box, while w and h represent its width and height. Coordinates should be normalized from the top left corner, ranging from 0 to 1.'

'Can you detail the location of {object} in the image? Format it as (x, y),{w, h}, where x and y indicate the bounding box center, and w and h represent the width and height. The coordinates should be normalized to a scale of 0 to 1 starting from the top left corner.'

'Provide the location of {object} in the image using the format (x, y),{w, h}. Here, x and y are the center coordinates of the bounding box, and w and h are the width and height. The coordinates begin at the top left corner and are normalized from 0 to 1.'

'Where is {object} positioned in the image? Use the format (x, y),{w, h}, where x and y denote the center coordinates of the bounding box, and w and h are the width and height. The coordinates should be normalized to a range of 0 to 1 starting from the top left corner.'

'Specify the location of {object} in the image in the format (x, y),{w, h}. In this format, x and y represent the bounding box center, and w and h are the width and height. The coordinates should start from the top left corner and be normalized between 0 and 1.'

'What is the exact position of {object} in the image? Format the coordinates as (x, y),{w, h}, where x and y are the center of the bounding box and w and h denote its width and height. The coordinates start from the top left corner and are normalized to a scale of 0 to 1.'

'Describe where {object} is located in the image using the format (x, y),{w, h}. Here, x and y indicate the bounding box center coordinates, and w and h specify its width and height. The coordinates should be normalized starting from the top left corner, within the range of 0 to 1.'

'Could you tell me the location of {object} in the image? Use the format (x, y),{w, h}, where x and y denote the center of the bounding box and w and h are the width and height. Coordinates start at the top left corner and should be normalized between 0 and 1.'

'Provide the coordinates of {object} in the image in the format (x, y),{w, h}. Here, x and y are the center of the bounding box, while w and h represent its width and height. The coordinates should start from the top left corner and be normalized to 0 to 1.'

'How is the {object} located in the image? List its coordinates using the format (x, y),{w, h}, where x and y are the center coordinates of the bounding box, and w and h indicate its width and height. The coordinates begin at the top left corner and are normalized to a range of 0 to 1.'

Table 30: Prompt candidates for object detection.

'Identify and describe each object in the image. For each object, list it in the format (x, y),{w, h}, where x and y represent the coordinates of the bounding box center, and w and h represent the width and height of the bounding box. The image coordinates should start from the top left corner and be normalized between 0 and 1.'

'Catalog all the objects present in the image. For every object, use the format (x, y),{w, h}, with x and y indicating the center of the object's bounding box coordinates, and w and h specifying the width and height. The coordinates are normalized from the top left corner, ranging from 0 to 1.'

'List each object in the image and describe it. Use the format (x, y),{w, h} for each object, where x and y denote the center coordinates of the bounding box, and w and h are the width and height of the bounding box. The coordinates should start from the top left corner and be normalized to a scale of 0 to 1.'

'Provide descriptions for all objects within the image. Each object should be listed using the format (x, y),{w, h}, where x and y are the coordinates of the bounding box center, and w and h are the width and height. The coordinates should be normalized, starting from the top left corner, within a range of 0 to 1.'

'Enumerate and describe every object found in the image. For each object, utilize the format (x, y),{w, h}, where x, y are the bounding box center coordinates and w, h are the dimensions (width and height) of the bounding box. The coordinates begin at the top left corner and are normalized between 0 and 1.'

'Detail all the objects within the image, listing each one using the format (x, y),{w, h}. Here, x and y represent the coordinates of the bounding box center, while w and h indicate the width and height. The coordinates start from the top left corner and are normalized to the range of 0 to 1.'

'Document each object present in the image. For each object, use the format (x, y),{w, h}, where x and y are the coordinates of the center of the bounding box, and w and h are the width and height. The coordinates should be normalized, starting from the top left corner, and range from 0 to 1.'

'For each object in the image, provide a description using the format (x, y),{w, h}. Here, x and y denote the coordinates of the bounding box center, and w and h represent the width and height of the bounding box. The coordinates are normalized to a scale of 0 to 1, starting from the top left corner.'

'Describe all the objects seen in the image, and list them using the format (x, y),{w, h}. The x and y values are the coordinates for the center of the bounding box, while w and h represent its width and height. The coordinates should be normalized from the top left corner, within a range of 0 to 1.'

'Identify and list each object found in the image. For each one, use the format (x, y),{w, h}. In this format, x and y are the coordinates for the bounding box center, and w and h are the width and height. The coordinates are to be normalized starting from the top left corner, ranging from 0 to 1.'

'List and describe each object in the image using the format (x, y),{w, h}. Here, x and y correspond to the coordinates of the bounding box center, and w and h specify the width and height of the bounding box. The coordinates should start from the top left corner and be normalized to the range of 0 to 1.'

'Provide a description for each object in the image, formatted as (x, y),{w, h}. The x and y values indicate the center coordinates of the bounding box, while w and h represent the width and height. The coordinates start from the top left corner and are normalized between 0 and 1.'

Table 31: (Continued) Prompt candidates for object detection.

'Catalog each object within the image, using the format (x, y),{w, h} for each one. In this format, x and y are the coordinates for the center of the bounding box, and w and h are the width and height. The coordinates should be normalized, beginning at the top left corner and ranging from 0 to 1.'
'Enumerate all the objects in the image, providing descriptions for each using the format (x, y),{w, h}. The x and y values represent the center coordinates of the bounding box, while w and h indicate its width and height. The coordinates are normalized starting from the top left corner, within a range of 0 to 1.'
'Describe each object in the image, listing them in the format (x, y),{w, h}. Here, x and y denote the center coordinates of the bounding box, and w and h specify the width and height. The coordinates should be normalized from the top left corner, ranging from 0 to 1.'

Table 32: Prompt candidates for action prediction. {scene} will be replaced with a list of objects in the scene.

'Could you detail the steps needed to transform the scene shown in the image into the second scene? The second scene is provided as a collection of object bounding boxes {scene}. The format for these bounding boxes is (x, y), {w, h}, where x and y represent the center coordinates, and w and h are the width and height. The coordinates should be normalized to a scale of 0 to 1, starting from the top left corner.'
'Can you describe what actions are required to rearrange the scene shown in the image to match the second scene? The second scene is given as a set of object bounding boxes {scene}. These bounding boxes follow the format (x, y), {w, h}, where x and y indicate the center coordinates, and w and h represent the width and height. The coordinates should start from the top left corner and be normalized to a scale of 0 to 1.'
'Could you list the steps necessary to modify the scene shown in the image to the second scene? The second scene is described as a collection of object bounding boxes {scene}. The bounding box format is (x, y), {w, h}, with x and y denoting the center coordinates, and w and h representing the width and height. The coordinates are normalized to a scale of 0 to 1, starting from the top left corner.'
'Can you explain what needs to be done to adjust the scene shown in the image to resemble the second scene? The second scene {scene} consists of object bounding boxes provided in the format (x, y), {w, h}. Here, x and y represent the center coordinates, and w and h are the width and height. The coordinates should start from the top left corner and be normalized to a scale of 0 to 1.'
'Could you outline the necessary actions to arrange the scene shown in the image into the second scene? The second scene is defined by a collection of object bounding boxes {scene}. These bounding boxes follow the format (x, y), {w, h}, where x and y denote the center coordinates, and w and h are the width and height. The coordinates start from the top left corner and should be normalized to a scale of 0 to 1.'
'Can you specify what needs to be done to convert the scene shown in the image into the second scene? The second scene is provided as a series of object bounding boxes {scene}. The format for these bounding boxes is (x, y), {w, h}, with x and y representing the center coordinates, and w and h indicating the width and height. Coordinates should be normalized from the top left corner to a scale of 0 to 1.'
'Could you describe the steps required to change the scene shown in the image to the second scene? The second scene is depicted as a collection of object bounding boxes {scene}. The bounding box format is (x, y), {w, h}, where x and y denote the center coordinates, and w and h represent the width and height. The coordinates are normalized to a scale of 0 to 1 starting from the top left corner.'

Table 33: (Continued) Prompt candidates for action prediction. {scene} will be replaced with a list of objects in the scene.

'Can you list the actions necessary to transform the scene shown in the image into the second scene? The second scene is described using object bounding boxes {scene}. The format of these bounding boxes is (x, y), {w, h}, where x and y are the center coordinates, and w and h represent the width and height. Coordinates should be normalized to a scale of 0 to 1 starting from the top left corner.'

'Could you explain the process to arrange the scene shown in the image to match the second scene? The second scene is provided as a collection of object bounding boxes {scene}. These bounding boxes are formatted as (x, y), {w, h}, where x and y represent the center coordinates, and w and h are the width and height. The coordinates should start from the top left corner and be normalized to a scale of 0 to 1.'

'Can you detail what needs to be done to rearrange the scene shown in the image to the second scene? The second scene is given as a series of object bounding boxes {scene}. The bounding box format is (x, y), {w, h}, where x and y denote the center coordinates, and w and h represent the width and height. Coordinates should be normalized to a scale of 0 to 1 starting from the top left corner.'

'Could you specify the steps needed to modify the scene shown in the image to resemble the second scene? The second scene is described as a set of object bounding boxes {scene}. These bounding boxes follow the format (x, y), {w, h}, where x and y represent the center coordinates, and w and h indicate the width and height. The coordinates start from the top left corner and should be normalized to a scale of 0 to 1.'

'Can you outline the necessary actions to change the scene shown in the image into the second scene? The second scene {scene} consists of object bounding boxes provided in the format (x, y), {w, h}, where x and y denote the center coordinates, and w and h represent the width and height. Coordinates should be normalized to a scale of 0 to 1 starting from the top left corner.'

'Could you describe the steps to adjust the scene shown in the image to the second scene? The second scene is given as a collection of object bounding boxes {scene}. The format for these bounding boxes is (x, y), {w, h}, where x and y represent the center coordinates, and w and h are the width and height. The coordinates should start from the top left corner and be normalized to a scale of 0 to 1.'

'Can you explain what needs to be done to transform the scene shown in the image into the second scene? The second scene is depicted using object bounding boxes {scene}. The bounding box format is (x, y), {w, h}, with x and y representing the center coordinates, and w and h indicating the width and height. The coordinates start from the top left corner and are normalized to a scale of 0 to 1.'

'Could you detail the steps necessary to convert the scene shown in the image to the second scene? The second scene is described as a set of object bounding boxes {scene}. These bounding boxes follow the format (x, y), {w, h}, where x and y represent the center coordinates, and w and h denote the width and height. The coordinates should be normalized to a scale of 0 to 1 starting from the top left corner.'

Table 34: Prompt candidates for future prediction. {pick and place} will be replaced with the action text.

'The image shows a scene with multiple objects. Now you {pick and place}, what will the scene look like? List the object bounding boxes. The bounding box format is (x, y), {w, h}, where x and y represent the center coordinates of the bounding box, and w and h are its width and height. The coordinates should start from the top left corner and be normalized to a scale of 0 to 1.'

'An image depicts a scene containing multiple objects. Now you {pick and place}, what will the scene look like? Write the list of object bounding boxes. The bounding boxes should be formatted as (x, y), {w, h}, where x and y denote the center coordinates, and w and h are the width and height. The coordinates start from the top left corner and are normalized to a scale of 0 to 1.'

'The image presents a scene with several objects. Now you {pick and place}, what will the scene look like? List the object bounding boxes. The format for these bounding boxes is (x, y), {w, h}, where x and y represent the center coordinates, and w and h are the width and height. Coordinates should start from the top left corner and be normalized to a scale of 0 to 1.'

'Displayed in the image is a scene containing multiple objects. Now you {pick and place}, what will the scene look like? Write down the list of object bounding boxes. These bounding boxes follow the format (x, y), {w, h}, with x and y as the center coordinates, and w and h as the width and height. The coordinates should be normalized starting from the top left corner to a scale of 0 to 1.'

'The image illustrates a scene with multiple objects. Now you {pick and place}, what will the scene look like? Write the list of object bounding boxes. The bounding boxes are formatted as (x, y), {w, h}, where x and y denote the center coordinates, and w and h represent the width and height. Coordinates should start from the top left corner and be normalized to a scale of 0 to 1.'

'The image depicts a scene with several objects. Now you {pick and place}, what will the scene look like? List the object bounding boxes. The bounding box format is (x, y), {w, h}, where x and y represent the center coordinates, and w and h denote the width and height. The coordinates should be normalized to a scale of 0 to 1 starting from the top left corner.'

'In the image, there is a scene with multiple objects. Now you {pick and place}, what will the scene look like? Write the list of object bounding boxes. The format of these bounding boxes is (x, y), {w, h}, where x and y indicate the center coordinates, and w and h represent the width and height. The coordinates start from the top left corner and are normalized to a scale of 0 to 1.'

'An image shows a scene with various objects. Now you {pick and place}, what will the scene look like? Write down the list of object bounding boxes. The bounding boxes follow the format (x, y), {w, h}, where x and y denote the center coordinates, and w and h are the width and height. The coordinates should start from the top left corner and be normalized to a scale of 0 to 1.'

'The image presents a scene containing several objects. Now you {pick and place}, what will the scene look like? List the object bounding boxes. The bounding box format is (x, y), {w, h}, where x and y represent the center coordinates, and w and h are the width and height. Coordinates should start from the top left corner and be normalized to a scale of 0 to 1.'

'The image displays a scene with multiple objects. Now you {pick and place}, what will the scene look like? Write the list of object bounding boxes. The bounding boxes should be in the format (x, y), {w, h}, where x and y denote the center coordinates, and w and h represent the width and height. The coordinates should start from the top left corner and be normalized to a scale of 0 to 1.'

'An image illustrates a scene with multiple objects. Now you {pick and place}, what will the scene look like? Write down the list of object bounding boxes. These bounding boxes are formatted as (x, y), {w, h}, where x and y represent the center coordinates, and w and h denote the width and height. Coordinates should be normalized to a scale of 0 to 1 starting from the top left corner.'

Table 35: (Continued) Prompt candidates for future prediction. {pick and place} will be replaced with the action text.

'The image shows a scene with various objects. Now you {pick and place}, what will the scene look like? List the object bounding boxes. The format for these bounding boxes is (x, y), {w, h}, with x and y representing the center coordinates, and w and h as the width and height. Coordinates should start from the top left corner and be normalized to a scale of 0 to 1.'

'Displayed in the image is a scene containing multiple objects. Now you {pick and place}, what will the scene look like? Write the list of object bounding boxes. The bounding box format is (x, y), {w, h}, where x and y denote the center coordinates, and w and h are the width and height. The coordinates start from the top left corner and are normalized to a scale of 0 to 1.'

'The image illustrates a scene with various objects. Now you {pick and place}, what will the scene look like? List the object bounding boxes. The bounding boxes are formatted as (x, y), {w, h}, where x and y indicate the center coordinates, and w and h represent the width and height. Coordinates should be normalized from the top left corner to a scale of 0 to 1.'

'An image depicts a scene with multiple objects. Now you {pick and place}, what will the scene look like? Write the list of object bounding boxes. The bounding box format is (x, y), {w, h}, where x and y represent the center coordinates, and w and h denote the width and height. The coordinates should start from the top left corner and be normalized to a scale of 0 to 1.'

Table 36: Prompt candidates for spatial relationship. {ego_obj} and {ref_obj} will be replaced with object names and {example} will be replaced with a random spatial relationship from the same image.

"Can you describe the relative spatial locations of {ego_obj} compared to {ref_obj} in this image? Use relative location words like left, right, above, below, etc. Also, find the 2D center distance and the Euclidean center distance between them. Your output must follow this format: {example}. The coordinates are image coordinates normalized to a scale of 0 to 1 starting from the top left corner.'

'Could you describe the relative spatial positions of {ego_obj} in comparison to {ref_obj} in this image? Use terms like left, right, above, below, etc. Also, calculate the 2D center distance and the Euclidean center distance between them. Your output should be formatted as follows: {example}. The coordinates are image coordinates normalized to a scale of 0 to 1 starting from the top left corner.'

'Please describe the relative spatial locations of {ego_obj} compared to {ref_obj} in this image. Use words like left, right, above, below, etc. Additionally, find the 2D center distance and the Euclidean center distance between them. Your output must be in this format: {example}. The coordinates are image coordinates normalized to a scale of 0 to 1 starting from the top left corner.'

'Can you explain the relative spatial positions of {ego_obj} compared to {ref_obj} in this image? Use terms such as left, right, above, below, etc. Also, determine the 2D center distance and the Euclidean center distance between them. Your output should match this format: {example}. The coordinates are image coordinates normalized to a scale of 0 to 1 starting from the top left corner.'

'Describe the relative spatial locations of {ego_obj} compared to {ref_obj} in this image using words like left, right, above, below, etc. Also, calculate the 2D center distance and the Euclidean center distance between them. Your output must follow this format: {example}. The coordinates are image coordinates normalized to a scale of 0 to 1 starting from the top left corner.'

Table 37: (Continued) Prompt candidates for spatial relationship. {ego_obj} and {ref_obj} will be replaced with object names and {example} will be replaced with a random spatial relationship from the same image.

'Could you describe the spatial relationship between {ego_obj} and {ref_obj} in this image using relative location words like left, right, above, below, etc.? Also, find the 2D center distance and the Euclidean center distance between them. Your output should be formatted as follows: {example}. The coordinates are image coordinates normalized to a scale of 0 to 1 starting from the top left corner.'

'Can you detail the relative spatial positions of {ego_obj} compared to {ref_obj} in this image? Use words like left, right, above, below, etc. Also, determine the 2D center distance and the Euclidean center distance between them. Your output must be in this format: {example}. The coordinates are image coordinates normalized to a scale of 0 to 1 starting from the top left corner.'

'Could you explain the spatial relationship between {ego_obj} and {ref_obj} in this image using terms such as left, right, above, below, etc.? Also, calculate the 2D center distance and the Euclidean center distance between them. Your output should match this format: {example}. The coordinates are image coordinates normalized to a scale of 0 to 1 starting from the top left corner.'

'Describe the relative spatial positions of {ego_obj} compared to {ref_obj} in this image. Use relative location words like left, right, above, below, etc. Also, find the 2D center distance and the Euclidean center distance between them. Your output must follow this format: {example}. The coordinates are image coordinates normalized to a scale of 0 to 1 starting from the top left corner.'

'Can you describe how {ego_obj} is positioned relative to {ref_obj} in this image using words such as left, right, above, below, etc.? Also, find the 2D center distance and the Euclidean center distance between them. Your output should be in this format: {example}. The coordinates are image coordinates normalized to a scale of 0 to 1 starting from the top left corner.'

'Could you detail the relative positions of {ego_obj} compared to {ref_obj} in this image using terms like left, right, above, below, etc.? Also, calculate the 2D center distance and the Euclidean center distance between them. Your output must be formatted as follows: {example}. The coordinates are image coordinates normalized to a scale of 0 to 1 starting from the top left corner.'

'Please describe the spatial relationship of {ego_obj} in comparison to {ref_obj} in this image using relative location terms such as left, right, above, below, etc. Additionally, find the 2D center distance and the Euclidean center distance between them. Your output should match this format: {example}. The coordinates are image coordinates normalized to a scale of 0 to 1 starting from the top left corner.'

'Can you describe the relative spatial locations of {ego_obj} compared to {ref_obj} in this image? Use relative location words like left, right, above, below, etc. Also, calculate the 2D center distance and the Euclidean center distance between them. Your output should follow this format: {example}. The coordinates are image coordinates normalized to a scale of 0 to 1 starting from the top left corner.'

'Could you describe the spatial locations of {ego_obj} relative to {ref_obj} in this image using words such as left, right, above, below, etc.? Additionally, find the 2D center distance and the Euclidean center distance between them. Your output must be in this format: {example}. The coordinates are image coordinates normalized to a scale of 0 to 1 starting from the top left corner.'

Table 38: Prompt candidates for temporal relationship {scene} will be replaced with a list of objects, {ego_obj} and {ref_obj} will be replaced with object names and {example} will be replaced with a random temporal relationship from the same image.

"The image shows a scene at the first timestamp, while the second image described as {scene} shows the next timestamp. Can you describe the change in the relative location of {ego_obj} compared to {ref_obj} between these two timestamps? Use relative distance words like getting closer or further away, etc. Also, find the change in the 2D center distance and the Euclidean center distance between the two images. Your output must follow this format: {example}. The coordinates are image coordinates normalized to a scale of 0 to 1 starting from the top left corner.'

'In the first timestamp, the image shows a scene, and the second image described as {scene} depicts the next timestamp. Can you describe the change in the relative location of {ego_obj} compared to {ref_obj} between these two timestamps? Use terms like getting closer or moving further away, etc. Additionally, find the change in the 2D center distance and the Euclidean center distance between the two images. Your output must follow this format: {example}. The coordinates are image coordinates normalized to a scale of 0 to 1 starting from the top left corner.'

'The scene in the first image is at the initial timestamp, and the second image described as {scene} shows the subsequent timestamp. Can you explain the change in the relative location of {ego_obj} compared to {ref_obj} between these two timestamps? Use words like getting closer or moving further apart, etc. Also, calculate the change in the 2D center distance and the Euclidean center distance between the two images. Your output should be formatted as follows: {example}. The coordinates are image coordinates normalized to a scale of 0 to 1 starting from the top left corner.'

'At the first timestamp, the image shows a scene, and the second image described as {scene} represents the next timestamp. Can you detail the change in the relative location of {ego_obj} compared to {ref_obj} between these two timestamps? Use relative distance words like moving closer or getting further away, etc. Additionally, find the change in the 2D center distance and the Euclidean center distance between the two images. Your output must follow this format: {example}. The coordinates are image coordinates normalized to a scale of 0 to 1 starting from the top left corner.'

'The first image shows a scene at an initial timestamp, and the second image described as {scene} depicts the next timestamp. Can you describe the change in the relative position of {ego_obj} compared to {ref_obj} between these two timestamps? Use terms such as getting closer or moving further apart, etc. Also, determine the change in the 2D center distance and the Euclidean center distance between the two images. Your output should follow this format: {example}. The coordinates are image coordinates normalized to a scale of 0 to 1 starting from the top left corner.'

'The initial timestamp shows a scene in the first image, and the second image described as {scene} represents the next timestamp. Can you describe how the relative location of {ego_obj} compared to {ref_obj} changes between these two timestamps? Use relative distance words like getting closer or moving further away, etc. Also, find the change in the 2D center distance and the Euclidean center distance between the two images. Your output must be in this format: {example}. The coordinates are image coordinates normalized to a scale of 0 to 1 starting from the top left corner.'

'The image shows a scene at the first timestamp, and the second image described as {scene} shows the subsequent timestamp. Can you detail the change in the relative location of {ego_obj} compared to {ref_obj} between these two timestamps? Use words like getting closer or moving further apart, etc. Also, calculate the change in the 2D center distance and the Euclidean center distance between the two images. Your output should be formatted as follows: {example}. The coordinates are image coordinates normalized to a scale of 0 to 1 starting from the top left corner.'

Table 39: (Continued) Prompt candidates for temporal relationship {scene} will be replaced with a list of objects, {ego_obj} and {ref_obj} will be replaced with object names and {example} will be replaced with a random temporal relationship from the same image.

'At the initial timestamp, the image shows a scene, and the second image described as {scene} depicts the next timestamp. Can you describe the change in the relative position of {ego_obj} compared to {ref_obj} between these two timestamps? Use relative distance terms such as getting closer or moving further away, etc. Also, find the change in the 2D center distance and the Euclidean center distance between the two images. Your output must follow this format: {example}. The coordinates are image coordinates normalized to a scale of 0 to 1 starting from the top left corner.'

'The scene in the first image is at the initial timestamp, and the second image described as {scene} shows the following timestamp. Can you describe the change in the relative location of {ego_obj} compared to {ref_obj} between these two timestamps? Use words like getting closer or moving further apart, etc. Additionally, calculate the change in the 2D center distance and the Euclidean center distance between the two images. Your output should follow this format: {example}. The coordinates are image coordinates normalized to a scale of 0 to 1 starting from the top left corner.'

'The first image shows a scene at an initial timestamp, and the second image described as {scene} depicts the next timestamp. Can you explain how the relative location of {ego_obj} compared to {ref_obj} changes between these two timestamps? Use relative distance words like moving closer or getting further away, etc. Also, determine the change in the 2D center distance and the Euclidean center distance between the two images. Your output must follow this format: {example}. The coordinates are image coordinates normalized to a scale of 0 to 1 starting from the top left corner.'

'The image shows a scene at the initial timestamp, and the second image described as {scene} shows the next timestamp. Can you describe the change in the relative position of {ego_obj} compared to {ref_obj} between these two timestamps? Use words like getting closer or moving further apart, etc. Also, calculate the change in the 2D center distance and the Euclidean center distance between the two images. Your output should follow this format: {example}. The coordinates are image coordinates normalized to a scale of 0 to 1 starting from the top left corner.'

'At the first timestamp, the image shows a scene, and the second image described as {scene} depicts the next timestamp. Can you detail how the relative location of {ego_obj} compared to {ref_obj} changes between these two timestamps? Use relative distance terms such as moving closer or getting further away, etc. Also, find the change in the 2D center distance and the Euclidean center distance between the two images. Your output must follow this format: {example}. The coordinates are image coordinates normalized to a scale of 0 to 1 starting from the top left corner.'

'The initial timestamp shows a scene in the first image, and the second image described as {scene} represents the next timestamp. Can you describe the change in the relative location of {ego_obj} compared to {ref_obj} between these two timestamps? Use words like getting closer or moving further apart, etc. Also, determine the change in the 2D center distance and the Euclidean center distance between the two images. Your output should follow this format: {example}. The coordinates are image coordinates normalized to a scale of 0 to 1 starting from the top left corner.'

'The first image shows a scene at the initial timestamp, and the second image described as {scene} shows the following timestamp. Can you describe how the relative position of {ego_obj} compared to {ref_obj} changes between these two timestamps? Use terms like moving closer or getting further away, etc. Additionally, find the change in the 2D center distance and the Euclidean center distance between the two images. Your output must follow this format: {example}. The coordinates are image coordinates normalized to a scale of 0 to 1 starting from the top left corner.'

