# OpenReview forum: "LLaRA: Supercharging Robot Learning Data for Vision-Language Policy"
_ICLR.cc/2025/Conference — ICLR 2025 Poster_

### Official Review · Reviewer_3YdQ · 2024-10-15

**Soundness:** 2
**Presentation:** 3
**Contribution:** 1
**Rating:** 3
**Confidence:** 3

**Summary:**

The authors propose to fine-tune an LLM into a robot policy by formatting robot decision-making as textual “conversations.” They provide an automated pipeline for converting existing demonstration data into their expected conversational text format. Additionally, they augment that dataset with additional helpful features, such as object bounding boxes extracted from some other vision model / simulation ground truth.

**Strengths:**

- LLaRA seems well-motivated, given recent interest in fine-tuning (V)LMs for control
- It seems to make intuitive sense that, if you’re starting from an instruction-following “conversational” policy, you’d want to fine-tune it to similarly respond to “conversations.”
  - That said, the extent to which this is the case is unclear: see questions.
- The authors communicate the format of their data and tasks extremely well, providing extensive examples.

**Weaknesses:**

**PLEASE READ MY FOLLOW-UP COMMENT.**

**[IGNORE] This paper is extremely similar to [Embodied Chain-of-Thought Reasoning (ECoT)](https://embodied-cot.github.io/)**, which precedes this work (being accepted at CoRL 2024) and goes uncited in the paper. For reference:
- **[IGNORE]** ECoT also proposes to format useful features for robot control into a string, then fine-tune a vision-language model into a robot policy that predicts said strings.
  - The paper does not explicitly state they formulate robot control as “chat,” but as reasoning instead.
  - However, this seems to be a somewhat superficial distinction; the only difference is the extent to which the “reasoning” is dressed up as a conversation. It is unclear if there’s a reason that one format would be better than another.
- **[IGNORE]** As with this paper, the ECoT authors propose to extract e.g., bounding boxes, which serve as an intermediate feature which the model predicts.
  - A minor difference is that ECoT always predicts the bounding boxes as part of generating its reasoning and action, whereas LLaRA requires a separate call of the model (if I am understanding Appendix A.3 correctly), which may introduce additional system complexity. In either case, both ECoT and LLaRA can accept detections from other models as well.
- **[IGNORE]** Both papers use pretrained models to "supercharge" robot demo datasets with additional automatically-generated synthetic annotations.
  - **[IGNORE]** While the reasoning steps employed by ECoT are not identical to the auxiliary datasets in LLaRA, a lot of the ones in the latter seem either simpler than the ones in ECoT (e.g., the future prediction step does not require reasoning by an external model, just formatting of existing data into a template / rephrased variations thereof) or are intractable or not robustly detectable in the real world with careful human annotation (action prediction, temporal prediction both are only easy if you have simulator state information).
  - In contrast, all experiments in ECoT were done in the real world, and all features and reasoning steps were extracted via pretrained foundation models. While noisy, the authors of that paper nonetheless showed empirical improvement in having the policy predict those features.
  - Additionally, ECoT demonstrates their pipeline on the Bridge dataset, which is much more representative of existing real-world robot demo datasets (with other datasets in collaborative efforts like Open-X-Embodiment having similar structure). As the BC datasets considered by LLaRA only do 2D pick-and-place high-level actions, it seems unclear if this will be practical for the more complex kinds of actions present in e.g., OXE again.

**Likewise, most of the limitations mentioned in the appendix seem to be addressed by the ECoT paper:**
> First, some concepts in the reference images still can not be easily and precisely described by language, while in many cases these features are critical for robot control. Due to the single image limit of the current VLM, we still rely on the object detector trained on a limited number of classes to describe the additional images in the task description, which limits the generalization ability of this method.

- **[IGNORE the first part, but the second part is still applicable]** ECoT specifies that they use a VLM scene captioner and open-vocabulary bounding box generator to circumvent this. This does not completely remove the issue that some scenes are just naturally hard to describe in language, but at least removes the limitation of using object detectors trained on a limited number of classes.

> Second, the information extracted from the dataset can be noisy, which may lead to model confusion. In Tab. 5, there are discrepancies such as the location of an object in a reference image differing from its actual location in the current image observation.

- **[IGNORE]** ECoT seems like it would be much less susceptible to noisy labels, since it’s just conditioning on them, rather than relying on them to exactly determine the action (parametrized by pixel coordinates that seem to be precisely dependent on the detected position of objects).
- **[IGNORE]** All main experiments in ECoT were conducted in the real world, where one must expect there to always be more noise than in simulation. It thus seems like said approach seems robust to the noise.
- Likewise, the real-world experiments for LLaRA seem very limited. In particular, from the examples shown in Table 13, it seems like all real world tasks can be completed by simply copying the values from the detector:
  - “Rotate the <obj>.at <x, y> by <r> degrees” is correctly solved by the command “Pick <obj> at <x, y>, rotate <r>”
  - “Put <obj1> at <x1, y1> on top of <obj2> at <x2, y2>” is correctly solved by the command “Pick <obj1> at <x1, y1>, rotate 0, place at <x2, y2>”
  - Given that no complex spatial reasoning is needed here, this seems like an insufficient demo of LLaRA’s performance.
  - It is likewise unclear why LLaRA pretrained in sim doesn’t succeed at this, or why LLaRA trained on real-world data doesn’t get 100% (intuitively the task seems exceedingly easy, since it just involves copying values?)

> Finally, our method relies on 2D to 2D mapping to convert image coordinates into actual actions, which may face challenges when the complex 3D movement of the robot is required. We believe enhancements in the mentioned aspects could further improve performance and broader applicability in complex, real-world applications.

- **[IGNORE]** ECoT states that they adopt the same tokenization scheme as OpenVLA / RT-2, which is significantly more expressive than the pick-and-place actions parametrized by start and end 2D location and rotations.
- Some of the tasks they test on seem to be very hard to express in the simple 2D-to-2D action parameterization used in LLaRA (e.g., wiping); it thus seems clearly important to demonstrate approaches that work well with such action representations (the “RT-2 style,” as named in the paper), but there is limited evidence that LLaRA could do so, especially since so much of it seems inherently tied to the simplified 2D-to-2D case (e.g., expressing 3D rotations in language is much more unwieldy than the 2D rotation case presented in this paper).
- Additionally, unless I am misunderstanding, LLaRA seems to only consider cases where tasks can be solved by a handful of 2D pick-and-place actions (with the real world experiments seemingly only to require one per episode). In the execution of that one action, the robot seems to be open-loop (that is, it cannot detect failure over the course of a single pick-and-place action, only before and after -- please correct me if I am misunderstanding). **[IGNORE]** In contrast, since ECoT uses "lower-level" actions, it can detect, reason about, and correct mistakes earlier.

**Given all this, it is unclear if LLaRA makes significant contributions over ECoT.** As discussed above, there are certainly subtle differences between the two approaches, but many of those differences seem to either be design choices or in ECoT's favor.

**Questions:**

**[All questions should NOT be ignored, they still apply.]**
- Why does sim LLaRA fail at the real-world tasks, if they seem to be so easy? Why doesn’t fine-tuned LLaRA achieve near 100% success, except in the last row of the FT section of Table 3?
- Does LLaRA ever experience e.g., failure to grasp, object slipping from gripper, or other similar failure cases in sim, and if so, is it able to recover?
- Similarly, it looks like in Table 5, LLaRA is trained on text traces that say “you have completed __.” Does it have to predict this itself, or is it just assumed that, after executing an earlier step, that step was successful, and so it can go in the action history?
  - If that’s the case, that seems to be another weakness compared to ECoT, as the latter approach has a reasoning step that determines which step of the plan it generated it should try to execute given the current observation. Thus, if it observes a failure case, it can return to and older step to try again
- A big point of ECoT (and even non-reasoning works that train a VLM into a robot policy, like OpenVLA) is that the VLM provides good visuo-semantic representations that could be useful for robots, especially for generalization. Is this observed by LLaRA, and if so, how / to what extent?
  - "In general, LLaRA demonstrates a robust capacity for generalization. Our observations confirm that it takes much less effort and data to transfer LLaRA models pretrained on simulated VIMA data to a real-world domain"
- Did formatting things as conversations actually help in some way, rather than just formatting in general (potentially templated) language? It seems like the conversational responses are nonetheless very structured and templated, so it’s unclear to me how much the VLM’s conversational abilities are retained (or even matter for pretraining).
- More broadly, did formatting things in natural language make a difference? What happens if the policy's response was always just in action tokens, akin to the RT-2-style?
  - Related: Did RT-2-style have the same expressivity as your approach? Since you're outputting <x, y> coordinates as string representations of floating points, it seems like your approach needs more tokens than what RT-2-style policies are even allowed to output. Since you output the rotation in degrees (from -359 to 359), it seems unfair to give RT-2 only a single rotation token with 256 bins. That seems to artificially make RT-2-style policies less expressive than yours.
- Similarly, what were the failure cases of RT-2 style? How did it achieve literally 0 across all real-world tests? Likewise, what were the empirical qualitative failures that made it work less well in simulation as well? As it stands, it seems to me that one such reason would be that, by expressing object positions and motions in the same coordinate system, it becomes artificially easier, since the VLM backbone may have a strong bias to being able to just learn to copy the object coordinates as the action parameters.
- Related: it seems like having prompts like "Place the object at <coordinates> in the bowl at <coordinates>" makes things a lot easier (when you've got a good object detector), especially when your actions are specifically parameterized in that coordinate space. What happens when you remove the coordinates in the prompt? This is important to test, since 2D pick-and-place of this form is one of the few robotics settings where that mapping from object coordinates to action coordinates is extremely trivial; most realistic robot learning applications will not be like that.

---

> ### Comment · Reviewer_3YdQ · 2024-11-14
>
> I've been notified by the Area Chair that the ECoT work referenced in my review falls under concurrent work under ICLR's guidelines -- that's completely my bad, and I apologize!
>
> Nevertheless, I am opting to maintain my score at this time. Many of the points I brought up still stand even without this comparison -- namely:
> 1. That this work does not evaluate on sufficiently complex environments, only 2D pick-and-place. It is unclear if this approach would generalize to other manipulation domains, like Bridge, with actions like wiping being hard to express in "conversational text" as described in this paper.
> 2. The tasks considered still seem to be solvable by copying values from either the task specification or from the object detection mechanism into the action space. It does not seem clear to me that one needs a complex LLM-based system for this.
> 3. Given the above, if the policy does learn some kind of "copying," it seems like LLaRA would be very susceptible to noisy labels (or noisy object detection generations), which seems like a major downside for real-world deployment.
> 4. All the points in my "Question" section still stand.
>
> I will edit my original comment to make it clear which sections can now be ignored. I again apologize for any inconvenience, and would appreciate it if the authors could focus on the remaining points I made instead.

---

> ### Author Response · Authors · 2024-11-25
> **(1/5) Comment for Reviewer 3YdQ**
>
> We thank the reviewer for revising the original review to follow the reviewer guidelines.
>
> We believe there are a couple of important misunderstandings going on, we would like to first clarify them (before going into individual detailed comments). We apologize for the confusion and we will improve the presentation in the revised paper:
>
> **1. The argument that LLaRA relies on object detection coordinate numbers for action generation is an important misunderstanding. We have models that do not use any object detection (inBC), and for the models using object detection (D-inBC), we only do so for the reference images. We also showed that LLaRA is able to address tasks not solvable by simply copying and pasting coordinates (VIMA dataset).**
>
> We would like to address and clarify the reviewer's first misunderstanding. `VLM backbone may have a strong bias to being able to just learn to copy the object coordinates as the action parameters.`
> Our inBC is a strong framework proposed in this paper, and all our inBC models solely rely on the provided image to make the action decision. It does not have any object detection or any coordinates in the prompt. Yet, as shown in Table 3, Table 16, and Table 18, inBC shows a reasonable accuracy much higher than RT-2 Style. It still achieves strong performance (88.3% average success rate) when finetuned with very limited data, as shown in the following additional experiments.
> | Protocol     | Method            | Details             | Add. Data       | T1 (%)     | T2 (%)     | T3 (%)     | Avg. (%)     |
> |--------------|-------------------|---------------------|-----------------|------------|------------|------------|--------------|
> | FT           | RT-2 Style        |                     | xArm-Det        | 0          | 0          | 0          | 0            |
> |              | LLaRA (Ours)      | inBC + Aux (D)      | xArm-Det        | 100        | 95         | 70         | 88.3         |
> |              |                   | inBC + Aux (D)      | xArm-Action     | 70         | 75         | 80         | 75           |
> | JT           | LLaRA (Ours)      | inBC + Aux (D)      | xArm-Det        | 75         | 85         | 70         | 76.6         |
> |              |                   | inBC + Aux (D)      | xArm-Action     | 50         | 90         | 50         | 63.3         |
>
> Our D-inBC benefits from object descriptions and coordinates **in the reference images (not the observation)**. However, since they are the coordinates of the reference images, their numbers are not directly compatible with the main input image.
>
> For example, in Table 5, the coordinates of the rainbow letter T `<b>(0.500, 0.594), {0.102, 0.188}</b>` are given in the reference image coordinates (first image in the first row of the table),  the wooden bowl at `<b>(0.457, 0.531), {0.195, 0.328}</b>` is also similar, and the coordinates are given regarding the second image of the first row in the table.
> These coordinates are totally different from the actual positions of the objects in the current observation (the pick point `<b>(0.480, 0.367)</b>` and the place point `<b>(0.727, 0.547)</b>`), shown in the images at the second and third rows of the table. Therefore, copying such wrong coordinates will not help solve the problem.

---

> ### Author Response · Authors · 2024-11-25
> **(1.5/5) Comment for Reviewer 3YdQ**
>
> Following the suggestion from the reviewer, we trained a version of D-inBC, which completely removes all the coordinate numbers in the prompt even from the reference images. Instead, we use a pretrained VLM (Qwen2-VL-7B-Instruct) to generate natural language descriptions of all reference images, omitting any use of image coordinates. These textual descriptions replaced the original lists of image coordinates in the input prompt for the D-inBC setting, resulting in a variant we refer to as **D-inBC (L)**, where L stands for language. Then we train on robotics data (VIMA) and the conversations describing the reference image. During inference, LLaRA first generates textual descriptions of all reference images using its own capabilities and subsequently leverages the generated text to perform the task.
>
> For example, the first image in Table 6 is described by the following text
> > The image displays three distinct objects, each with unique colors, shapes, textures, and spatial relationships. 1. **Object on the Left:** - **Color:** Red - **Texture:** Matte - **Shape:** Triangular - **Spatial Relationship:** Positioned to the far left of the image. 2. **Object in the Middle:** - **Color:** Green and white - **Texture:** Speckled - **Shape:** Arrowhead - **Spatial Relationship:** Situated to the right of the first object. 3. **Object on the Right:** - **Color:** Multicolored (red, blue, green, and yellow) - **Texture:** Glossy - **Shape:** Trapezoidal - **Spatial Relationship:** Located to the far right of the image. These objects are arranged in a linear sequence, with each subsequent object placed to the right of the previous one, creating a horizontal alignment. The colors and textures vary, adding visual interest to the composition.
>
> This text will be used to replace the original list of object coordinates in the last row of Table 6
>
> > <scene><p>red letter V</p> at <b>(0.254, 0.570), {0.094, 0.156}</b>.<p>green paisley letter V</p> at <b>(0.500, 0.570), {0.094,0.156}</b>. <p>rainbow letter V</p> at <b>(0.746, 0.570), {0.094,0.156}</b>.</scene>
>
> Real-world experiment results are attached below
> | Protocol     | Method            | Details             | Add. Data       | T1 (%)     | T2 (%)     | T3 (%)     | Avg. (%)     |
> |--------------|-------------------|---------------------|-----------------|------------|------------|------------|--------------|
> | FT           | RT-2 Style        |                     | xArm-Det        | 0          | 0          | 0          | 0            |
> |              |                   |                     |                 |            |            |            |              |
> |              | LLaRA (Ours)      | inBC + Aux (D)      | xArm-Det        | 100        | 95         | 70         | 88.3         |
> |              |                   | inBC + Aux (D)      | xArm-Action     | 70         | 75         | 80         | 75           |
> |              |                   | D-inBC (L) + Aux (D)| xArm-Det        | 65         | 90         | 70         | 75           |
> |              |                   | D-inBC (L) + Aux (D)| xArm-Action     | 75         | 90         | 75         | 80           |
>
> In addition, VIMA benchmark is challenging and cannot be easily solved by copy-pasting coordinates. Many of the tasks require a proper language understanding and are not solvable without it, not to mention that none of our models has access to the object coordinates in the current observations.
>
> For example, the task of `manipulate_old_neighbor` could not be solved by copy-pasting either. The task description is
>
> > `First put {object image} into {another object image}, then put the object that was previously at its south into the same {the second object image}`.
>
> (Visual reference is at https://github.com/vimalabs/VIMABench/raw/main/images/tasks/manipulate_old_neighbor.gif)
> Besides the ability to understand the object identity given in image format, the model needs to have both memory (`previously at`) and understanding of the spatial location (the word `south`) to solve the task properly.
> That said, even if we had access to the object detection results on the current observation (which we don’t actually have),  the majority of the VIMA benchmark tasks (12 out of 17) could not be easily solved by copying the values: for example, scene_understanding; rearrange; rearrange then restore; follow motion; follow order; sweep_without_exceeding; sweep_without_touching; same_texture; same_shape; manipulate_old_neighbor; pick_in_order_then_restore; rotate and twist.

---

> ### Author Response · Authors · 2024-11-25
> **(2/5) Comment for Reviewer 3YdQ**
>
> **2. LLaRA is capable of completing tasks beyond 2D pick-and-place.**
>
> VIMA has multiple tasks, that are much more than 2D pick and place. For instance, tasks like  `sweep_without_exceeding` (https://github.com/vimalabs/VIMABench/raw/main/images/tasks/sweep_without_exceeding.gif) require the policy to control the robot with a spatula to push objects and to understand the meaning of `exceeding`. We empirically find that our method could solve such problems. Additionally, our method could learn to generate a sequence of image coordinates to present tasks like wiping, which is similar to the sweep tasks.

---

> ### Comment · Reviewer_3YdQ · 2024-11-26
> **Reply to Authors**
>
> Thanks to the authors for the response (though the comments say 2 of 5, so I am not sure they’re done sending them?)
>
> - I appreciate the clarification on reference images. Just to further clarify: the policy receives these reference images as well as the observation then? In that case, it does seem that this is quite a VIMA bench specific concept – most real-world robotics datasets do not have snippets showing exactly what the objects being referenced are, usually they just have goal images or freeform language instructions. It thus remains unclear to me if “supercharging” on labels of this domain-specific data type is viable for general robot policy learning.
> - Additionally, if that is the case, it seems like a cause of success for tasks with reference images just comes from knowing visually what the objects referenced look like. It’s not clear to me that other robotics domains are like this: even if the policy knows exactly what it should be interacting with, if the object structure is strange, it might still not be able to pick that object up when using parallel jaw grippers rather than suction.
> - Thank you for clarifying that VIMA bench does include wiping. I think this was unclear due to the description in the paper saying “The robot action space is two 2D coordinates for pick and place positions and two quaternions for rotations.” Additionally, while it does have these tasks, it seems like the sweeping example presented above likewise uses actions parameterized by 2D coordinates; I think my initial concern thus still stands, since RT-2 action tokenization can still easily express tasks such as pouring and the like (as demonstrated by OpenVLA, which uses the RT-2 action tokenization scheme), while there has been insufficient demonstration that your approach helps with tasks requiring 6DoF actions.
> - I may be wrong, but my impression is that VIMA bench will not be able to test for such tasks, but since so many real-world applications require them, I think it is important to show that LLaRA can be applied to them. I recognize that the above criticisms are shortcomings of VIMA bench and not your approach, but since that’s the only benchmark tested on, I don’t think it provides sufficient evidence of LLaRA’s general applicability to robotics.
> - While you pointed out tasks in VIMA bench that require some degree of e.g., spatial relationship understanding, I have two concerns:
>   - It seems like your supercharging approach seems particularly tailored for enforcing that the VLM policy should learn such relations.
>   - None of these more complex tasks are tested in the real world – the three chosen tasks, place in bowl / rotate some number of degrees / place on tray, are either pick and place or simpler; they all seem to be solvable just from knowing the position of relevant objects (and for rotation, copying the instructed number of degrees into the action). Even if I misunderstood the fact that the given object locations are in the reference image and not the actual observation, it seems like the tasks should not be that complex for a policy that can identify objects from its representations.
> - Is the newly trained policy only evaluated on the three real-world tasks? I suspect that it would damage performance even more if done on the VIMA bench task you gave as an example in your response, where there are three intermediate steps that would all need these general, freeform language descriptions. It does seem to already degrade performance in the real world tasks you evaluated on, but since all three seem to be solvable in one step and just requiring knowing the location of a single target object in the scene (as well as the bowl or plate), I suspect that the freeform language is also particularly usable or beneficial in this case.
> - Again, I can’t really take anything away from the comparison with RT-2 policies in the real world that flat out achieve zero. I would really need to know more details about the failure cases (what’s the systematic reason that the RT-2-style policy fails?) and how fair the tokenization actually is. This also goes for the sim results where RT-2 underperforms, but is more salient for the real-world experiments.
> - I would appreciate answers to my remaining questions.
>
> Given the above, I am choosing to maintain my score, but decrease my confidence in it at this time. I understand that it may be impractical in the remaining time, but I think that results on tasks that are more similar to the complex ones encountered in real-world robot learning datasets would be extremely important in changing my current assessment of the present approach.

---

> > ### Author Response · Authors · 2024-11-26
> > **(3/5) Comment for Reviewer 3YdQ**
> >
> > We also have some clarifications on our real-robot experiment settings, and why RT-2 Style performs poorly.
> >
> > **3-1**. Although our real-robot experiment setting might look simple, it is designed to be extremely challenging in a different aspect: visual representation transfer, which is the focus of this paper. We wanted to show with our experiments that VLM to VLA representation adaptation becomes possible and much faster with our 'instruction style' action formulation and 'self-supervised auxiliary' training data.
> >
> > Our real-world setting is very challenging for all the models (as we can see from GPT-4o and RT-2-Styles that fail) because of the large data domain gaps and the amount of training data provided. We provided a very limited amount of training data, both in the simulation pretraining and the real-world finetuning.
> >
> > In the pretraining, we only provide the models \~14k simulated training samples from 8k episodes in VIMA (i.e., 1.2% setting), making the action policy learning very challenging from the beginning. Then we finetune it with 1.2k real-world image frames, with limited overlap in task similarity to the real-world tasks.
> >
> > In order to further clarify the limitation of RT-2 with the limited amount of training data, we also added an experiment where RT-2-Style is pretrained with 100% of VIMA data (which is unfair as it now uses ~80x more data than LLaRA), and then finetune it with 1.2k real-world image frames as before. The below table shows the results:
> >
> > | Protocol | Method                    | Add. Data               | Task          | T1 (%) | T2 (%) | T3 (%) | Avg. (%) |
> > |----------|---------------------------|-------------------------|---------------|--------|--------|--------|----------|
> > | FT       | RT-2 Style                | 1.2% VIMA data          | xArm-Det      | 0      | 0      | 0      | 0        |
> > |          | RT-2 Style                | 100% VIMA data          | xArm-Det      | 0      | 0      | 0      | 0        |
> > |          | LLaRA (Ours)              | inBC                    | xArm-Det      | 0      | 0      | 0      | 0        |
> > |          |                           | inBC                    | xArm-Action   | 30     | 45     | 5      | 26.6     |
> > |          |                           | inBC + Aux (D)          | xArm-Det      | 100    | 95     | 70     | 88.3     |
> > |          |                           | inBC + Aux (D)          | xArm-Action   | 70     | 75     | 80     | 75       |
> > |          |                           | D-inBC (L) + Aux (D)    | xArm-Det      | 65     | 90     | 70     | 75       |
> > |          |                           | D-inBC (L) + Aux (D)    | xArm-Action   | 75     | 90     | 75     | 80       |
> > |          |                           | D-inBC + OD             | xArm-Det      | 0      | 0      | 0      | 0        |
> > |          |                           | D-inBC + OD             | xArm-Action   | 45     | 80     | 55     | 60       |
> > |          |                           | D-inBC + Aux (C) + OD   | xArm-Det      | 70     | 90     | 70     | 76.6     |
> > |          |                           | D-inBC + Aux (C) + OD   | xArm-Action   | 90     | 100    | 85     | 91.6     |
> >
> > Note that all LLaRA models use 1.2% of VIMA data.
> >
> > As we are able to observe, even with 100% VIMA data for the pretraining, RT-2 style suffered. The amount of real-robot training data is simply lacking for RT-2 to adjust its VLA model for the new environment. On the other hand, LLaRA was able to benefit from it better.
> >
> > **3-2**. In addition, following the suggestion from the reviewer, we have added experiments with a new set of tasks that require more language understanding and spatial reasoning.
> > These tasks include:
> >
> > - Put all yellow objects into the bowl: There is a bowl and 3 objects on the table. Between 1 and 3 of the objects are yellow. A success state is one in which all yellow objects are in the bowl and no non-yellow objects are in the bowl.
> >
> > - Transport the baked goods to the large bowl: There is a bowl and 3 objects on the table. Either 1 or 2 of the objects are baked goods. A success state is one in which all baked goods are in the large bowl and none of the non-baked goods are in the large bowl.
> >
> > - Feed the pet healthy food: There is an animal on the table and 2 objects. One of the objects is a fruit or vegetable and the other is a baked good. The 2 objects are equidistant from the animal to start. A success state is one in which the healthy food is sitting next to the animal while the unhealthy food is further away from the animal.
> >
> > Without a proper language understanding, these tasks cannot be easily solved. These tasks are particularly challenging as none of these tasks were seen in the training data in either sim or real. Some of the objects in the scene (e.g., `baked goods` and `bowl`) were never seen by the model either during the robot data training, and each episode takes multiple steps to finish.

---

> ### Author Response · Authors · 2024-11-26
> **(3.5/5) Comment for Reviewer 3YdQ**
>
> We compare RT-2 style and LLaRA(inBC + Aux (D)) both first trained on VIMA-8k and then finetuned on xArm-Det, which is the model used in the previous table without any changes. That is, the new tasks and the unseen tasks for the model. For each task, we run 10 randomly initialized episodes and report the success rate. The below table shows the results.
> | Task                                       | RT-2 | LLaRA |
> |--------------------------------------------|------|-------|
> | Put all yellow objects into the bowl       | 0    | 60    |
> | Transport the baked goods to the large bowl| 0    | 50    |
> | Feed the pet healthy food                  | 0    | 50    |
>
>
> As we are able to observe, LLaRA is able to handle tasks requiring spatial language understanding tasks to a certain degree. It sometimes gets confused due to its smaller LLM size (7B), but it shows the potential to effectively leverage the knowledge of LLaVA, enabling efficient transfer to robot control with minimal data. This capability is primarily facilitated by our key contributions: the alignment between action and image coordinates, and the integration of self-supervised auxiliary data.
>
> RT-2 style on the other hand, simply fails to utilize the limited amount of training data provided in our setting, despite using the same model architecture.

---

> ### Author Response · Authors · 2024-11-26
> **Reply to the reviewer**
>
> We apologize for posting our response a bit slowly. We have been waiting for some experimental results, corresponding to your suggestions. All the answers to all the questions will become available in ~1 hours.
>
> Regarding the new feedback you provided:
>
> We believe the biggest confusion comes from our lack of discussion on self-supervised (auxiliary) data in computer vision and how modern vision-language models (VLMs) have been benefiting from them. We apologize and we will revise the paper to include these explicitly. Let us try to clarify here (also gave a similar clarification to aFtC):
>
> What happened with the VLM research in the past ~1 year was an extensive amount of training dataset curation including those with automatically generated "object bounding boxes" (or "visual groundings"). Researchers have discovered that using a pretrained object detection module to extract their location information and then formulating such information in natural language sentences allows much better construction of the VLM **training** data. For instance, LLaVA training data was generated based on the MS COCO-trained object detection and GPT4-based sentence descriptions using them. Many newer models including Llama3, Qwen2-VL, BLIP-3, and more also follow this concept of bounding box-based self-supervised data generation, all confirming their benefits. It’s now a widely accepted paradigm.
>
> What we mean by "supercharging" in this paper is exactly such a paradigm extended for robot learning. In addition to VLM’s bounding box-based training data curation, we introduce robot-specific auxiliary tasks such as future prediction. Notice that this concept was not too popular at the time of RT-2 and was not picked up, as RT-2 is based on a bit older VLM training framework (PaLI and PaLM).
>
> All we wanted to do in this paper was to replicate the success of more modern VLM training in robot action policy learning, and we confirmed this with one of the challenging (and known) robotics simulation: VIMA. It’s a benchmark accepted by the community and is known to be challenging due to its language complexity. There is multiple evidence that this paradigm benefits VLM training, as mentioned above. We are adding one more evidence on top of that, this time, with robot learning data. This makes us believe our observation to transfer to other robotics datasets, and we are showing the potential for this.
>
> We do agree that the action space we focus on in this paper is translations and rotations in 2D image coordinates (which we adopted from VIMA). This is our design choice to enable faster representation transfer from a computer vision-based VLM to the robot action model. However, we would like to emphasize that VIMA and our real-robot settings have a totally different axis of difficulty: visual representation transfer with limited data, which conventional approaches like RT-2 suffer. We believe the new findings of our paper are orthogonal to other VLA models focusing on different aspects of the problem such as 3D manipulation. What we hope to bring with this paper is the importance of visual representation transfer in robot learning and how to do so with more modern techniques. This is one of the reasons why we are submitting this paper to ICLR, not ICRA.
>
> In conclusion, the goal of this paper is not to release the final generalist VLA product-level model that is trained on tons of data and is capable of solving every benchmark. As we mentioned in the beginning, our goal is to replicate the recent success of the computer vision paradigm (i.e., data curation) in robotics. We want to show the potential for future research that other researchers can benefit from.
>
> We will also continue to provide answers to your concerns over the next 1~2 days. Please do not hesitate letting us know the remaining concerns.

---

> ### Author Response · Authors · 2024-11-26
> **(4/5) Comment for Reviewer 3YdQ**
>
> We will now try our best to address each of the reviewers' comments individually:
>
> > The paper does not explicitly state they formulate robot control as “chat,” but as reasoning instead.
>
> > However, this seems to be a somewhat superficial distinction; the only difference is the extent to which the “reasoning” is dressed up as a conversation. It is unclear if there’s a reason that one format would be better than another.
>
> The conversation design is motivated by LLaVA [1] and many other computer vision studies on dataset curation [2 - 16]. It has been confirmed in many research studies on vision-language models for images [2-8, 12] as well as videos [13-16] that user-assistant conversation style significantly benefits the entire model training. Also, some of the studies (e.g., [2, 4, 9, 10]) explicitly show that structured or templated conversation performs comparably to LLM-generated diverse conversations. We wanted to replicate their successes in the robot learning domain, and we empirically confirm that our template-based single-turn conversations are simple and efficient, without the need to generate long sequences during inference.
>
> > Likewise, the real-world experiments for LLaRA seem very limited. In particular, from the examples shown in Table 13, it seems like all real world tasks can be completed by simply copying the values from the detector:
> > - “Rotate the <obj>.at <x, y> by <r> degrees” is correctly solved by the command “Pick <obj> at <x, y>, rotate <r>”
> > - “Put <obj1> at <x1, y1> on top of <obj2> at <x2, y2>” is correctly solved by the command “Pick <obj1> at <x1, y1>, rotate 0, place at <x2, y2>”
> > - Given that no complex spatial reasoning is needed here, this seems like an insufficient demo of LLaRA’s performance.
> > - It is likewise unclear why LLaRA pretrained in sim doesn’t succeed at this, or why LLaRA trained on real-world data doesn’t get 100% (intuitively the task seems exceedingly easy, since it just involves copying values?)
>
> We believe most of the concerns in this paragraph have been addressed by the first several posts. To provide more evidence, in addition to the experiments on new real-world tasks posted earlier, in Figure 14 of the original submission in the appendix, we show that LLaRA is able to finish real-world multi-step unseen tasks while manipulating unseen objects and benefit from the knowledge that exists in LLaVA. Specifically in the right-hand side example, the task description is `Get the weight of the tomato and put it back.` which requires the model to understand the connection between the `weight` and a scale and remember the previous position of the tomato. Such examples demonstrate the strong reasoning ability of LLaRA.
>
> Please just let us know if further clarifications are needed.
>
> > Some of the tasks they test on seem to be very hard to express in the simple 2D-to-2D action parameterization used in LLaRA (e.g., wiping); it thus seems clearly important to demonstrate approaches that work well with such action representations (the “RT-2 style,” as named in the paper), but there is limited evidence that LLaRA could do so, especially since so much of it seems inherently tied to the simplified 2D-to-2D case (e.g., expressing 3D rotations in language is much more unwieldy than the 2D rotation case presented in this paper).
>
> We acknowledge that the action in LLaRA is based on 2D image coordinates, which could be limiting in some cases. We have some discussion on it in Appendix D. This was our design choice to solve VIMA Benchmarks, and the focus of this paper is more on showing the potential of the self-supervised auxiliary data and action aligning for VLM to VLA representation learning. As we also discuss in the common post (at the top), this design allows much faster robot learning with limited data, and we confirm this experimentally (Figure 5, Table 14-20). The new findings of our paper are orthogonal to other VLA models focusing on different aspects of the problem such as 3D manipulation. We believe the observations on efficient VLM to VLA representation transfer we share with this paper are valuable not only to the robot learning community but also to the computer vision community.

---

> ### Author Response · Authors · 2024-11-26
> **(4.25/5) Comment for Reviewer 3YdQ**
>
> > Additionally, unless I am misunderstanding, LLaRA seems to only consider cases where tasks can be solved by a handful of 2D pick-and-place actions (with the real world experiments seemingly only to require one per episode). In the execution of that one action, the robot seems to be open-loop (that is, it cannot detect failure over the course of a single pick-and-place action, only before and after -- please correct me if I am misunderstanding).
>
> First, as we mentioned above,  there are tasks like `sweep_without_exceeding` that require the policy to control the robot with a spatula to push objects, and we empirically find that our method could solve this problem. In addition, results in Figure 14 in the appendix show that LLaRA is able to finish real-world multi-step unseen tasks while manipulating unseen objects.
>
> The pick-and-place action space is chosen simply because this is the setting used by VIMA. Though a single pick-and-place action is atomic, we empirically found that our model can recover from a failed action. We run additional tests where the simulated environment is revised so that there is a 10% chance that the action fails (which we believe is much higher than the actual case). The results listed below show that without any changes, our existing model is robust enough to such failure.
> |           | Prob. of Failure | L1 | L2 | L3  |
> |-----------|----|----|----|------|
> | LLaRA D-inBC + Aux(B) - VIMA80k | 0  | 90.0 | 88.1 | 79.2 |
> | LLaRA D-inBC + Aux(B) - VIMA80k | 10  | 87.7 | 86.2 | 75.8 |
>
> > Why does sim LLaRA fail at the real-world tasks, if they seem to be so easy? Why doesn’t fine-tuned LLaRA achieve near 100% success, except in the last row of the FT section of Table 3?
>
> As discussed earlier, all our real-world experiments are performed in a sim-to-real setting with very limited data, which makes the tasks very difficult for the models. The setting here is very challenging to the model and therefore RT-2 style baselines easily fail. Despite this, our models exhibit strong performance on real-world tasks with very limited in-domain data.
>
> > Does LLaRA ever experience e.g., failure to grasp, object slipping from gripper, or other similar failure cases in sim, and if so, is it able to recover?
>
> Yes, as discussed above, we run additional tests where the simulated environment is revised so that there is a 10% chance that the action fails (which we believe is much higher than the actual case). Our existing model can still keep up the performance with less than a 4.5% success rate drop among all the tasks.
>
> > Similarly, it looks like in Table 5, LLaRA is trained on text traces that say “you have completed __.” Does it have to predict this itself, or is it just assumed that, after executing an earlier step, that step was successful, and so it can go in the action history?
> If that’s the case, that seems to be another weakness compared to ECoT, as the latter approach has a reasoning step that determines which step of the plan it generated it should try to execute given the current observation. Thus, if it observes a failure case, it can return to and older step to try again
>
> The action history formatted in `you have finished _` is just a way to keep action history (i.e., it’s just a prompting style), where we simply repeat what was executed in the last step, and our model does not assume it succeeds or fails. As mentioned in the above comment, our policy learns to recover from the failed actions based on the training. We apologize for the confusion and will further clarify this in the revised paper.
>
> > A big point of ECoT (and even non-reasoning works that train a VLM into a robot policy, like OpenVLA) is that the VLM provides good visuo-semantic representations that could be useful for robots, especially for generalization. Is this observed by LLaRA, and if so, how / to what extent?
> "In general, LLaRA demonstrates a robust capacity for generalization. Our observations confirm that it takes much less effort and data to transfer LLaRA models pretrained on simulated VIMA data to a real-world domain"
>
> In Table 3 (as well as the table in the previous post), our model shows good sim-to-real performance, which suggests that our model shows the potential to learn good visuo-semantic representations with only 8k simulated episodes. Once more, we emphasize that one of our main focuses with LLaRA was to provide it the capability to learn and adapt quickly with a limited amount of training data.

---

> ### Author Response · Authors · 2024-11-26
> **(4.5/5) Comment for Reviewer 3YdQ**
>
> > More broadly, did formatting things in natural language make a difference? What happens if the policy's response was always just in action tokens, akin to the RT-2-style?
>
> We highlight the RT-2 Style (I) settings in our paper, which uses the exact same image coordinate action space as our method, but all the actions are presented in special action tokens as the reviewer suggested. Results in Figure 5 and Table 9 show that such a setting significantly underperforms our method.
>
> > Related: Did RT-2-style have the same expressivity as your approach? Since you're outputting <x, y> coordinates as string representations of floating points, it seems like your approach needs more tokens than what RT-2-style policies are even allowed to output. Since you output the rotation in degrees (from -359 to 359), it seems unfair to give RT-2 only a single rotation token with 256 bins. That seems to artificially make RT-2-style policies less expressive than yours.
>
> We follow the spirit of the **original RT-2** where the rotation range from -180 to 180 is first normalized to [0, 1] and then discretized into 256 bins. Our rotation action space is larger compared to 256 possible values in RT-2, but this difference is because we wanted to respect and replicate the original RT-2, and all training and test data do not include any cases where the absolute rotation is larger than 180 degrees.
> In terms of x and y actions, we have 256 possible values due to the image resolution while RT-2 style also has 256 bins. Meanwhile, we observed that the majority of the failure pattern of RT-2 is caused by failure to generate proper positions to operate instead of rotation.
>
> > Similarly, what were the failure cases of RT-2 style? How did it achieve literally 0 across all real-world tests? Likewise, what were the empirical qualitative failures that made it work less well in simulation as well? As it stands, it seems to me that one such reason would be that, by expressing object positions and motions in the same coordinate system, it becomes artificially easier, since the VLM backbone may have a strong bias to being able to just learn to copy the object coordinates as the action parameters.
>
> We observe that RT-2 style trained on VIMA-8k could not generate proper action sequences when testing in the real-world environment, either Zero-shot or Finetune setting. This is due to the fact that we only trained the model with 8k simulated examples before transferring it to the real-world domain.
>
> The main limitation of the RT-2 Style was that it required much more training data to converge to a reasonable accuracy (in simulation as well). In the end, the accuracy was reasonable with 80K training samples (~60% accuracy in Tables 18-20), but it was still significantly lower than LLaRA. RT-2 Style failed simply because it didn’t learn fast enough.
> If we add the sim-to-real domain difference on top of that, it is not surprising that RT-2 Style performs extremely poorly in this setting.
>
> In our new experiments reported in the previous table, when trained with 100% VIMA data, the model starts to generate action sequences, however, they are not very meaningful.
> Meanwhile, we respectfully disagree that an LLM behavior of “copy the object coordinates as the action parameters.” happened during the inference, which we clarified at the beginning of this post.

---

> ### Author Response · Authors · 2024-11-26
> **(5/5) Comment for Reviewer 3YdQ**
>
> > Related: it seems like having prompts like "Place the object at <coordinates> in the bowl at <coordinates>" makes things a lot easier (when you've got a good object detector), especially when your actions are specifically parameterized in that coordinate space. What happens when you remove the coordinates in the prompt? This is important to test, since 2D pick-and-place of this form is one of the few robotics settings where that mapping from object coordinates to action coordinates is extremely trivial; most realistic robot learning applications will not be like that.
>
> As discussed at the beginning of the post, inBC never receives any coordinates in the prompt and still achieves strong performance. D-inBC used coordinates from the **reference image**, but they were not the coordinates for the observation image.
> In addition, (as discussed in the above post), following the suggestion, we implemented a version of D-inBC that does not use any coordinate information from the images: D-inBC (L).
>
> **Conclusion**
>
> In conclusion, the primary focus of this paper is to enable faster training of the VLA model particularly for limited training data. All our design choices center around “how to enable easy/fast transfer of learned representations in a VLM to representations necessary for a robot model”. We show with our experiments that our approach does enable better utilization of training data than doing this in RT-2 style (while making the inference speed similar).
>
> We believe many of our findings including the introduction of various self-supervised auxiliary data and the use of text-based action representations are orthogonal to many concurrent researches and have the potential to benefit follow-up studies, particularly in the aspect of representation learning. We respectfully assert that these contributions deserve careful attention.
>
> We are happy to address any further questions or concerns the reviewer may have. Thank you for your thoughtful feedback.
>
> **Reference**
>
> [1] Liu, Haotian, et al. "Visual instruction tuning." Advances in neural information processing systems 36 (2023).
>
> [2] Cai, Mu, et al. "ViP-LLaVA: Making Large Multimodal Models Understand Arbitrary Visual Prompts." Proceedings of the IEEE/CVF Conference on Computer Vision and Pattern Recognition. 2024.
>
> [3] Zhu, Deyao, et al. "Minigpt-4: Enhancing vision-language understanding with advanced large language models." arXiv preprint arXiv:2304.10592 (2023).
>
> [4] Gong, Tao, et al. "Multimodal-gpt: A vision and language model for dialogue with humans." arXiv preprint arXiv:2305.04790 (2023).
>
> [5] Ouyang, Long, et al. "Training language models to follow instructions with human feedback." Advances in neural information processing systems 35 (2022): 27730-27744.
>
> [6] Li, Bo, et al. "Mimic-it: Multi-modal in-context instruction tuning." arXiv preprint arXiv:2306.05425 (2023).
>
> [7] Wei, Jerry, et al. "Simple synthetic data reduces sycophancy in large language models." arXiv preprint arXiv:2308.03958 (2023).
>
> [8] Honovich, Or, et al. "Unnatural instructions: Tuning language models with (almost) no human labor." arXiv preprint arXiv:2212.09689 (2022).
>
> [9] Jia, Baoxiong, et al. "Sceneverse: Scaling 3d vision-language learning for grounded scene understanding." European Conference on Computer Vision. Springer, Cham, 2025.
>
> [10] Ranasinghe, Kanchana, et al. "Learning to localize objects improves spatial reasoning in visual-llms." Proceedings of the IEEE/CVF Conference on Computer Vision and Pattern Recognition. 2024.
>
> [11] Zhang, Shengyu, et al. "Instruction tuning for large language models: A survey." arXiv preprint arXiv:2308.10792 (2023).
>
> [12] Li, Chen, et al. "Vision-language instruction tuning: A review and analysis." arXiv preprint arXiv:2311.08172 (2023).
>
> [13] Lin, Bin, et al. "Video-llava: Learning united visual representation by alignment before projection." arXiv preprint arXiv:2311.10122 (2023).
>
> [14] Maaz, Muhammad, et al. "Video-chatgpt: Towards detailed video understanding via large vision and language models." arXiv preprint arXiv:2306.05424 (2023).
>
> [15] Li, KunChang, et al. "Videochat: Chat-centric video understanding." arXiv preprint arXiv:2305.06355 (2023).
>
> [16] Zhang, Hang, Xin Li, and Lidong Bing. "Video-llama: An instruction-tuned audio-visual language model for video understanding." arXiv preprint arXiv:2306.02858 (2023).

---

> ### Comment · Reviewer_3YdQ · 2024-11-26
> **Discussion Period Response (1 / 2)**
>
> Thanks again to the authors for the detailed response. I greatly appreciate how deeply they engaged with my concerns.
>
> I am willing to increase my score by one level, but I do not think I am willing to increase further than that.
> EDIT: My bad -- I thought there was an option for weak reject, but this is not the case. In that case, I would prefer to stay at the current level then.
>
> The main concern I have is that robotics domains in 2D action spaces are fundamentally very limited. The limiting factor in such cases is ultimately just knowing where objects are and where they need to go. The language complexity demonstrated is and highlighted by the authors is 1) knowing what language refers to which objects and locations, 2) knowing relative spatial relations (e.g., “to the south of”), 3) history and dynamics (on the scale of a handful of key timesteps, e.g., for the “return the object to where it was” example), and 4) semantic associations between concepts and objects (e.g., knowing that scales are used for weighing things). To be clear, these three language grounding cases are useful for both 2D action spaces and more general 6DoF manipulation. Additionally, I recognize that the “supercharging” data is useful for enforcing a bias towards VLMs learning these generalizable visuo-linguistic relationships.
>
> The problem is, it seems like for 2D action space task domains, knowing these things practically solves the task, since the actual motion between a specified pick location and place location is effectively abstracted away. That is, if I did produce a bounding box of the place location and the considered object, it’s easy to determine the correct action. In that sense, the considered type of “supercharging” supplemental data is especially helpful for 2D action spaces. If the model receives lots of that data that helps it learn these concepts, then it makes sense that it would only need a bit of robot data to use these concepts (and their underlying representations) to learn the considered robotics tasks with limited demonstration data. To that end, I think the paper demonstrates this thoroughly.
>
> However, this is critically very different from the more general 6DoF visuomotor manipulation that has recently gotten a lot of attention. While it’s still useful to ground the instruction language (its referent objects, motions, locations, and spatial relations) in the visual scene, knowing that stuff doesn’t solve the task: given the instruction “place the <object> in the bowl,” even with perfect localization and grounding, the policy needs to take the finer-grained actions to grasp the object, move it sufficiently high up to avoid knocking the bowl over, move over the top of the bowl, and then let go. These are all considerations which are intrinsically missing from the 2D case.
>
> Just to reiterate: I now understand that your policies generally do not receive (ground truth or predicted) locations of objects in the scene, and that the locations are referring to objects in the reference image. What I am saying is that it makes sense that having the latter as effectively an auxiliary training objective enforces that the model knows to look for a particular object in the scene, which is an extremely strong and helpful learned bias for 2D action space domains in particular.
>
> This also leads into another point: I appreciate that you have clarified the RT-2 baseline’s expressivity (though I still think it makes sense to increase the number of tokens to ensure it has at least degree-level accuracy – you can’t express -180 to 180 degrees in one token with 256 bins, and the number of bins is not an intrinsic part of RT-2).
>
> However, it seems to be a fundamentally inappropriate baseline in this situation, even putting aside any possible design choice issues. Namely, in its original 6DoF setting, each trajectory contains tens or hundreds of observation-action pairs, and RT-2 (and similar VLAs) are trained to map each image to its corresponding 6DoF action. Thus, it’s a setting with many “dense” observations per task – the robot sees many examples of “appropriate” observation-action mappings for each trajectory (simply due to trajectories having many steps). This higher-frequency closed-loop paradigm is necessary for a lot of challenging tasks, and likewise gives rise to e.g., the empirical retrying behavior observed in RT-2, OpenVLA, etc.
>
> [Continued]

---

> > ### Comment · Reviewer_3YdQ · 2024-11-26
> > **Discussion Period Response (2 / 2)**
> >
> > The above is the setting that RT-2 has demonstrably excelled in. It makes sense then, that in a very low-data and (comparatively) low episode length domain like your version of VIMA bench, it would struggle, and that enforcing certain kinds of learned semantic, visual, and relational biases would be the only way for generalization from such little data.
> >
> > (You could say that the above is a weakness of RT-2 that’s worth addressing. While I might agree, it’s unclear if it matters much, since many tasks people care about DO have long episodes and fine-grained actions like with RT-2. In either case, it seems like considering only 2D action spaces is a much larger limitation.)
> >
> > However, in that case, it seems like a better baseline would be a sufficiently engineered Code as Policies system. While you have shown that GPT-4o cannot perform the tasks well zero-shot, the baseline seems quite simple – it does not use, e.g., the object detectors and other functionalities present in the original Code as Policies paper, as well as integration with all the reference image stuff (I’m not sure if you guys tested that in your baseline, since the prompt in the appendix doesn’t really mention the reference images, but I would imagine having GPT-4o or some other VLM take in the reference images with some grounding language would make it very good at finding the object in the observation. Likewise, some [MOKA](https://arxiv.org/abs/2403.03174)-style keypointing would probably also help, if you want all detections to be from GPT-4o). At the very least, a baseline like that would serve as a better approximation of the types of systems people actually use for this setting. It seems like RT-2 is simply not an informative baseline to compare against.
> >
> > I recognize that 2D domains like VIMA bench have certainly been used in popular VLM + robotics papers before, including SayCan (for its sim experiments), the aforementioned Code as Policies, and CLIPPort. However, this domain is very limited – given how many recent papers have demonstrated that VLM pretraining is very helpful for general 6DoF manipulation, I think the idea would be a much more impactful and salient if it similarly showed the “supercharging” is a good idea for the aforementioned more complex domains. As it stands, it seems that the demonstrated visuo-semantic generalization and transfer is intuitively and empirically helpful in the 2D cases, but it is does not provide sufficient evidence that it is that impactful or beneficial for the 3D and 6DoF case considered by many major robotics applications.
> >
> > We are not supposed to request additional experiments at this time (and I likewise think it would be extremely unreasonable to you to test this in a rebuttal anyway). However, given the choice of benchmark, I do not believe it has provided sufficient evidence of the extent to which this approach would be useful for more complex robotics settings. It’s totally possible that LLaRA could apply to that case, but as it stands, I do not believe it’s been demonstrated, motivating my score. Again, I greatly appreciate all the engagement and the provided experiments.

---

> > > ### Author Response · Authors · 2024-12-03
> > > **New results on 3D manipulation**
> > >
> > > Thanks for clarifying that the criticism is more toward the VIMA dataset and its action output formulation, rather than the LLaRA model itself
> > >
> > > In order to address the concern of the reviewer further, utilizing 3~4 days, we attempted to extend LLaRA to a subset of the 3D manipulation benchmark LIBERO [1]. Similar to LLaRA on VIMA, we used the camera calibration to align our XY action outputs with the normalized image coordinates. On top of this, we introduced an additional single scaled value (in the action space) to represent the distance change between the robot gripper and the camera (denoted by the tag `<d>`), and two additional scaled values to represent the roll and pitch rotations of the gripper (denoted by the tag `<r>`).
> > >
> > > We fine-tuned Qwen2-VL-2B-Instruct[2] for four epochs using the inBC data format derived from LIBERO-90. During data preparation, we downsampled the dataset by consolidating multiple actions into single-step actions and utilizing only one camera view, which makes this a much more challenging setting compared to the original dataset paper or the others. Remarkably, our model, trained on just 5.7% of the images (77277 out of 1338086) from a single camera view, achieves performance comparable to the baselines, which use two camera views and are trained on 100% of the images. The average performance of our method over 50 episodes is summarized in the table below, with baseline performance values directly referenced from the original paper.
> > >
> > > |                                        | Resnet-T | Resnet-RNN | ViT-T | LLaRA |
> > > |----------------------------------------|----------|------------|-------|-------|
> > > | close the top drawer of the cabinet    | 45       | 45         | 60    | 40    |
> > > | close the bottom drawer of the cabinet | 75       | 40         | 55    | 78    |
> > > | close the microwave                    | 20       | 10         | 10    | 20    |
> > >
> > > Despite the limited time during the discussion phase, very limited training data, and minimal engineering or parameter tuning, our method achieves performance comparable to the baselines, demonstrating its ability to successfully transfer its success on the VIMA environment to more 3D manipulation tasks.
> > >
> > > **Reference**
> > >
> > > [1] Liu, Bo, et al. "Libero: Benchmarking knowledge transfer for lifelong robot learning." Advances in Neural Information Processing Systems 36 (2024).
> > >
> > > [2] Wang, Peng, et al. "Qwen2-vl: Enhancing vision-language model's perception of the world at any resolution." arXiv preprint arXiv:2409.12191 (2024).

---

> > > ### Author Response · Authors · 2024-12-03
> > >
> > > **Regarding the criticism on the VIMA setup**
> > >
> > > We believe VIMA is a well respected environment and dataset, and arguing against the setup of this established environment itself is beyond the scope of the paper. We do agree that this is a **different** setup than other 6 DoF setups, with different axes of difficulties. Research focusing on these different axes will be complementary, and we believe neither of these axes should be discounted.
> > >
> > > By no means RT-2-like approaches are the ultimate solution for the VIMA setup, and we are experimentally confirming this. They significantly suffer as we have shown, meaning that something’s missing in these approaches and formulation. If this setup is so easy, then such RT-2-like approaches should be obtaining 100% accuracy easily with 100% of the data being provided. We are showing it is not the case because they lack visual representation transfer.
> > >
> > > We need to conduct research in this VIMA-like setup to overcome such challenges. They are complementary and their findings need to be combined in future research. We believe killing a paper simply because it focuses on the challenges that previous models cannot address is quite unfair. How would the research community progress if we cannot release a paper solely because it touches the different aspects of the challenges? FYI: the original VIMA paper was published at ICML 2023, which is a machine learning conference just like ICLR. It is really difficult for us to accept that our ICLR submission needs to be discounted because we are building on top of the dataset and the setup from the same machine learning community.
> > >
> > >
> > > **Regarding the dense training data collection/setup**
> > >
> > > We tried a version of RT-2-Style where it received training data at a much higher frequency (still with image coordinate-based action space), but we empirically found that it performed identically worse than its standard frequency version. We do not believe that the failure of RT-2-Style approaches under limited training data is due to the data frequency. It is more due to its difficulty in visual representation transfer, as we have been trying to emphasize throughout our paper and the rebuttal. We need more training trajectories with really different visual configurations (not just higher frequency action data with the same scene), in order to fully train these approaches, which are often very expensive to get particularly in the real-world. We will add more discussions on this in the final version of the paper.
> > >
> > > **Code as Policies baseline**
> > >
> > > It will be an interesting baseline to explore further, but we also want to point out that it requires external object detectors and other functionalities **during the inference**. We emphasize once more that our model is a standalone model not requiring any object detector or other components for the inference. We are doing a fair comparison to RT-2-Style and GPT-4o, as they do not rely on any other external module identical to our setup. Running the external modules and running LLMs on top of them also increases the number of tokens they need to process significantly during the inference, contributing to the runtime.

---

> > > ### Author Response · Authors · 2024-12-03
> > >
> > > > Is the newly trained policy only evaluated on the three real-world tasks? I suspect that it would damage performance even more if done on the VIMA bench task you gave as an example in your response, where there are three intermediate steps that would all need these general, freeform language descriptions. It does seem to already degrade performance in the real world tasks you evaluated on, but since all three seem to be solvable in one step and just requiring knowing the location of a single target object in the scene (as well as the bowl or plate), I suspect that the freeform language is also particularly usable or beneficial in this case.
> > >
> > >
> > > Following the request, we report D-inBC(L) on VIMA. D-inBC(L) performed equally well as D-inBC on VIMA. D-inBC(L) generally outperforms inBC, leveraging the additional information from the reference image.
> > >
> > > | Method                 | L1 | L2 | L3 | Avg |
> > > |------------------------|----------|----------|----------|---------|
> > > | inBC + Aux(D)         | 59.2     | 58.8     | 52.1     | 56.7 |
> > > | D-inBC(L) + Aux(D)    | 61.9     | 54.2     | 55.4     | 57.2 |
> > > | D-inBC + Aux(D)    | 62.3      | 57.7     | 52.1    | 57.3 |
> > >
> > > > Related: Did RT-2-style have the same expressivity as your approach? Since you're outputting <x, y> coordinates as string representations of floating points, it seems like your approach needs more tokens than what RT-2-style policies are even allowed to output. Since you output the rotation in degrees (from -359 to 359), it seems unfair to give RT-2 only a single rotation token with 256 bins. That seems to artificially make RT-2-style policies less expressive than yours.
> > >
> > > As suggested by the reviewer, we modified the RT-2 setting by using two tokens to represent rotation, resulting in 65,536 options to encode rotation. This new configuration, named RT-2 Style (2R), introduces a significantly larger action space compared to our method. The model performances trained on the VIMA-8K dataset are summarized below:
> > >
> > > | Method          | L1   | L2   | L3   |
> > > |------------------|-------|-------|-------|
> > > | RT-2 Style      | 3.8   | 3.1   | 1.7   |
> > > | RT-2 Style (2R) | 1.9   | 2.3   | 1.7   |
> > > | inBC            | 57.3  | 46.2  | 42.9  |
> > >
> > > As shown above, an additional token for rotation does not significantly change the performance of the RT-2 Style baseline.

---

### Official Review · Reviewer_HVHL · 2024-10-17

**Soundness:** 3
**Presentation:** 3
**Contribution:** 3
**Rating:** 6
**Confidence:** 4

**Summary:**

The authors introduce a mechanism for (1) leveraging VLMs for robot task completion and (2) augmentation of trajectories with object/task auxiliary data.

**Strengths:**

They present these results on both a simulated and physical benchmark. The paper also includes a substantial set of results in the appendix.

**Weaknesses:**

I'm concerned about the generality of the work, based in large part on the inconclusive trends presented.  I'll provide a series of questions below but my primary concern is that it's unclear on training.  It appears that training either has no effect or hurts performance in most cases.  There's some improvement when two epochs are run (in some conditions, but not all).  Similarly, how much data and when/where/why it helps are unclear.  This doesn't detract from the fact that a nice system was deployed but it does make it difficult to determine where this research leads in the future.

**Questions:**

1. See weakness discussion
2. There's work on the limitations of numerical representations in transformers.  Presumably this means we should see failure cases due to misunderstanding in this work, can the authors provide those examples or justify why they wouldn't occur in this domain?
3. Are the augmentations validated for correctness?
4. Can Fig 5 be augmented with another plot that show how many training samples (not just trajectories) each approach has?
5. Why was 12% chosen (e.g. Table 1), is that simply the first time the model outperforms VIMA? Do the results plateau afterwards?
6. Fig 6, small confirmation that the X-axis means "2x, 4x, ..." ?
7. L434: What's the coverage on 0-180 [sort of related to question #2]
8. JT is worse than FT, why? -- similar confusion or missing justifications for Tables 18/19

---

> ### Author Response · Authors · 2024-11-21
> **(1/2) Comment for Reviewer HVHL**
>
> We thank the reviewer for the insightful comments and for highlighting important areas for clarification. Below, we will try our best to address the points raised:
>
> > it's unclear on training. It appears that training either has no effect or hurts performance in most cases.
>
> To clarify, our method adopts a pretrained VLM, and we then fine-tuned it with our robotics dataset. What we show in this paper is that (1) training such a pretrained VLM with the robotics data (with our action representation) enables successful robot tasks, and that (2) our self-supervised auxiliary data further benefits the robot action policy learning.
>
> Our model trains quickly, within 1\~2 epochs. We observe that the initial pretrained VLM (i.e., LLaVA 1.5) obtains 0\~1% success rate on VIMA experiments without our robot data training (Table 12 in Appendix), and it converges to 80\~90% after 1\~2 epochs (Figure 5 and Table 19 in the appendix). In addition, Figure 6 and Table 14-19 show that training with more auxiliary data further enhances the model.
>
> We will provide more detailed training curves in the final version of the paper.
>
> > Similarly, how much data and when/where/why it helps are unclear.
>
> We provide detailed ablation studies in Figure 6 and Table 2, demonstrating that more auxiliary data generally improves performance. Additional ablation results in different scales, exploring different auxiliary data settings, are available in Tables 14-19 in Appendix, where we mention the relative size to the BC dataset of each auxiliary dataset.
>
> For example, in Table 14 line 1588 we have 800 episodes from VIMA which contain 1403 samples; 1 in  Loc column means we also have 1403 samples from the localization dataset in the training set. We also attach the exact number of samples in each dataset in the below table. Note that these numbers reflect only the number of examples used to train the model, and they are all generated from the same amount of original data without additional action labels or observations to make a fair comparison. We thank the reviewer for the suggestions and we will include them in the revised paper.
>
> | Original dataset size | VIMA-0.8k | VIMA-8k | VIMA-80k |
> |-----------------------|-----------|---------|----------|
> | All methods           | 1403      | 13922   | 139587   |
>
> | Total dataset size with auxiliary data | VIMA-0.8k | VIMA-8k   | VIMA-80k    |
> |----------------------------------------|-----------|-----------|-------------|
> | RT-2 Styles        | 1403      | 13922     | 139587      |
>  | LLaRA w/o Aux  | 1403      | 13922     | 139587      |
> | LLaRA w/ Aux (D)                       | 9821      | 97454     | 977109      |
>
> In summary, we find that aligning robot actions to image coordinates and leveraging self-supervised auxiliary data improves performance significantly. Our systematic study of six auxiliary datasets confirms that our method outperforms baselines, even with considerably less training data.
>
> > There's work on the limitations of numerical representations in transformers. Presumably this means we should see failure cases due to misunderstanding in this work, can the authors provide those examples or justify why they wouldn't occur in this domain?
>
> We acknowledge that transformers can struggle with numerical representations. To mitigate this, we ensure sufficient exposure to numerical text through our auxiliary datasets. This is inspired by the instruction tuning in computer vision such as LLaVA [1]. Our results demonstrate that this approach allows the transformer to develop the required numerical awareness. We will clarify this further in the revised paper.
>
> > Are the augmentations validated for correctness?
>
> All auxiliary data is generated using template-based methods, with exact templates detailed in Tables 21-32 in Appendix. These templates ensure correctness in our data generation process, as it’s a rule-based deterministic process)
>
> > Can Fig 5 be augmented with another plot that show how many training samples (not just trajectories) each approach has?
>
> Certainly, please check the anonymous link below. Notice that these numbers reflect only the number of examples used to train the model, and they are all generated from the same amount of original data (which is used in the original Fig. 5) without additional action labels or observations to make a fair comparison.
>
> https://drive.google.com/file/d/1qPcZc1ICH9EKbcKPWFuFp-whQY65oRS0/view?usp=drive_link

---

> > ### Comment · Reviewer_HVHL · 2024-11-25
> > **Thank you! Example mismatch**
> >
> > I really appreciate the inclusion of this new figure, but I think this confirms my concern/confusion.  We are not comparing methods with the same number of examples? The largest dataset RT-2 model corresponds to the mid-point of several LLaRA results and has quite the positive upward trend.  Is it possible to get a true head-to-head comparison by training data?  I *think* your results will still come out strongest?

---

> > > ### Author Response · Authors · 2024-11-27
> > > **Kind Reminder**
> > >
> > > Dear reviewer HVHL,
> > >
> > > This is a kind reminder that we tried our best to carefully address the concerns you raised in your review and provide the revised figure you suggested.
> > > Please let us know what you think. If there are any remaining concerns or additional points/questions you'd like us to address further, please also let us know.
> > >
> > > Thank you once again for your valuable feedback and time.
> > >
> > > Best regards,
> > >
> > > Authors of paper 8021

---

> > > > ### Comment · Reviewer_HVHL · 2024-11-28
> > > > **Thank you for engaging and sharing updates**
> > > >
> > > > I think the new plots really help clarify the comparisons and trends.  Future iterations of this work will be helped by inclusion of these and responses to all the reviewers as they have greatly clarified the motivation and key contributions of the work.

---

> > > > > ### Author Response · Authors · 2024-12-03
> > > > >
> > > > > We thank the reviewer once more for the constructive feedback. We will include the added illustrations, discussions, and clarifications in the final version of the paper, and we hope the reviewer can consider these when finalizing the decision.
> > > > >
> > > > > If there are any further concerns or clarifications you want us to address, please just let us know.

---

> ### Author Response · Authors · 2024-11-21
> **(2/2) Comment for Reviewer HVHL**
>
> > Why was 12% chosen (e.g. Table 1), is that simply the first time the model outperforms VIMA? Do the results plateau afterwards?
>
> The 12% was chosen empirically as a hyperparameter when constructing the datasets (80k out of 660k episodes). Based on the trends shown in Figure 5, we believe the results may plateau at some point. However, we did not test this as (1) the current model achieves satisfactory performance - our method trained on only 12% data performs better than other baselines like VIMA or RT-2 Style trained on 100% data, and (2) running experiments with 100% data will be computationally expensive.
>
> > Fig 6, small confirmation that the X-axis means "2x, 4x, ..." ?
>
> Yes, that is correct. Thank you for pointing this out, and we will ensure it is clarified in future versions of the paper.
>
> > L434: What's the coverage on 0-180 [sort of related to question #2]
>
> The rotation angle in Task 2 is from the following set: {30, 90, 120, 180}.
>
> > JT is worse than FT, why? -- similar confusion or missing justifications for Tables 18/19
>
> In the joint training (JT) setting, the model is required to learn across two domains (simulated and real-world), which introduces a significant domain gap. This likely prevents the model from learning an optimal distribution for either domain. In contrast, the fine-tuning (FT) setting focuses solely on the real-world domain, simplifying the learning process. We will clarify this in the final version of the paper. Thanks.
>
> We are happy to address any further questions or concerns the reviewer may have. Thank you for your thoughtful feedback.
>
> **Reference**
>
> [1] Liu, Haotian, et al. "Visual instruction tuning." Advances in neural information processing systems 36 (2023).

---

> ### Author Response · Authors · 2024-11-25
>
> We are sorry for the confusion that the previous figure does not include the results presented in Table 9. We also added a new graph that averages the three levels of the tasks (L1, L2, and L3):
> https://drive.google.com/file/d/1OIcUacEEEGiJYOK7sxINaoJ1uW1LASKI/view?usp=drive_link
>
> As shown in the new figure and Table 9, our model, even only trained with 12% data, can outperform RT-2 style that was trained on 100% data.
>
> We would like to address another potential misunderstanding: as stated in Section 5, our auxiliary data is generated in a self-supervised manner using the exact same amount of robot behavior cloning data, without relying on any additional action labels or observations. All head-to-head comparisons have been conducted using the same original dataset size.
>
> What we present in this paper is a model that can be trained much faster with less number of robot episodes, compared to the conventional method like RT-2, and we are introducing the use of 'instruction style data’ and 'self-supervised auxiliary data' to enable that in robot learning.
>
> We hope this clarification resolves your concerns, and we are happy to address any further questions you may have.

---

### Official Review · Reviewer_wHkn · 2024-11-01

**Soundness:** 3
**Presentation:** 2
**Contribution:** 2
**Rating:** 6
**Confidence:** 4

**Summary:**

The paper aims to transform behavior cloning data into a form that is more accessible for VLM training, enabling the VLM to leverage its pretraining for downstream tasks. The approach involves converting the BC dataset into image-text data and having the model output 2D coordinates rather than the 3D outputs. The study introduces curated datasets to assist in 2D grounding, action prediction, next observation prediction, and understanding spatial relationships between objects. The model is evaluated against RT-2 and VIMA, showing performance gains, particularly in low-data regimes.

**Strengths:**

- The paper reformulates BC data into an image-text format compatible with VLMs. This is not a new idea (RT-2), but they do introduce a new output space and auxiliary objectives.
- Restricting output to 2D coordinates sets the model apart from prior 3D-focused models like RT-2, aligning well with tasks where 3D positioning is unnecessary. Various curated datasets support 2D grounding and action prediction, aiding spatial and relational understanding.
- The model outperforms versions of VIMA and RT-2, particularly in low-data settings.
- They show gains on simulation and real tasks.

**Weaknesses:**

- It was not directly clear to me which components contribute to performance in the RT-2 and VIMA comparison. RT-2 also uses auxiliary tasks and internet data; it's unclear how the proposed auxiliary tasks compares. An ablation of the auxiliary tasks used in RT-2 versus LLaRA would help isolate the contribution of each component.
- The contributions of each dataset are unclear; a more interpretable format is needed to highlight key influences on performance.

**Questions:**

- RT-2 also co-trains on auxillary tasks and internet data. How does this compare to the proposed approach?
- For the comparison with VIMA, can additional insights be provided on which specific modeling choices (beyond single-view versus dual-view inputs) contribute to LLaRA’s superior performance?
- Can the comparisons with RT-2 and VIMA be unified to allow for a clearer, combined comparison?
- More analysis on Figure 6 is needed to understand the dataset contribution. Which datasets contribute most? A visualization or table that clearly shows the impact of each dataset on different aspects of model performance would be helpful.
- Is the primary difference between LLaRA and the RT-2 baseline solely the output token space?

---

> ### Author Response · Authors · 2024-11-21
> **Comment for Reviewer wHkn**
>
> We thank the reviewer for the comments and for recognizing our major contributions, including the introduction of a new action space aligned to image coordinates, the introducing auxiliary data in a self-supervised manner, and the outstanding performance of our method especially with limited data. We would like to further address the reviewer’s concerns and provide the following clarifications:
>
> > It was not directly clear to me which components contribute to performance in the RT-2 and VIMA comparison. RT-2 also uses auxiliary tasks and internet data; it's unclear how the proposed auxiliary tasks compares. An ablation of the auxiliary tasks used in RT-2 versus LLaRA would help isolate the contribution of each component.
>
> One of the contributions of our paper is to provide a way to generate self-supervised auxiliary data given robot trajectories, in addition to the original behavior cloning data. This has the effect of enhancing the existing robot data, which RT-2 has not attempted using for its training. RT-2 uses Internet VQA data, which is orthogonal to our self-supervised robotics data.
> With our experiments (Figure 5, 6, Table 9 and Table 14-19 in the appendix), we are showing the benefit of adding such auxiliary data for the robot model training on top of a pretrained VLM (trained with internet VQA data).
>
> > The contributions of each dataset are unclear; a more interpretable format is needed to highlight key influences on performance.
>
> > More analysis on Figure 6 is needed to understand the dataset contribution. Which datasets contribute most? A visualization or table that clearly shows the impact of each dataset on different aspects of model performance would be helpful.
>
> In the main paper, we show an ablation study in Figure 6 and Table 2, where we find that, in general, as more auxiliary data is added, performance improves. We demonstrate that the model benefits from all these different types of auxiliary data we generate and provide. In addition to that, in Appendix, we have provided more detailed ablations across datasets presented in Tables 16-19. These results generally demonstrate the benefit of auxiliary data, although the gains may decrease in certain settings as the training data volume increases. We will revise the paper to include more details regarding such ablation in the main paper.
>
> > For the comparison with VIMA, can additional insights be provided on which specific modeling choices (beyond single-view versus dual-view inputs) contribute to LLaRA’s superior performance?
>
> Yes. In Appendix B, we have provided more detailed ablations on various modeling choices, including action history, multi-step planning, multi-image inputs, the detector, and the prompt template. Our current best practice design includes action history in the query text, asks VLM to plan multiple future steps in a single conversation, and puts only one image (which is the current observation) in the conversation. We will clarify this further in the revised paper.
>
> > Can the comparisons with RT-2 and VIMA be unified to allow for a clearer, combined comparison?
>
> We provide a more detailed comparison between our method, RT-2, and VIMA in Appendix B.2, Table 9. Table 9 shows that LLaRA learns much faster with much less data (12% vs. 100%), and  obtains higher performance despite that. It would be great if the reviewer could check and let us know whether this resolves the concern. We will revise the paper to include some of these results in the main paper accordingly.
>
> > Is the primary difference between LLaRA and the RT-2 baseline solely the output token space?
>
> There are two major differences. As noted by the reviewer, one key distinction is the output token space: LLaRA aligns action in natural language with image coordinates, whereas RT-2 employs specialized discrete action tokens. We confirm that this enables much faster training. But in addition to this, our work also uniquely explores the impact of auxiliary robot data generated from the existing behavior cloning dataset without relying on external datasets, an approach that, to our knowledge, has not been explored in prior robot learning works with VLMs.
>
> We are happy to address any further questions or concerns and would welcome additional feedback or suggestions for improving our work.  Thank you.

---

> ### Comment · Reviewer_wHkn · 2024-11-25
> **Thank you for the response**
>
> I thank the authors for their comments and clarifications regarding the auxiliary objective ablations and modeling choices. I do appreciate the improvement in accuracy and efficiency over baselines on the tabletop tasks shown. However, I still have concerns related to the generalizability, scalability, and novelty compared to RT-2 with regard to the auxiliary objectives, given it is a main contribution. I have addressed a specific comment below.
>
> > One of the contributions of our paper is to provide a way to generate self-supervised auxiliary data given robot trajectories, in addition to the original behavior cloning data. This has the effect of enhancing the existing robot data, which RT-2 has not attempted using for its training. RT-2 uses Internet VQA data, which is orthogonal to our self-supervised robotics data.
>
> I believe the VQA and captioning data RT-2 uses is readily available on the internet and they show co-training on it does enhance the existing robotics performance, while the auxiliary objectives defined require precise object detection and may not provide as large a benefit in more realistic environments and for tasks outside of tabletop pick and place. It also seems highly specific to tasks that may not need 3D understanding, given the auxiliary tasks are defined in image coordinates.

---

> > ### Author Response · Authors · 2024-11-25
> >
> > We thank the reviewer for the feedback.
> >
> > What we would like to emphasize is that, even for such table top settings, RT-2 style training fails to provide a reasonable accuracy when we don't have a sufficient amount of robot training data. Figure 5 in the paper shows such trend (measured with the VIMA tasks). When 100% of the robot data is provided, the RT-2 style training converges to a reasonable accuracy (still lower than ours due to the absence of the self-supervised training data as shown in Table 9), but when the amount of training data is low (e.g., 8k episodes which is 1.2% of the data), the RT-2 style suffers significantly.
> >
> > We need a different solution than RT-2, if we want to train a working model under such setting.
> >
> > This is the reason why we are bringing the concept of "visual instruction tuning" and "self-supervised auxiliary data" from the computer vision community into the robot learning. The previous RT-2 model could be viewed as a robotics successor of the older style VLM learning frameworks like PaLI and PaLM, which are already ~2 years old. Since then, there has been a good amount of research progress in computer vision. Researchers discovered that formulating data in the conversation style greatly benefits the model training, and the use of self-supervision could further improve the VLM model training. Such (newer) models include LLaVA, Llama3, Qwen2-VL, BLIP-3, and so on. For instance, the use of "object bounding boxes" (or "grounding") in data generation in LLaVA, Llama3, and Qwen2-VL shows how the auxiliary tasks are benefitting computer vision model training. Similar to us, they take advantage of a pretrained object detector, and the VLMs trained with such data shows a meaningful improvement. The use of pre-trained object detector has been confirmed to be ok and beneficial in this context.
> >
> > We wanted to replicate this success in robot action policy learning. Our VLM is already pretrained with an internet VQA data (i.e., it already benefits from them), and we are doing the robot learning on top of them. We show that the instruction style robotics data is beneficial, and how adding robotics auxiliary data further helps. Furthermore, we introduce robot learning specific auxiliary data, including action prediction and future predictions, formulated in the VLM instruction tuning style, which we believe are our original contributions.
> >
> > We do agree that the action space we focus in this paper is translations and rotations in 2D image coordinates. This is our design to choice to enable faster representation transfer from a computer vision-based VLM to the robot action model. However, we would like to emphasize that VIMA and our real-robot settings have a totally different axis of difficulty: visual representation transfer with limited data, which conventional approaches like RT-2 suffer. We believe the new findings of our paper are orthogonal to other VLA models focusing on different aspects of the problem such as 3D manipulation. What we hope to bring with this paper is the importance of visual representation transfer in robot learning and how to do so with more modern techniques.

---

> > > ### Author Response · Authors · 2024-12-03
> > > **New results on 3D manipulation**
> > >
> > > In order to address the concern of the reviewer further, utilizing 3~4 days, we attempted to extend LLaRA to a subset of the 3D manipulation benchmark LIBERO [1]. Similar to LLaRA on VIMA, we used the camera calibration to align our XY action outputs with the normalized image coordinates. On top of this, we introduced an additional single scaled value (in the action space) to represent the distance change between the robot gripper and the camera (denoted by the tag `<d>`), and two additional scaled values to represent the roll and pitch rotations of the gripper (denoted by the tag `<r>`).
> > >
> > > We fine-tuned Qwen2-VL-2B-Instruct[2] for four epochs using the inBC data format derived from LIBERO-90. During data preparation, we downsampled the dataset by consolidating multiple actions into single-step actions and utilizing only one camera view, which makes this a much more challenging setting compared to the original dataset paper or the others. Remarkably, our model, trained on just 5.7% of the images (77277 out of 1338086) from a single camera view, achieves performance comparable to the baselines, which use two camera views and are trained on 100% of the images. The average performance of our method over 50 episodes is summarized in the table below, with baseline performance values directly referenced from the original paper.
> > >
> > > |                                        | Resnet-T | Resnet-RNN | ViT-T | LLaRA |
> > > |----------------------------------------|----------|------------|-------|-------|
> > > | close the top drawer of the cabinet    | 45       | 45         | 60    | 40    |
> > > | close the bottom drawer of the cabinet | 75       | 40         | 55    | 78    |
> > > | close the microwave                    | 20       | 10         | 10    | 20    |
> > >
> > > Despite the limited time during the discussion phase, very limited training data, and minimal engineering or parameter tuning, our method achieves performance comparable to the baselines, demonstrating its ability to successfully transfer its success on the VIMA environment to more 3D manipulation tasks.
> > >
> > > **Reference**
> > >
> > > [1] Liu, Bo, et al. "Libero: Benchmarking knowledge transfer for lifelong robot learning." Advances in Neural Information Processing Systems 36 (2024).
> > >
> > > [2] Wang, Peng, et al. "Qwen2-vl: Enhancing vision-language model's perception of the world at any resolution." arXiv preprint arXiv:2409.12191 (2024).

---

> > ### Author Response · Authors · 2024-11-27
> > **Kind Reminder**
> >
> > Dear reviewer wHkn,
> >
> > This is a kind reminder that we tried our best to carefully address the concerns you raised in your review.
> > Please let us know what you think. If there are any remaining concerns or additional points/questions you'd like us to address further, please also let us know.
> >
> > Thank you once again for your valuable feedback and time.
> >
> > Best regards,
> >
> > Authors of paper 8021

---

> ### Comment · Reviewer_wHkn · 2024-12-03
> **Thank you**
>
> Thank you for the clarifications. I see the contributions in terms of performance with limited data and the inclusion of localization- and action-based auxiliary tasks beyond just captioning and QA tasks. I have raised my score. The limitations—such as the somewhat limited spatial relationships defined, the 2D objectives that may lead to 3D failures, the need for different auxiliary objectives for more complex manipulation tasks, and further analysis of the policy's failure modes—should be discussed in greater detail.

---

### Official Review · Reviewer_aFtC · 2024-11-04

**Soundness:** 3
**Presentation:** 3
**Contribution:** 3
**Rating:** 6
**Confidence:** 4

**Summary:**

The manuscript proposes a framework for performing auxiliary instruction-tuning of LLaVA, using a robotics behavior cloning dataset that has been converted into a conversation-style instruction-tuning dataset, with the goal of enabling the model to better understand spatiotemporal context.

**Strengths:**

The pursuit of robotics-oriented pretext tasks for training large-capacity open source models remains quite compelling.

**Weaknesses:**

Section 2, Section 6: The manuscript introduces relevant contemporary approaches — some with strong inductive biases — but the experimental comparisons with the method proposed by the manuscript remain shallow. I would like to see some direct comparisons with existing VLAs in the experiments section, in addition to ablations on, e.g., different action space representations (e.g., versus RoboPoint).

Section 2 (L146-148): The manuscript states, "Moreover, all the aforementioned studies lack of comprehensive investigation into the methods for generating auxiliary datasets from existing robot data, as well as the implications of integrating such datasets." OpenVLA uses mixtures of exisisting OXE component datasets, for a next action token-prediction pretext.

Section 5: The dependency on high-quality behavior cloning data may limit LLaRA’s performance in environments significantly different from the training data. Other methods cited by the manuscript do not share this limitation.

Section 5: The pretext task proposed by this manuscript, which relies on object locations (in a tabletop coordinate system) and joint rotations will be less general than, e.g., the next-token prediction pretext and action tokenization proposed by OpenVLA. The manuscript should better motivate the reduced scope, whereas many of the models mentioned in Section 2 pursue general training and execution paradigms: across domains, embodiment, and task families.

&nbsp;

**Questions:**

No additional questions — please see above.

---

> ### Author Response · Authors · 2024-11-21
> **(1/2) Comment for Reviewer aFtC**
>
> We thank the reviewer for their insightful comments and suggestions, which have greatly helped us improve the manuscript. Below, we address the specific concerns raised:
>
> > Section 2, Section 6: The manuscript introduces relevant contemporary approaches — some with strong inductive biases — but the experimental comparisons with the method proposed by the manuscript remain shallow. I would like to see some direct comparisons with existing VLAs in the experiments section, in addition to ablations on, e.g., different action space representations (e.g., versus RoboPoint).
>
> We appreciate the reviewer highlighting contemporary approaches. The primary reason we did not directly compare to these methods is that they target different action spaces rather than VIMA, which is the primary setting of our work. To address this concern:
>
> We would like to highlight the experiments in Tables 5 and 9, where we perform ablations, such as RT-2 Style (I), which employs special tokens for actions aligned to image coordinates.
>
> Additionally, Tables 14–19 in the appendix provide extensive ablations on the use of auxiliary data.
>
> RoboPoint, on the other hand, is focused on predicting spatial affordances, giving outputs as 2D point coordinates. However, since it cannot provide key information like where exactly to pick up an object or how to handle its rotation, it’s hard to turn those predictions into a workable robot control policy in either VIMA or our real-world robot setting. That’s why we initially didn’t include it as a baseline for comparison.
>
> To best address the reviewer’s concern, we adopt and modify the pretrained RoboPoint-Vicuna-1.5-13B model for VIMA evaluation, by making two conversations per step. For the first conversation, we ask where the robot should pick and in the second conversation, we ask where the object should be placed. The performance is listed below:
> | Method | L1  | L2 | L3 |
> |-----------------------|-----------|---------|----------|
> | RoboPoint-Vicuna-1.5-13B         |  0.8 %   | 1.5%    |  0.0% |
>
> We are able to observe that the pretrained model provided by RoboPoint is unable to handle the tasks in VIMA.
>
> > Section 2 (L146-148): The manuscript states, "Moreover, all the aforementioned studies lack of comprehensive investigation into the methods for generating auxiliary datasets from existing robot data, as well as the implications of integrating such datasets." OpenVLA uses mixtures of exisisting OXE component datasets, for a next action token-prediction pretext.
>
> We would like to clarify that both LLaRA and OpenVLA employ next-token prediction as a pretext task. However, **LLaRA goes beyond this by leveraging a self-supervised auxiliary dataset**, incorporating tasks such as object localization and detection, future/action prediction, and spatial-temporal relationship understanding. In contrast, OpenVLA relies solely on next-token prediction and behavior cloning datasets. Moreover, OXE datasets, which OpenVLA is trained on, are specifically formatted for only behavior cloning datasets.
>
> Our approach, on the other hand, enriches existing datasets by generating auxiliary data in a self-supervised manner, without requiring additional action labels or observations. To the best of our knowledge, this self-supervised augmentation strategy has not been explored in prior work. Such a strategy can potentially further benefit multiple datasets in OXE as well, which we plan to contribute in the future.
>
> > Section 5: The dependency on high-quality behavior cloning data may limit LLaRA’s performance in environments significantly different from the training data. Other methods cited by the manuscript do not share this limitation.
>
> We would like to clarify that the data introduced in Section 5 is automatically generated by our framework from existing behavior cloning data, without requiring additional action labels or observations. They are not really (high-quality) behavior cloning data, but rather auxiliary data similar to self-supervised learning data in computer vision we automatically generate.
>
> We want to emphasize once more that we are using the same amount of behavior cloning data as all the other approaches.
> Additionally, the zero-shot transfer experiment in Table 3 demonstrates that our method, despite being trained on limited simulated data, performs reasonably well in real-world scenarios with unseen objects, highlighting its robustness.

---

> ### Author Response · Authors · 2024-11-21
> **(2/2) Comment for Reviewer aFtC**
>
> > Section 5: The pretext task proposed by this manuscript, which relies on object locations (in a tabletop coordinate system) and joint rotations will be less general than, e.g., the next-token prediction pretext and action tokenization proposed by OpenVLA. The manuscript should better motivate the reduced scope, whereas many of the models mentioned in Section 2 pursue general training and execution paradigms: across domains, embodiment, and task families.
>
> We appreciate this feedback and would like to emphasize that our method builds upon the next-token prediction pretext task (and the loss). It’s just that our approach has the capability to add more auxiliary data obtained in a self-supervised manner (without any additional action labels) on top of it, in order to further benefit the training.
>
> As we also mentioned in the common post, this greatly benefits the faster training of the model with much less training data, and we believe such observation is valuable to the community to explore this direction further. We will clarify this in the paper and emphasize the contributions.
>
> In addition, while our action tokenization —using text-based image coordinate descriptions is different from RT-2—our experiments (e.g., Figure 5 and Table 9) demonstrate that this design is more data-efficient than RT-2-style baselines and requires less training time. We will emphasize them in the paper further to justify the design choices and highlight the advantages of our approach where data efficiency and training time are critical.
>
> We are happy to address any further questions or concerns and would welcome additional feedback or suggestions for improving our work.  Thank you.

---

> > ### Comment · Reviewer_aFtC · 2024-11-25
> > **Thanks for the response**
> >
> > I appreciate the authors' responses! They indeed alleviated some concerns.
> >
> > I have a few more comments/questions:
> >
> > 1. In my original review, I said: "Section 5: The dependency on high-quality behavior cloning data may limit LLaRA’s performance in environments significantly different from the training data. Other methods cited by the manuscript do not share this limitation." I am actually not satisfied with the authors' response on this point, as even the auxiliary dataset is predicated on the quality, diversity, and (cross-domain) generality of the original BC data (not to mention the model's training behavior). As the manuscript and the author responses emphasize, no semantic information is being added beyond that which already exists in the VIMA-bench data and, by extension, no limitations in the above properties are being corrected by the proposed approach. This concern was continued in the next (and last) point of my original review: many of the models mentioned in Section 2 pursue general training and execution paradigms — across domains, embodiment, and task families. In my opinion, this renders discussions about increased training efficiency in the proposed approach unconvincing and irrelevant.
> >
> > 2. The manuscript and rebuttal responses emphasize the self-supervised auxiliary data given robot trajectories as a significant part of the manuscript's contributions. Beyond just the ablation experiments, a good way to show the effectiveness of that data would be to use it to pre-train baselines as well. Without '*-RT-2-Aux-*' experiments, for example, it is difficult to fully understand what contributes to improved performance.
> >
> > 3. The manuscripts shows that the model trains quickly, e.g., within 1-2 epochs; the authors mention that Figure 6 and Table 14-19 show that training with more auxiliary data further enhances the model. This sort of optimization regime is usually a precursor to some very negative ramifications, such as overfitting, catastrophic forgetting, and negative transfer. Could the authors comment on whether/how the pre-trained model retains its capabilities, even after the proposed auxiliary pretext task?
> >
> > 4. All of the auxiliary data is generated using fixed template-based methods. This is great from the standpoint of interpretability, but not so great from the perspective of enabling diversity of the data and eliminating the possibility of dataset bias in this curation process. Could the authors comment on how diversity of the auxiliary dataset was maintained, provide intuition about how the format of the data is optimal for the underlying pretext objective, and how biases in the auxiliary dataset were minimized during curation?

---

> > > ### Author Response · Authors · 2024-11-26
> > > **(1/2) Further Clarification for Reviewer aFtC**
> > >
> > > We thank the reviewer for the response and for raising these discussions. Let us clarify these further with this post.
> > >
> > > > In my original review, I said: "Section 5: The dependency on high-quality behavior cloning data may limit LLaRA’s performance in environments significantly different from the training data. Other methods cited by the manuscript do not share this limitation." I am actually not satisfied with the authors' response on this point, as even the auxiliary dataset is predicated on the quality, diversity, and (cross-domain) generality of the original BC data (not to mention the model's training behavior). As the manuscript and the authors' responses emphasize, no semantic information is being added beyond that which already exists in the VIMA-bench data, and, by extension, no limitations in the above properties are being corrected by the proposed approach. This concern was continued in the next (and last) point of my original review: many of the models mentioned in Section 2 pursue general training and execution paradigms — across domains, embodiment, and task families. In my opinion, this renders discussions about increased training efficiency in the proposed approach unconvincing and irrelevant.
> > >
> > > We believe the biggest confusion comes from our lack of discussion on self-supervised (auxiliary) data in computer vision and how modern vision-language models (VLMs) have been benefiting from them. We apologize and we will revise the paper to include these explicitly. Let us try to clarify here:
> > >
> > > What happened with the VLM research in the past ~1 year was an extensive amount of training dataset curation including those with automatically generated "object bounding boxes" (or "visual groundings"). Researchers have discovered that using a pretrained object detection module to extract their location information and then formulating such information in natural language sentences allow much better construction of the VLM training data. For instance, LLaVA training data was generated based on the MS COCO-trained object detection and GPT4-based sentence descriptions using them. Many newer models including Llama3, Qwen2-VL, BLIP-3, and more also follow this concept of bounding box-based self-supervised data generation, all confirming their benefits. It is now being very widely accepted as a general data generation pipeline, and the generality of this concept has been confirmed across many different domains.
> > >
> > > Even though such bounding box-based auxiliary data does not require any manual human annotations, they are bringing additional semantic information to the VLM for two reasons:
> > > 1. They allow the VLM to inherit knowledge from the pre-trained object detection module (i.e., grounding knowledge)
> > > 2. The conversation-style training data using such bounding boxes allows further learning of the spatial structure of the scene (e.g., spatial relations between two objects based on their bounding box locations).
> > >
> > > All we wanted to do in this paper was to replicate this success in robot action policy learning. Such auxiliary dataset generation does not require high quality behavioral cloning dataset. Whether the dataset is high quality or not, (training) "images" are all we need to generate the auxiliary dataset. Our approach will just try to gain more knowledge (on visual grounding and spatial relations) from such boosted data and make the representation learning better, regardless of which images are in the training dataset.
> > > Our (pre)training data is VIMA. It is one of the standard datasets with BC trajectories and we are not using any other than the provided trajectories. We believe this makes our comparison to other models as fair as possible.
> > >
> > > > The manuscript and rebuttal responses emphasize the self-supervised auxiliary data given robot trajectories as a significant part of the manuscript's contributions. Beyond just the ablation experiments, a good way to show the effectiveness of that data would be to use it to pre-train baselines as well. Without '-RT-2-Aux-' experiments, for example, it is difficult to fully understand what contributes to improved performance.
> > >
> > > We thank the reviewer for the suggestion. Following the suggestion, we are working on these new '-RT-2-Aux-' experiments and we will update you in 1~2 days once they are ready.

---

> > > > ### Author Response · Authors · 2024-11-26
> > > > **(2/2) Further Clarification for Reviewer aFtC**
> > > >
> > > > > The manuscripts shows that the model trains quickly, e.g., within 1-2 epochs; the authors mention that Figure 6 and Table 14-19 show that training with more auxiliary data further enhances the model. This sort of optimization regime is usually a precursor to some very negative ramifications, such as overfitting, catastrophic forgetting, and negative transfer. Could the authors comment on whether/how the pre-trained model retains its capabilities, even after the proposed auxiliary pretext task?
> > > >
> > > > We believe convergence within 1\~2 epochs and faster training isn't necessarily a sign of overfitting, when it comes to the large-scale VLMs (with billions of parameters) and when **finetuning** on top of them. Please note that the VLM we are taking advantage of (LLaVA) was already pretrained with lots of internet VQA data. Even our VIMA pretraining on top of that is technically a finetuning on top of it.
> > > >
> > > > Lots of modern VLM literature suggests finetuning with 1\~3 epochs is an effective and efficient strategy. For instance, BLIP-3 paper finetunes with just a single epoch. LLaVA-OneVision also does the same thing with 1 epoch. The original LLaVA also trains the model with 1~3 epochs.
> > > >
> > > > Our model retaining the behavioral cloning capability has been confirmed in all of our experiments including VIMA and real-robot. We follow the exact standard evaluation protocol of VIMA, and obtain the state-of-the-art results on it in terms of the task success rate.
> > > >
> > > > > All of the auxiliary data is generated using fixed template-based methods. This is great from the standpoint of interpretability, but not so great from the perspective of enabling diversity of the data and eliminating the possibility of dataset bias in this curation process. Could the authors comment on how diversity of the auxiliary dataset was maintained, provide intuition about how the format of the data is optimal for the underlying pretext objective, and how biases in the auxiliary dataset were minimized during curation?
> > > >
> > > > This indeed is a valid concern. We are motivated by several studies (e.g., [1-4]) on vision-language models, which demonstrate that structured or template-based conversations can perform comparably to those generated by large language models (LLMs). The underlying intuition is to provide sufficient and focused information to help the model acquire the necessary self-supervised knowledge with the templating, rather than being distracted by overly diverse sentences.
> > > >
> > > > To this end, we designed a basic *set* of multiple templates to ensure the accuracy and relevance of information about different objects, while also eliminating potential hallucinations and errors introduced by LLMs. This approach also significantly reduces the computational cost. In our setting, we empirically selected 15 distinct prompts for each auxiliary task. This trade-off helps prevent the model from overfitting to specific prompts while avoiding distraction from excessive template variability.
> > > >
> > > > Meanwhile, Figure 13 in the appendix shows that our model is not biased to a particular prompt template and it is robust to different input prompts.
> > > > During the initial phase of our research, we also tried LLM-based instruction sentence data generation. We empirically found that the model accuracy is similar in both settings, confirming the observations from [1-4].
> > > >
> > > > We are happy to address any further questions or concerns the reviewer may have. Thank you.
> > > >
> > > > **Reference**
> > > >
> > > > [1] Cai, Mu, et al. "ViP-LLaVA: Making Large Multimodal Models Understand Arbitrary Visual Prompts." Proceedings of the IEEE/CVF Conference on Computer Vision and Pattern Recognition. 2024.
> > > >
> > > > [2] Gong, Tao, et al. "Multimodal-gpt: A vision and language model for dialogue with humans." arXiv preprint arXiv:2305.04790 (2023).
> > > >
> > > > [3] Jia, Baoxiong, et al. "Sceneverse: Scaling 3d vision-language learning for grounded scene understanding." European Conference on Computer Vision. Springer, Cham, 2025.
> > > >
> > > > [4] Ranasinghe, Kanchana, et al. "Learning to localize objects improves spatial reasoning in visual-llms." Proceedings of the IEEE/CVF Conference on Computer Vision and Pattern Recognition. 2024.

---

> > > ### Author Response · Authors · 2024-11-26
> > > **Additional results on RT-2 Style + Aux**
> > >
> > > As suggested by the reviewer, we conducted experiments using the RT-2 Style dataset combined with the same auxiliary data we used in inBC + Aux(D). The results on VIMA-8k are presented in the table. This new setting significantly outperformed the RT-2 Style baseline, showing the effectiveness of the self-supervised auxiliary data we are introducing.
> > > Simultaneously, RT-2 Style (using specialized action tokens) still remained inferior to our inBC approach. This highlights the advantages of the proposed auxiliary data as well as the importance of conversation-style training of our approach, particularly when the training data is limited.
> > > | Model             | L1 | L2 | L3 |
> > > |--------------------|----------|----------|----------|
> > > | RT-2 Style        | 3.8      | 3.1      | 1.7      |
> > > | RT-2 Style + Aux (D) | 36.9     | 35.8     | 32.1     |
> > > | inBC | 57.3 |  46.2 | 42.9 |
> > > | inBC + Aux (D)    | 59.2     | 58.8     | 52.1     |

---

> > > ### Author Response · Authors · 2024-11-27
> > > **Kind reminder**
> > >
> > > Dear reviewer aFtC,
> > >
> > > This is a kind reminder that we tried our best to carefully address the concerns you raised in your review and we've provided the RT-2-Aux experiments you suggested.
> > > Please let us know what you think. If there are any remaining concerns or additional points/questions you'd like us to address further, please also let us know.
> > >
> > > Thank you once again for your valuable feedback and time.
> > >
> > > Best regards,
> > >
> > > Authors of paper 8021

---

> > > > ### Author Response · Authors · 2024-12-03
> > > >
> > > > Since the discussion period is ending soon, could you please let us know if there are any remaining concerns? If possible, could you consider the added RT-2-Aux experiments when finalizing your decision?
> > > >
> > > > Thanks,
> > > >
> > > > Authors of paper 8021

---

### Author Response · Authors · 2024-11-21
**General Rebuttal for All Reviewers**

We thank all the reviewers for their comments and suggestions. In this rebuttal post, we would like to highlight our contributions and clarify the relationship between our work and other previous works.


The focus of this paper is on "representation finetuning". We study how easily and quickly we can transfer learned representations in a VLM to representations necessary for a VLA model. We investigate what are the possible approaches to make such transfer/adaptation as quickly as possible (with limited data), and show that:
Image coordinate-based textual action representation has the potential to enable much faster robot learning. Especially when the robotics training data is limited, our design can better benefit from the general VLMs pretrained with internet data.
We also demonstrate that self-supervised auxiliary data can significantly benefit VLA model training, without any additional action labels or observations, which distinguishes our work from prior work.


Furthermore, during inference, our method efficiently generates actions directly as concise text, without the need to produce contextual or auxiliary text.


The new findings of our paper are orthogonal to other VLA models focusing on different aspects of the problem such as 3D manipulation. We believe the observations on VLM to VLA representation transfer we share with this paper are valuable not only to the robot learning community but also to the computer vision community.

---

### Author Response · Authors · 2024-12-04
**Interaction Summary and Collection of New Experiments**

Below is our summary of the interactions with the reviewers. We also added the collection of new experiments conducted during the discussion in this thread.

---

**Reviewer aFtC**

In the initial review, the reviewer mainly asked for experiments and clarifications on the following points:
1. Experiments on direct comparisons with existing VLAs in the experiments section
2. Further clarification on the similarities and differences between our method and other methods like RT-2, OpenVLA and so on
3. The use of auxiliary data and the way we generate it

Our first response, including additional experiments regarding RoboPoint as suggested by the reviewer and other clarifications, helped address some of the reviewer’s concerns. (https://openreview.net/forum?id=iVxxgZlXh6&noteId=rk1DD9bk72)

After that, the reviewer brought more comments/questions:
1. The connection between our method and the related methods discussed in section 2
2. Request for additional experiments on RT-2 + Aux
3. The risk of overfitting
4. The decision to use fixed template-based methods for data generation

Then we provided detailed clarifications on the questions the reviewer proposed and conducted new experiments on RT-2 + Aux as suggested by the reviewer. (https://openreview.net/forum?id=iVxxgZlXh6&noteId=bgfOzjkAO0)

After that, the reviewer did not respond and we assume we’ve addressed all the concerns raised.

---
**Reviewer wHkn**

In the initial review, the reviewer asked for further clarifications related to topics:

1. The detailed contribution of each auxiliary dataset
2. The difference between our method and RT-2 baselines, especially in the definition of the auxiliary data.

After our response (https://openreview.net/forum?id=iVxxgZlXh6&noteId=s4abgzJhS7), the reviewer acknowledged the ablations and the modeling choices and brought other concerns related to the generalizability, scalability, and novelty of our method compared to RT-2 with regard to the auxiliary objectives.

Then in this post (https://openreview.net/forum?id=iVxxgZlXh6&noteId=GQ7HUtqAxa), we highlighted our strong performance with limited training data, which is where the RT-2 baseline suffers. In addition, we emphasize the contribution/focus of our paper is on visual representation learning and its transfer, extending many recent computer vision research.

At this point, the reviewer raised the score from 5 to 6 and reminded us to discuss the limitations in greater detail.

In order to further clarify regarding the concerns on 3D manipulation, we show initial observations in this post (https://openreview.net/forum?id=iVxxgZlXh6&noteId=nGzYguFEpB) and we believe we showed the sufficient potential of LLaRA on 3D manipulation.

---
**Reviewer HVHL**

In the initial review, the reviewer brought concerns about the limited numerical representations in transformers; asked about the correctness of the data augmentation; asked for more clarifications regarding the experiment settings, and provided constructive feedback on the figures and other presentations.

We responded to these questions, especially how auxiliary data can help overcome these issues, and provide more implementation details in this post (https://openreview.net/forum?id=iVxxgZlXh6&noteId=D3vVH4eerT)

After that, the reviewer provided further feedback on Figure 5 and we updated Figure 5 to include the data originally presented in Table 9 (https://openreview.net/forum?id=iVxxgZlXh6&noteId=U5IDkLMoY2).

The reviewer acknowledged the new figure, and agreed that our responses `have greatly clarified the motivation and key contributions of the work.`

---

> ### Author Response · Authors · 2024-12-04
> **Continued Interaction Summary**
>
> **Reviewer 3YdQ**
>
> Initially, the reviewer gave our submission a score of 3 with a confidence of 5. At that time, the majority of the criticisms were due to several misunderstandings and comparisons to a paper called ECoT, which, per the reviewer guidelines, should be considered as concurrent/contemporaneous work.
>
> Then the reviewer marked most of the original comments related to ECoT to be **ignored**, after being reminded by AC (https://openreview.net/forum?id=iVxxgZlXh6&noteId=uTNMaoxCs2). We started to prepare our rebuttal to the revised comment, mainly focusing on clarifying the misunderstandings the reviewer has  (https://openreview.net/forum?id=iVxxgZlXh6&noteId=8wM4kRO0X3). For example, we clarified with detailed explanations that the VIMA benchmarks and real-world robot experiments we adopted are very challenging and could not be solved by copying and pasting values. We also added experiments including new real-world robot experiments to better support our claims.
>
> The reviewer acknowledged the misunderstandings and reduced his/her confidence score from 5 to 3 while maintaining the review score of 3.
>
> After the above interactions, we further provided additional new experiments to clarify more issues brought up by this reviewer in the initial review, like three new real-world tasks, and additional experiments with unstable actions. (https://openreview.net/forum?id=iVxxgZlXh6&noteId=KcBFB7diHh)
>
> We also provide more background context on self-supervised (auxiliary) data in computer vision and how modern vision-language models (VLMs) have been benefiting from them. We emphasized that our goal is to replicate the recent success of the computer vision paradigm (i.e., data curation) in robotics. We clarified that our technical contribution is orthogonal and complementary to previous research focusing more on 3D manipulation.
>
> After completing this round of additional experiments and addressing all the questions raised, the reviewer expressed a willingness to increase the score by one level. However, after realizing that there is no "weak rejection" option in ICLR, the reviewer left the score unchanged and introduced new concerns, such as:
>
> 1. “Robotics domains in 2D action spaces are fundamentally very limited.”
> 2. “This is critically very different from the more general 6DoF visuomotor manipulation that has recently gotten a lot of attention.”
>
> We again clarify that the focus of our paper is on visual representation learning and we emphasized that our action space is something we inherit from VIMA and the new findings of our paper are orthogonal to others focusing on different aspects of the problem such as 3D manipulation. (https://openreview.net/forum?id=iVxxgZlXh6&noteId=bOw2As73dl)
>
> Furthermore, corresponding to the reviewer's concern, we invested significant effort (over 3\~4 days) to extend LLaRA to 3D manipulation tasks. Without an extensive amount of tuning due to the limited time, we still found that our method was capable of solving such 3D manipulation tasks. We believe we showed the sufficient potential of transferring LLaRA’s success on 3D manipulation as well. (https://openreview.net/forum?id=iVxxgZlXh6&noteId=9kOpW5eHtx)
>
> In addition, we clarified again the contributions and the intuition of our paper (https://openreview.net/forum?id=iVxxgZlXh6&noteId=bOw2As73dl) and we reported more observations on D-inBC variants and RT-2 style baselines with more advantages (https://openreview.net/forum?id=iVxxgZlXh6&noteId=rk1wG6yt3e)
>
> Since we reported these observations, we did not get any particular response from the reviewer.

---

> ### Author Response · Authors · 2024-12-04
> **Summary of newly added experimental results during the rebuttal phase**
>
> **Summary of newly added experimental results during the rebuttal phase**
>
> 1. Introduced a new D-inBC variant (*D-inBC(L)*) that uses natural language descriptions of the reference image instead of structured object lists (3YdQ).
> https://openreview.net/forum?id=iVxxgZlXh6&noteId=rk1wG6yt3e
>
> 2. Extended real-world experiments to include new LLaRA settings, such as *inBC + Aux* and *D-inBC(L)*.
> https://openreview.net/forum?id=iVxxgZlXh6&noteId=KcBFB7diHh
>
> 3. Extended real-world experiments on three more new unseen tasks.
> https://openreview.net/forum?id=iVxxgZlXh6&noteId=vFV43wzcHu
>
> 4. Benchmarked an additional VLA model, RoboPoint (aFtC), for comparison.
> https://openreview.net/forum?id=iVxxgZlXh6&noteId=rk1DD9bk72
>
> 5. Proposed a new setting, *RT-2 Style + Aux*, and verified the value of incorporating self-supervised auxiliary data (aFtC).
> https://openreview.net/forum?id=iVxxgZlXh6&noteId=N0A9Hok52M
>
> 6. Added a setting, *RT-2 Style (2R)*, which uses two action tokens to represent rotations, significantly expanding the action space compared to our method. However, it still significantly underperforms our method (3YdQ).
> https://openreview.net/forum?id=iVxxgZlXh6&noteId=rk1wG6yt3e
>
> 7. Tested the robustness of our model in failure grasp scenarios (3YdQ).
> https://openreview.net/forum?id=iVxxgZlXh6&noteId=lnjm2xo9ra
>
> 8. Presented initial observations on extending LLaRA to 3D manipulation tasks, demonstrating the potential of our method for broader applications (wHkn, 3YdQ).
> https://openreview.net/forum?id=iVxxgZlXh6&noteId=9kOpW5eHtx
>
> At this stage, we believe we have thoroughly addressed all the concerns raised by the reviewers. We kindly request that the reviewers take all of the discussions above into consideration when finalizing their review.
>
> With best regards,
>
> Authors of submission 8021

---

### Meta-Review · Area_Chair_jE2G · 2024-12-18

**Metareview:**

The paper presents LLaRA (Large Language and Robotics Assistant), a framework that transforms behavior cloning data into conversation-style instruction tuning data for robot learning. The key innovation is using self-supervised auxiliary data generation from existing robot trajectories to enhance visual-language model performance, without requiring additional action labels or observations.

Reviewer 3YdQ provided the most extensive feedback and engagement throughout the review process. Their initial review heavily emphasized similarities to ECoT, though this was later acknowledged as concurrent work per ICLR guidelines. More fundamentally, 3YdQ raised persistent concerns about the limitations of 2D pick-and-place action spaces, arguing that success in such domains primarily requires object localization rather than complex manipulation skills needed for 6-DoF tasks. While these concerns about 2D action spaces are thoughtfully argued, they perhaps go beyond evaluating this specific paper's contributions, as VIMA is an established benchmark in the robotics community (published at ICML 2023). Given that 2D manipulation is a legitimate and active research area, we should not reject the paper solely on these grounds. Therefore, while we value 3YdQ's detailed technical feedback, we place greater weight on the evaluations from the other three reviewers who assessed the paper's contributions within its intended scope.

Reviewer wHkn focused on questions about generalizability and comparisons to RT-2, particularly regarding auxiliary objectives. The authors' responses demonstrating strong performance with limited training data partially addressed these concerns, leading to an increased score from 5 to 6. However, questions remained about broader applicability beyond the demonstrated tasks.

Reviewer HVHL raised technical concerns about numerical representations in transformers and data augmentation validation. Through discussion, they were satisfied with the authors' clarifications about their approach to handling these challenges. The discussion culminated in agreement that the responses "greatly clarified the motivation and key contributions."

Reviewer aFtC questioned the novelty compared to existing approaches and requested additional experimental comparisons. The authors provided new results comparing to RoboPoint and RT-2 variants, though the reviewer's final position remains somewhat ambiguous.

Near the end of the discussion period, the authors presented preliminary results extending LLaRA to 3D manipulation tasks using the LIBERO benchmark. While these results showed promise, they were introduced very late in the process without sufficient time for thorough reviewer evaluation.

While placing appropriate weight on the various reviewer concerns, and acknowledging that 2D manipulation research remains valuable, this AC leans towards accepting the paper.

**Additional Comments On Reviewer Discussion:**

Included in metareview.

---

### Decision · Program_Chairs · 2025-01-22

Accept (Poster)